# Commensal consortia decolonize Enterobacteriaceae via ecological control

Munehiro Furuichi[1,2,14], Takaaki Kawaguchi[1,2,14], Marie-Madlen Pust[3,4,14], Keiko Yasuma-Mitobe[1,14], Damian R. Plichta[3], Naomi Hasegawa[1], Takashi Ohya[1,2], Shakti K. Bhattarai[5], Satoshi Sasajima[1], Yoshimasa Aoto[6], Timur Tuganbaev[1,7], Mizuki Yaginuma[1], Masahiro Ueda[2,6], Nobuyuki Okahashi[2,8,9], Kimiko Amafuji[6], Yuko Kiridoshi[6], Kayoko Sugita[1], Martin Stražar[3], Julian Avila-Pacheco[3], Kerry Pierce[3], Clary B. Clish[3], Ashwin N. Skelly[1], Masahira Hattori[2,10], Nobuhiro Nakamoto[11], Silvia Caballero[12], Jason M. Norman[12], Bernat Olle[12], Takeshi Tanoue[1,2], Wataru Suda[2,10], Makoto Arita[2,7,9], Vanni Bucci[5], Koji Atarashi[1,2,7], Ramnik J. Xavier[3,4,13 ✉] & Kenya Honda[1,2,7 ✉]

Persistent colonization and outgrowth of potentially pathogenic organisms in the intestine can result from long-term antibiotic use or inflammatory conditions, and may perpetuate dysregulated immunity and tissue damage[1,2]. Gram-negative Enterobacteriaceae gut pathobionts are particularly recalcitrant to conventional antibiotic treatment[3,4], although an emerging body of evidence suggests that manipulation of the commensal microbiota may be a practical alternative therapeutic strategy[5–7]. Here we isolated and down-selected commensal bacterial consortia from stool samples from healthy humans that could strongly and specifically suppress intestinal Enterobacteriaceae. One of the elaborated consortia, comprising 18 commensal strains, effectively controlled ecological niches by regulating gluconate availability, thereby re-establishing colonization resistance and alleviating *Klebsiella*- and *Escherichia*-driven intestinal inflammation in mice. Harnessing these activities in the form of live bacterial therapies may represent a promising solution to combat the growing threat of proinflammatory, antimicrobial-resistant Enterobacteriaceae infection.

The discovery and clinical application of potent antimicrobial agents has been a double-edged sword, saving countless lives worldwide while simultaneously spurring the evolution and expansion of multidrug-resistant organisms that have become an important threat to global health. In particular, Gram-negative Enterobacteriaceae such as *Escherichia* and *Klebsiella* species have emerged as important multidrug-resistant nosocomial pathogens for which limited therapeutic options exist[3,4]. Broad-spectrum antibiotics are often used to treat multidrug-resistant Enterobacteriaceae, which may further aggravate dysbiosis and thus impair colonization resistance. In addition to antibiotic treatment, inflammatory conditions also predispose to Enterobacteriaceae outgrowth[1,2,8–12]. Indeed, inflammatory bowel disease (IBD) is often associated with dysbiosis and enrichment of Enterobacteriaceae[13–15], and persistence of Enterobacteriaceae helps to perpetuate intestinal inflammation and nosocomial infections with other microbes[16–19]. Moreover, the proliferation of Enterobacteriaceae in the gut constitutes a major risk factor for systemic infection and is

linked to increased mortality rates[20,21]. Several clinical and preclinical studies have found faecal microbiota transplantation (FMT) to be efficacious in reducing levels of Enterobacteriaceae in the intestine[5–7]. Therefore, manipulation of the gut microbiota represents a promising approach to treat IBD and infection with multidrug-resistant organisms. However, FMT therapies have exhibited mixed results, safety concerns and production impracticalities secondary to inherent batch-to-batch variability[5]. Overcoming these limitations requires identification of specific bacteria or consortia that are capable of decolonizing Enterobacteriaceae and elucidation of their mechanisms of action.

## An 18-strain consortium can decolonize *Klebsiella*

*Klebsiella* species comprise a major aetiology of nosocomial infections[3,4]. We previously isolated multidrug-resistant *Klebsiella* strains from patients with IBD, including the *Klebsiella pneumoniae* Kp-2H7 strain, which can expand and persist in the intestine in the setting of

[1]Department of Microbiology and Immunology, Keio University School of Medicine, Tokyo, Japan. [2]RIKEN Center for Integrative Medical Sciences, Yokohama, Japan. [3]Infectious Disease and Microbiome Program, Broad Institute of MIT and Harvard, Cambridge, MA, USA. [4]Center for Computational and Integrative Biology, Massachusetts General Hospital and Harvard Medical School, Boston, MA, USA. [5]Department of Microbiology and Physiological Systems, Program in Microbiome Dynamics, UMass Chan Medical School, Worcester, MA, USA. [6]JSR-Keio University Medical and Chemical Innovation Center, Keio University School of Medicine, Tokyo, Japan. [7]Human Biology Microbiome Quantum Research Center (Bio2Q), Keio University, Tokyo, Japan. [8]Department of Bioinformatic Engineering, Graduate School of Information Science and Technology, Osaka University, Osaka, Japan. [9]Division of Physiological Chemistry and Metabolism, Graduate School of Pharmaceutical Sciences, Keio University, Tokyo, Japan. [10]Cooperative Major in Advanced Health Science, Graduate School of Advanced Science and Engineering, Waseda University, Tokyo, Japan. [11]Division of Gastroenterology and Hepatology, Department of Internal Medicine, Keio University School of Medicine, Tokyo, Japan. [12]Vedanta Biosciences, Cambridge, MA, USA. [13]Department of Molecular Biology, Massachusetts General Hospital, Boston, MA, USA. [14]These authors contributed equally: Munehiro Furuichi, Takaaki Kawaguchi, Marie-Madlen Pust, Keiko Yasuma-Mitobe. ✉e-mail: xavier@molbio.mgh.harvard.edu; kenya@keio.jp

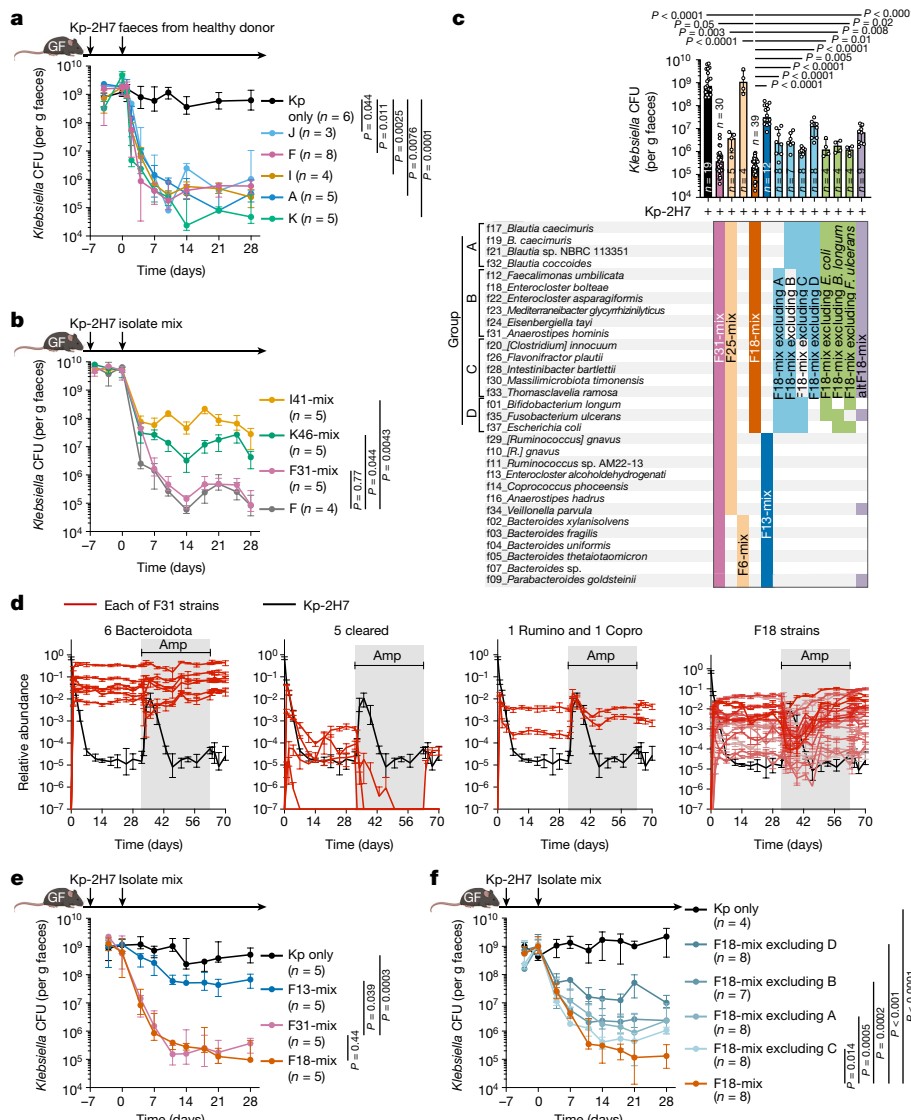

**Fig. 1 | Elaboration of an 18-strain-consortium capable of decolonizing**
***Klebsiella.* a–c,e,f**, GF B6 mice were monocolonized with Kp-2H7, followed by
oral administration of stool samples from one of five healthy human donors
(A, F, I, J or K) (**a**) or the indicated mixture of bacterial isolates (**b,c,e,f**). Faecal
CFUs of Kp-2H7 throughout the experiment (**a,b,e,f**) or on day 28 (**c**). **d**, GF mice
(*n* = 5) were monocolonized with Kp-2H7 (day −7), treated with F31-mix (day 0)
and then given ampicillin (200 mg l⁻¹) via drinking water (days 32 to 63).

The abundance of each of the 31 strains was determined by quantitative
PCR (qPCR) in two technical replicates and average data are shown. Rumino,
*Ruminococcus*; Copro, *Coprococcus*. See also Extended Data Fig. 2. Data in
**a**–**c,f** are median ± interquartile range (IQR) of representative data from two
independent experiments with similar results. The day 28 data are compared
by Kruskal–Wallis test using the Benjamini–Hochberg correction for multiple
comparisons.

antibiotic-induced dysbiosis and promote T helper 1 (T_H1) cell-mediated
inflammation[17]. We set out to identify human gut commensals that pro-
mote decolonization of Kp-2H7 using the strategy outlined in Extended
Data Fig. 1a. Germ-free (GF) C57BL/6 (B6) mice were monocolonized
with Kp-2H7 and then orally inoculated with a stool sample from one of
five healthy Japanese human donors (A, F, I, J or K). Efficacy of Kp-2H7
decolonization by FMT from each donor was examined by longitudi-
nally quantifying faecal Kp-2H7 colony-forming units (CFUs). FMT
from all donors resulted in a 3 to 4 log reduction in Kp-2H7 abundance
(Fig. 1a). We cultured stool samples from donors F, I and K using 6 dif-
ferent types of media, and isolated 37 strains (31 unique strains when
deduplicated) from donor F, 41 strains from donor I, and 46 strains
from donor K based on 16S ribosomal RNA (rRNA) gene sequencing
followed by whole-genome sequence analysis (Extended Data Fig. 1b
and Supplementary Table 1). A mixture of bacterial strains isolated
from each donor was inoculated into Kp-2H7-monocolonized mice to
test *Klebsiella*-decolonization capacity. The mixture of 31 strains from

donor F (designated F31-mix) was most effective, and the magnitude
and kinetics of Kp-2H7 reduction were similar to those induced by
donor F faecal microbiota (Fig. 1b,c).

To identify a minimal effector consortium from F31-mix, mice mono-
colonized with Kp-2H7 were treated with F31-mix and then given ampi-
cillin via drinking water to perturb microbiota homeostasis (Fig. 1d and
Extended Data Fig. 2a). Ampicillin treatment resulted in a transient
surge in the abundance of Kp-2H7 (which carries β-lactamase[17]), whereas
the 31 strains showed variable trajectories. The majority of Bacillota
strains exhibited an inverse abundance pattern compared with Kp-2H7,
whereas Bacteroidota strains remained largely unchanged (Fig. 1d
and Extended Data Fig. 2a). We thus divided F31-mix into two groups:
6 Bacteroidota strains (F6-mix) and 25 other strains (F25-mix). F25-mix
treatment reduced Kp-2H7 abundance to a similar extent as did F31-mix,
whereas F6-mix had no effect (Fig. 1c and Extended Data Fig. 3a). From
F25-mix, we excluded five strains that were cleared during ampicillin
treatment and two strains (one *Ruminococcus* and one *Coprococcus*)

that exhibited trajectories similar to Kp-2H7 (Fig. 1d and Extended Data Fig. 2a). A Spearman's rank correlation test indicated that most of the remaining 18 strains were significantly inversely associated with Kp-2H7 abundance (Extended Data Fig. 2b). We thus tested the activity of these 18 strains (F18-mix) and observed a robust reduction in faecal Kp-2H7 CFUs, with similar magnitude and kinetics to mice treated with the parental F31-mix (Fig. 1c,e). Full-length 16S rRNA gene sequencing of longitudinally collected faecal samples confirmed successful colonization by all 18 strains without contamination, as well as persistent suppression of Kp-2H7 (Extended Data Fig. 3b). Administration of the 13 strains (F13-mix) that were excluded from F31-mix was far less effective at decolonizing Kp-2H7 than F18-mix (Fig. 1c,e and Extended Data Fig. 3b). In addition to its capacity to decolonize *Klebsiella*, F18-mix exerted potent colonization resistance activity against Kp-2H7, significantly outperforming F13-mix (Extended Data Fig. 3c).

We also generated 7 derivative consortia of F18-mix by subtracting various combinations of bacterial species, yielding consortia that ranged in size from 12 to 17 strains. These consortia exhibited varying capacities to decolonize Kp-2H7 in vivo, although none was as effective as the full F18-mix (Fig. 1c,f). In particular, derivative subsets lacking either group A (4 *Blautia* strains), B (6 Lachnospiraceae strains), C (5 Bacillota strains) or D (3 strains from other phyla) all exhibited reduced decolonization capacity compared with the full F18-mix (Fig. 1c,f). These results suggest that the F18 members act cooperatively and that all phylogenetic components are required to achieve maximal Kp-2H7 suppression (Supplementary Discussion 1).

## F18-mix preferentially suppresses Enterobacteriaceae

We next examined the capacity of F18-mix to decolonize extended-spectrum β-lactamase (ESBL)$^+$ *Escherichia coli* and carbapenemase (CPM)$^+$ *K. pneumoniae* strains, both of which are exigent threats to public health[4,22]. For comparison, we additionally analysed F31-mix, F13-mix, I41-mix and K46-mix, as well as complete faecal microbiota from donor F. F18-mix was highly effective at suppressing intestinal ESBL$^+$ *E. coli* and CPM$^+$ *K. pneumoniae* in gnotobiotic mouse, with similar magnitude and kinetics as donor F faecal microbiota, achieving 3 to 4 log reductions in faecal CFUs (Fig. 2a). F18-mix was also highly effective at decolonizing *Klebsiella aerogenes* (strain Ka-11E12) and adherent-invasive *E. coli* (AIEC, strain LF82), both of which have been implicated in IBD pathogenesis[17,23]. K46-mix was as efficacious as F18-mix at decolonizing the tested *Klebsiella* and *E. coli* strains, whereas F13-mix and I41-mix were far less effective (Fig. 2a). Notably, members of Pseudomonadota other than Enterobacteriaceae, including *Campylobacter upsaliensis* and *Pseudomonas aeruginosa*, were largely resistant to decolonization by all tested consortia including F18-mix (Extended Data Fig. 3d).

We additionally investigated the effects of donor-derived consortia from donors F, K and I on Gram-positive pathogens, including vancomycin-resistant *Enterococcus faecium* (VRE) and *Clostridioides difficile*, which are also considered to be high-priority multidrug-resistant threats[4]. K46-mix was effective against both VRE (ATCC 700221) and *C. difficile* (BAA1382), whereas I41-mix was not (Fig. 2a). Notably, F31-mix was as efficacious as K46-mix at decolonizing VRE, although this effect was blunted against *C. difficile*. By contrast, F18-mix was less effective against these Gram-positive pathogens (Fig. 2a). Collectively, these results indicate that the process of narrowing down F31-mix to F18-mix led to the selection of commensals that were preferentially able to decolonize Enterobacteriaceae.

## Effect of F18-mix on commensals and colitis

Next, we explored whether F18-mix colonization would affect the abundance of other commensals. We selected seven commensal strains from our culture collection and simultaneously inoculated these strains along with Kp-2H7 into GF mice. We then administered F18-mix by oral gavage and monitored the faecal abundance of each strain over time. All F18-mix-derived strains successfully colonized, accompanied by a notable decrease in Kp-2H7 abundance. The levels of commensal Bacillota and Bacteroidota strains remained largely stable, although there was a reduction in the levels of low-abundance strains (*Bifidobacterium* and *Collinsella*) among the seven strains (Extended Data Fig. 3e,f). To examine the effect of F18-mix on commensals in the setting of a more complex microbiota, GF mice were colonized with either I41-mix, K46-mix or both of these consortia together along with a *Clostridium scindens* strain (a total of 88 strains), followed by oral administration of F18-mix. Strains derived from donors I or K remained largely stable, although some low-abundance members, such as *Bifidobacterium*, *Collinsella* and *Megasphaera*, showed reductions (Extended Data Fig. 3g, Supplementary Table 2 and Supplementary Discussion 2). Together, these data suggest that F18-mix can reduce *Klebsiella* levels in the intestine without significantly affecting commensal communities.

To further probe clinical translatability, we tested the ability of F18-mix to decolonize Enterobacteriaceae in the context of a complex microbiota associated with IBD. GF mice were inoculated with faecal microbiota from patients with Crohn's disease (CD15) or ulcerative colitis (UC5) that exhibited enrichment of either *K. pneumoniae* or ESBL$^+$ *E. coli*, respectively (Fig. 2b,c). The mice were then treated with vancomycin to generate ecological niches amenable to F18-mix engraftment, followed by F18-mix gavage, thus mimicking a potential clinical treatment regimen using live biotherapeutic products[24]. Gut microbiome composition was examined by full-length 16S rRNA gene sequencing (Fig. 2b,c) and *K. pneumoniae* or *E. coli* burden was determined by counting faecal CFUs (Extended Data Fig. 4a,b). All F18 strains engrafted successfully within the IBD-associated microbial communities, resulting in a concomitant increase in microbial diversity and suppression of *K. pneumoniae* and *E. coli* (Fig. 2b,c and Extended Data Fig. 4a,b). As expected, vancomycin treatment alone did not suppress these pathobionts. Thus, the F18-mix can exert anti-Enterobacteriaceae activity in the context of several clinically relevant, complex microbiota.

Kp-2H7 is a strong inducer of intestinal T$_H$1 cells and can act as a colitogenic pathobiont in the context of a genetically susceptible host, such as interleukin-10-deficient (*Il10$^{-/-}$*) mice[17]. To test the effect of F18-mix on Kp-2H7-driven colitis, GF *Il10$^{-/-}$* mice were monocolonized with Kp-2H7 and then orally inoculated with either F18-mix or F13-mix. Similarly to wild-type mice, *Il10$^{-/-}$* mice treated with F18-mix but not F13-mix showed a 3 to 4 log reduction in Kp-2H7 abundance (Extended Data Fig. 4c). In contrast to F13-mix, F18-mix significantly reduced histological scores of colitis, faecal levels of lipocalin-2 and calprotectin (both of which serve as sensitive biomarkers of intestinal inflammation), and T$_H$1 cell frequency (Extended Data Fig. 4d–g). Similar results were observed when *Il10$^{-/-}$* GF mice were colonized with a ulcerative colitis-associated microbiota containing ESBL$^+$ *E. coli* (UC5). F18-mix treatment successfully decolonized ESBL$^+$ *E. coli* and protected against intestinal inflammation (Fig. 2d–f). In sum, F18-mix can reduce intestinal Enterobacteriaceae burden and alleviate IBD-like inflammation without disrupting the gut commensal community, suggesting high translational potential.

## Exploring mechanisms of *Klebsiella* decolonization

Next, we sought to reveal the mechanisms underlying F18-mix-mediated *Klebsiella* suppression. F18-mix was able to efficiently reduce intestinal Kp-2H7 burden in mice that were deficient in IFNγ response (*Ifngr$^{-/-}$*), Toll-like receptor signalling (*Myd88$^{-/-}$ Ticam1$^{-/-}$* (*Ticam1* is also known as *Trif*)), or innate and adaptive lymphocytes (*Rag2$^{-/-}$ Il2rg$^{-/-}$*) (Extended Data Fig. 5a,b), suggesting a mechanism that is independent of the canonical immune system. We also assessed intestinal epithelial cell transcriptomic responses in GF mice colonized with effective (F31-mix or F18-mix) versus less-effective (F13-mix) microbial consortia.

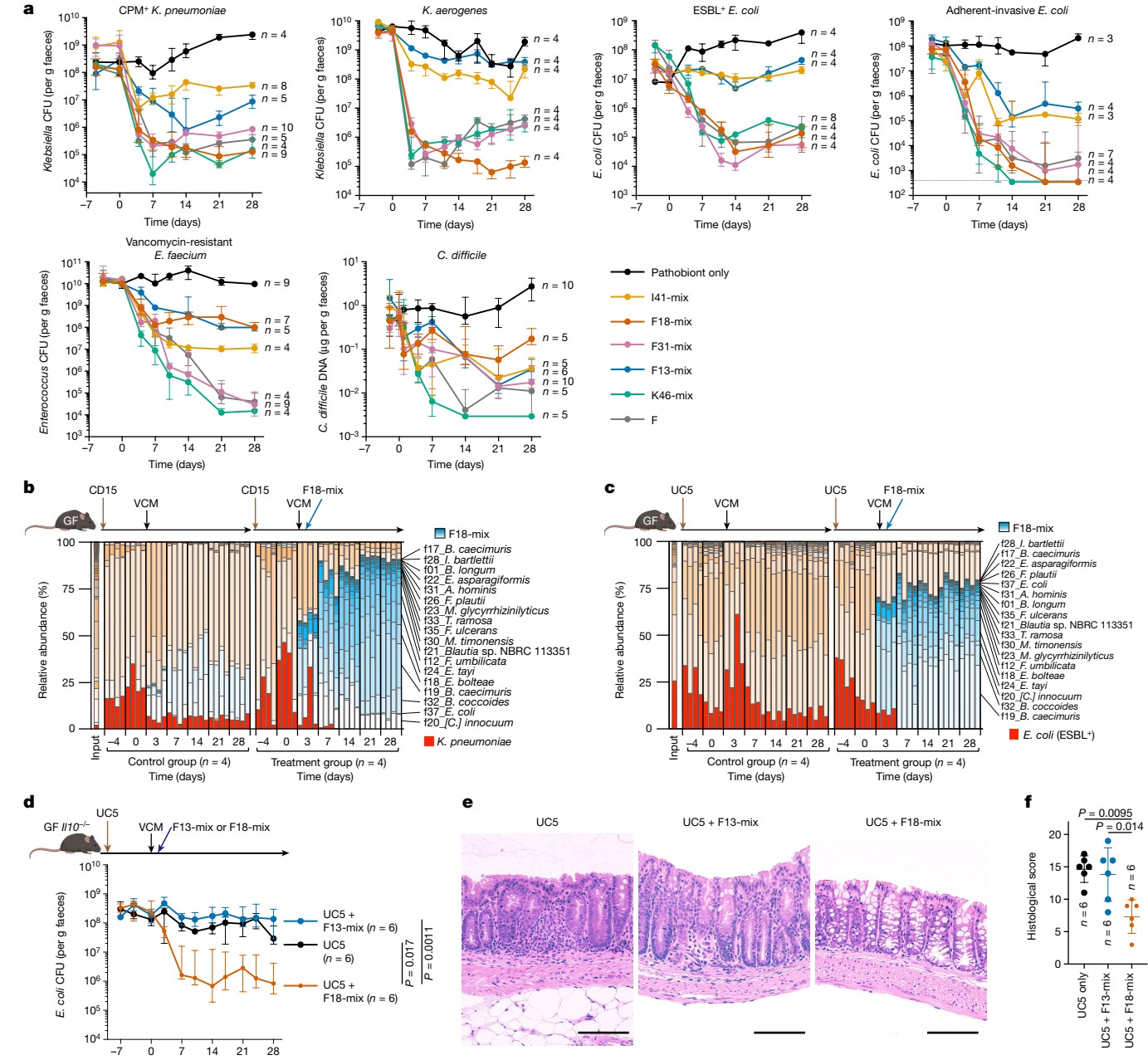

**Fig. 2 | F18-mix controls intestinal pathogens and colitis. a**, GF B6 mice (*n* = 3–10 per group) were monocolonized with the indicated pathogenic or antibiotic-resistant strain, and then treated with the indicated bacterial mixture. Faecal pathobiont load was examined by counting CFUs or by qPCR of bacterial DNA (for *C. difficile*). **b,c**, GF B6 mice were colonized with faecal microbiota from a patient with Crohn's disease (CD15) containing a high level of *K. pneumoniae* (**b**) or from a patient with ulcerative colitis (UC5) containing ESBL⁺ *E. coli* (**c**). All mice were subsequently treated with vancomycin (VCM), and half of the mice received oral F18-mix administration four times over the

next two days. Full-length 16S rRNA gene sequencing was performed on faecal samples to determine the relative abundance of detected strains. **d–f**, GF *Il10*⁻/⁻ mice (*n* = 6 per group) were colonized with UC5 microbiota and then treated with F18-mix, F13-mix or vehicle control; faecal CFUs of ESBL⁺ *E. coli* throughout the experiment (**d**), representative haematoxylin and eosin staining of the colon on day 28 (**e**; scale bars, 200 μm) and histological colitis scores on day 28 (**f**) are shown. Data in **a,d,f**, are median ± IQR and are compared by Kruskal–Wallis test using the Benjamini–Hochberg correction for multiple comparisons.

However, we could not discern consistent shifts in genes or pathways that were specifically modulated by the effective consortia, including those encoding previously reported putative anti-*Klebsiella* gene products such as antimicrobial peptides and PPARγ-regulated molecules[25,26] (Extended Data Fig. 5c). Therefore, although host factors could still have a role in F18-mix-mediated Kp-2H7 decolonization, we focused our line of inquiry on interbacterial interactions.

Caecal suspensions from mice colonized with F31-mix or F18-mix, but not F13-mix, strongly suppressed the growth of Kp-2H7 upon in vitro

anaerobic coculture (Extended Data Fig. 6a). This suppressive effect was abrogated when coculture was performed under aerobic conditions or when the caecal suspension was filtered or heat-inactivated, suggesting that live F18-mix activity is required. Liquid chromatography–mass spectrometry (LC–MS) analysis revealed that mice colonized with effective consortia (F31-mix or F18-mix) contained higher caecal levels of 4-hydroxybenzoic acid (4-HBA), cholic acid, acetate and butyrate than mice colonized with less-effective consortia (F13-mix or F18-mix minus phylogenetic groups A, B, C or D) or Kp-2H7 alone

(Extended Data Fig. 6b). These molecules have previously been implicated in colonization resistance mechanisms against Enterobacteriaceae[27–29]. However, cholic acid was completely ineffective at all concentrations tested, and 4-HBA, acetate and butyrate exerted only weakly suppressive effects, requiring a high concentration (and in the case of acetate and butyrate, low pH) to inhibit Kp-2H7 growth in vitro (Extended Data Fig. 6c). Furthermore, high-level butyrate exposure via tributyrin feeding did not affect the kinetics or magnitude of F18-mix- or F13-mix-mediated Kp-2H7 suppression in vivo (Extended Data Fig. 6d).

We next examined whether F18-mix acted by modulating previously reported processes associated with Enterobacteriaceae intestinal fitness. However, the deletion of Kp-2H7 genes associated with quorum sensing (*lsrA*, *lsrC*, *lsrB*, *lsrD* and *lsrR*)[30], biofilm formation (*csgD*)[31], stress response (*rpoS*)[31] and nitrate respiration (*narG*, *narZ*)[1,8] did not significantly affect sensitivity to F18-mix-mediated decolonization (Extended Data Fig. 7a). These results suggest that F18-mix exerts anti-*Klebsiella* activity via alternative mechanisms.

## Suppression of *Klebsiella* via gluconate competition

We next compared the *Klebsiella* transcriptome in mice colonized with Kp-2H7 plus F18-mix versus Kp-2H7 alone. Co-colonization with F18-mix markedly shifted the Kp-2H7 transcriptional landscape, affecting pathways linked to carbon and amino acid metabolism, as well as the phosphotransferase system (Extended Data Fig. 7b). Notably, monocolonized Kp-2H7 exhibited robust expression of genes involved in the metabolism of various carbohydrates, including gluconate, glucose, xylose, glucuronate, arabinose, galacturonate and ribose (Extended Data Fig. 7c). By contrast, the presence of F18-mix attenuated expression of these genes (Extended Data Fig. 7c), suggesting that F18-mix might inhibit *Klebsiella* growth by competing for carbohydrates. This hypothesis aligns with previous studies that underscore the role of various carbohydrates in defining Enterobacteriaceae niches in the gut, including glucose[32], glucarate[33], galactitol[34–36] and β-glucosidic sugars[37], among others[38,39].

To identify Kp-2H7 genes that affect its intestinal fitness upon co-colonization with F18-mix, we generated a highly saturated transposon insertion mutant library with approximately $8 \times 10^5$ distinct mutations in Kp-2H7, which is estimated to contain more than 100 insertion mutants per gene (Fig. 3a). GF mice were inoculated with a Kp-2H7 pool containing all transposon insertion mutants (Kp-TPs) followed by oral gavage with F18-mix or F13-mix. Faecal samples were collected longitudinally and analysed by transposon insertion sequencing[40,41] (Tn-seq). After treatment with F18-mix, 194 Kp-2H7 mutants displayed significantly reduced fitness, many of which were deficient in carbohydrate and amino acid metabolism (Fig. 3b and Supplementary Table 3). Specifically, mutations in genes involved in the metabolism of carbohydrates such as gluconate, glucose and fructose accelerated *Klebsiella* suppression in F18-mix-treated mice relative to untreated and F13-mix-treated mice (Supplementary Table 3), confirming that F18-mix competes with Kp-2H7 for these carbon sources.

Comparing the relative abundance of Kp-TPs within each mouse revealed that those deficient in the HTH-type transcriptional regulator gene (*gntR*) were cleared by day 10 in F18-mix-treated mice, but persisted at high levels in untreated and F13-mix-treated mice (Fig. 3c). GntR is reported to function as a gluconate operon repressor[42,43]. In *Klebsiella* and other Enterobacteriaceae strains, gluconate is first phosphorylated by gluconate kinase (GntK/IdnK) to 6-phosphogluconate (gluconate-6P), which is in turn reduced to 2-keto-3-deoxy-6-phosphogluconate (KD6PG) by gluconate-6P dehydratase (Edd) and eventually converted into pyruvate and glyceraldehyde 3-phosphate (GA3P) by KD6PG aldolase (Eda) (in the Entner–Doudoroff pathway[44]; Fig. 3d). GntR represses genes encoding gluconate transporter (*gntU*), gluconate kinase (*gntK/idnK*), and Entner–Doudoroff pathway enzymes

(*edd* and *eda*) in *Klebsiella* when environmental gluconate is limited[42,43] (Fig. 3d). Thus, the marked difference in clearance of Kp-2H7 *gntR* mutants when co-colonized with F18- versus F13-mix suggested that gluconate metabolism may be a critical mechanistic underpinning of the observed F18-mix-mediated *Klebsiella* suppression. Consistently, transcriptomic analysis of Kp-2H7 genes in faecal samples revealed severe suppression of gluconate operon genes (*gntK*, *gntU* and *edd*) following treatment with F18-mix, but not F13-mix (Extended Data Fig. 7d). We generated a *gntK*-deficient isogenic Kp-2H7 mutant and examined its fitness in vitro and in vivo (Fig. 3e and Extended Data Fig. 8a). As expected, growth of Kp-2H7 Δ*gntK* was stunted when cultured with gluconate as the sole carbon source, whereas it was not affected in glucose-supplemented minimal media (Extended Data Fig. 8a). We inoculated GF mice with a 1:1 mixture of wild-type and Δ*gntK* Kp-2H7 and then treated them with F18-mix or F13-mix. Compared with wild-type Kp-2H7, Δ*gntK* Kp-2H7 exhibited heightened sensitivity to decolonization by the otherwise poorly efficacious F13-mix (yielding a 3 to 4 log reduction) as well as by F18-mix (yielding a 5 to 6 log reduction) (Fig. 3e), suggesting that the regulation of GntK-dependent gluconate metabolism is involved in *Klebsiella* suppression in the intestine.

We also generated a Δ*gntR* Kp-2H7 strain and studied it as above. The Δ*gntR* strain was resistant to F13-mix treatment but more sensitive to F18-mix treatment compared with wild-type Kp-2H7, consistent with findings from our transposon mutant experiments (Extended Data Fig. 8b). Transcriptomic profiling and in vitro culture studies of the Δ*gntR* Kp-2H7 strain suggested that GntR may have a dual role: suppressing genes involved in gluconate metabolism while enhancing the expression of genes involved in glucosamine metabolism (Extended Data Fig. 8a,c). This dual functionality could explain the context-dependent differential fitness of Δ*gntR* Kp-2H7: whereas *gntR* deletion enhances gluconate-driven growth, it may impair growth under gluconate-deprived conditions due to futile expression of gluconate operon genes and the impaired expression of glucosamine metabolism genes (Supplementary Discussion 3).

Quantitative LC–MS of faecal samples from GF mice on a regular CL-2 diet identified gluconate as one of the most abundant carbohydrates, whereas glucosamine levels were approximately 20-fold lower (Fig. 3f). Substantial amounts of faecal gluconate were observed even after mice were fed a gluconate-deficient AIN93G diet (Extended Data Fig. 9a), implying that both dietary intake and host production contribute to the pool of intestinal gluconate. Faecal gluconate concentration decreased by 1 log in Kp-2H7-monocolonized mice and by 2 logs in F18-mix-colonized mice, but was only marginally affected in F13-mix-colonized mice (Fig. 3g), supporting the hypothesis that F18-mix mediates *Klebsiella* suppression via gluconate deprivation. We also examined gluconate and other major carbon sources in the faeces of mice colonized with Kp-2H7, F18-mix, F13-mix, K46-mix or I41-mix. Gluconate levels were greatly reduced in mice colonized with effective consortia (F18-mix and K46-mix), and only slightly reduced in mice colonized with less-effective consortia (I41-mix and F13-mix). This pattern was unique to gluconate, as no other carbohydrate exhibited such a clear association with *Klebsiella*-decolonization capability (Extended Data Fig. 9b). Moreover, dietary gluconate deprivation reduced Kp-2H7 load in monocolonized mice (Extended Data Fig. 9c) and also enhanced F18-mix-mediated *Klebsiella* decolonization (Fig. 3h). Conversely, dietary supplementation with excess gluconate significantly diminished the *Klebsiella*-suppressive effect of F18-mix (Fig. 3h). After sequentially inoculating Kp-2H7-monocolonized mice with each of the F18 strains at 5-day intervals over a 95-day period, there was a corresponding cumulative decrease in faecal gluconate levels that paralleled the reduced Kp-2H7 load (Extended Data Fig. 9d,e). Together, these results suggest that the F18 strains function cooperatively to suppress *Klebsiella* by competitively reducing gluconate availability.

Of the F18 strains, eight effectively consumed gluconate in vitro (Fig. 4a). A mixture of these eight gluconate-utilizing strains (F8-mix)

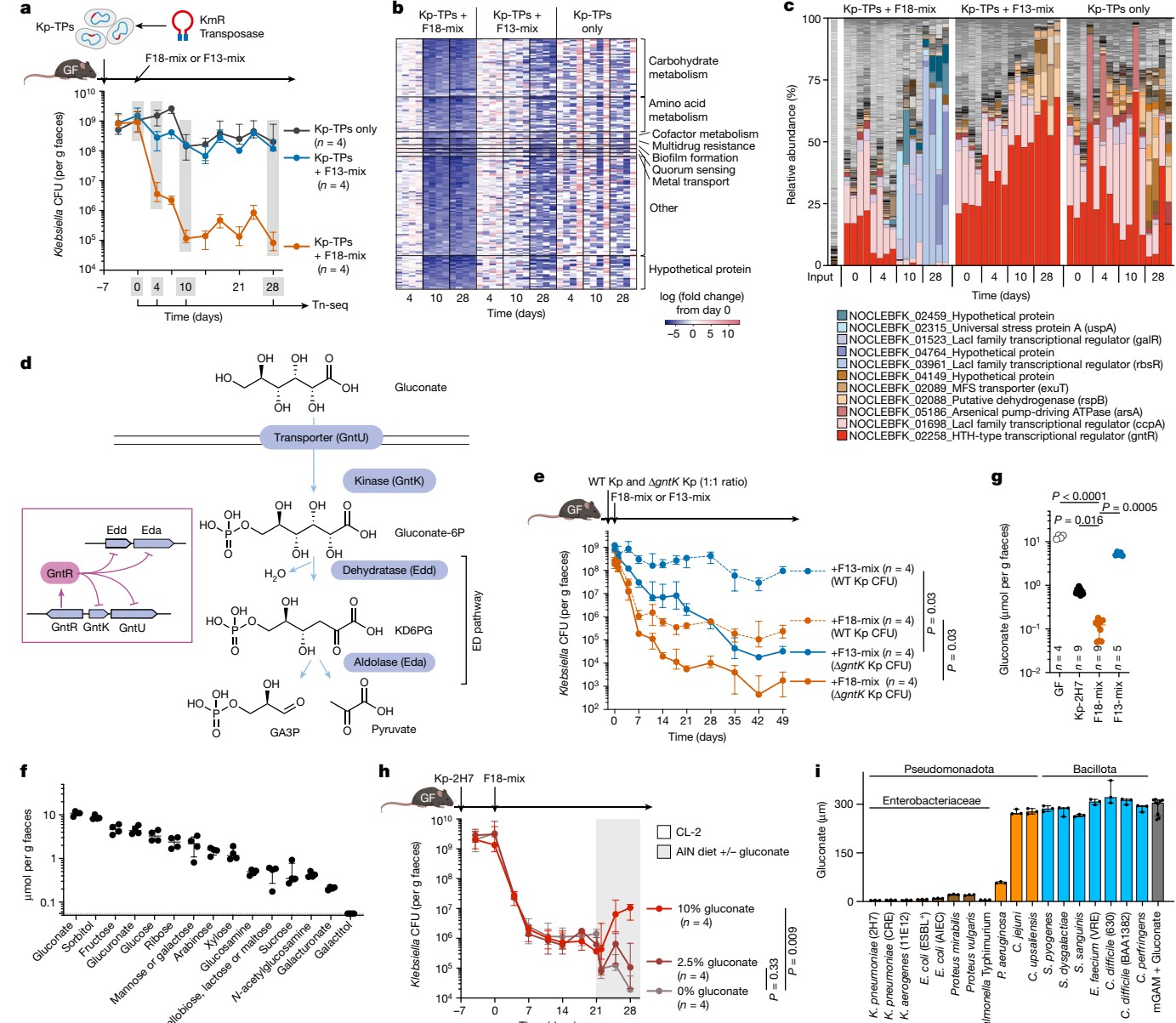

**Fig. 3 | F18-mix competes with Kp-2H7 for gluconate. a**, Random insertion mutagenesis using the Tn5-based transposon yielded $8 \times 10^5$ kanamycin-resistant (KmR) Kp-2H7 mutants (Kp-TPs). Kp-TPs were pooled and administered to GF B6 mice, followed by oral administration of F18-mix or F13-mix. Faecal samples were collected and sequenced on days 0, 4, 10 and 28 to determine CFUs. **b**, Heat map shows 194 Kp-2H7 genes that were significantly down-regulated by F18-mix administration. **c**, Relative abundance of Kp-TP mutants in each mouse (four mice per group). Mutants representing more than 15% of the total reads in any sample are noted in the legend. **d**, Gluconate metabolic pathway in *K. pneumoniae*. GntR suppresses expression of genes encoding gluconate transporter (*gntU*), gluconate kinase (*gntK*) and Entner–Doudoroff (ED) pathway enzymes (*edd* and *eda*). **e**, GF mice were colonized with a 1:1 mixture of wild-type (WT) and Δ*gntK* Kp-2H7, followed by oral administration of F18-mix or F13-mix. Faecal Kp-2H7 CFUs are shown, representative of two independent experiments. **f**, LC–MS/MS analysis of the indicated carbon source in faeces of GF mice (*n* = 4) fed a nutrient-rich (CL-2) diet. **g**, Faecal gluconate levels in GF mice or GF mice colonized with Kp-2H7, F18-mix or F13-mix. **h**, GF mice were colonized with Kp-2H7, followed by oral administration of F18-mix. On day 21, the diet was switched from CL-2 to a gluconate-deficient (AIN93G) diet supplemented with 0%, 2.5% or 10% gluconate. Faecal Kp-2H7 CFUs are shown, representative of two independent experiments. **i**, Pathogenic strains were incubated with 300 μM gluconate for 48 h (*n* = 3 biological replicates). Gluconate concentration in the culture supernatant was measured by LC–MS/MS. Data in **a**,**e**–**i**, are median ± IQR, and are compared by Kruskal–Wallis test using the Benjamini–Hochberg correction for multiple comparisons (**g**,**h**) or by two-sided Mann–Whitney U test (**e**).

substantially reduced Kp-2H7 load in a gnotobiotic setting, though to a slightly lesser extent than did the full F18-mix (Fig. 4b), further supporting the gluconate competition hypothesis. Notably, preferential gluconate utilization is a relatively specific feature of Enterobacteriaceae such as *Klebsiella* and *Escherichia*, whereas strains from other groups including *Pseudomonas*, *Campylobacter*, *Streptococcus*, *E. faecium* and *C. difficile* did not consume gluconate efficiently (Fig. 3i).

This might explain why F18-mix treatment was selectively effective against Enterobacteriaceae over other pathobionts (Fig. 2a).

Notably, LC–MS analysis revealed that F18-mix effectively utilized a range of carbohydrates favoured by Kp-2H7 in addition to gluconate (Extended Data Fig. 9b), suggesting that F18-mix alters intestinal ecological niches by reducing the availability of multiple carbohydrates. Moreover, although Kp-2H7 distribution correlated with gluconate

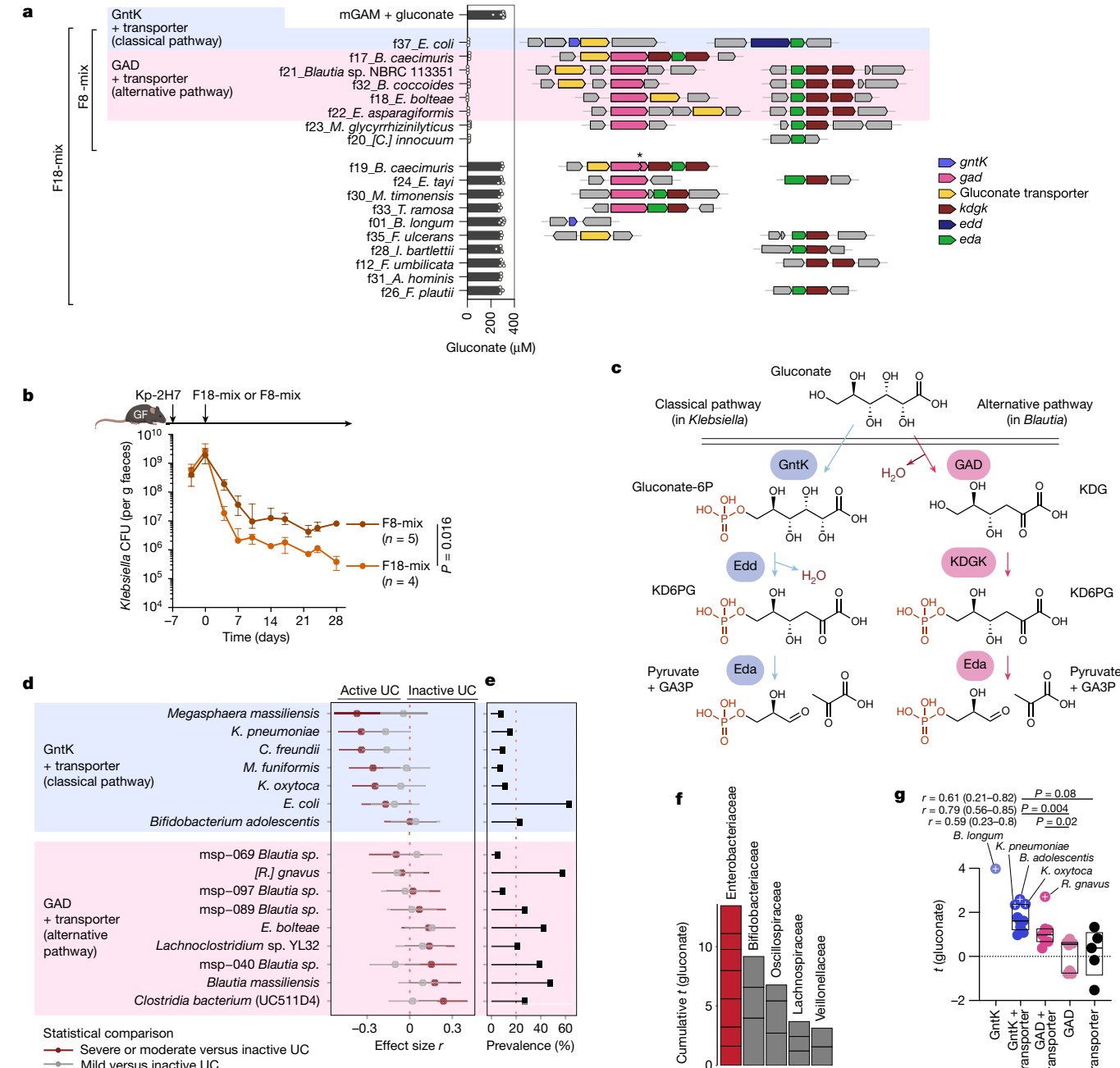

**Fig. 4 | Association between strains carrying gluconate pathway genes and IBD. a**, Left, in vitro gluconate consumption capacity of each of the F18 strains (*n* = 3 biological replicates; median ± IQR). Right, genome neighbourhood of putative gluconate metabolism genes identified in the F18 strains. Asterisk indicates a non-functional frameshift mutation. **b**, GF B6 mice were monocolonized with Kp-2H7 and treated with F8-mix or F18-mix. Faecal Kp-2H7 CFUs are shown as median ± IQR; representative of two independent experiments. The day 28 data were compared by two-sided Mann–Whitney U test. **c**, Classical and alternative gluconate metabolism pathways typically found in *Klebsiella* and *Blautia* species. **d**, Iterative comparative species abundance analysis between paediatric ulcerative colitis (UC) samples with moderate or severe (*n* = 57) or mild (*n* = 64) versus inactive (*n* = 119) disease. Dots and line segments represent *r* effect sizes and confidence intervals

obtained by bootstrapping. Species were grouped on the basis of gluconate-related gene combinations in MSP bins. **e**, MSP prevalence across the cohort (*n* = 240). Taxa without species annotation and reference genome remain classed as MSP. **f,g**, A mixed-effects model quantified the relationship between species abundance and gluconate, controlling for calprotectin and subject in PROTECT (*n* = 84). **f**, Cumulative *t*-values (coefficients adjusted for standard error) demonstrate predominantly positive associations between gluconate abundance and Enterobacteriaceae. **g**, Circles indicate MSPs with gluconate genes. Plus signs represent associations between species and gluconate with Benjamini–Hochberg adjusted *P* values < 0.05. The effect size *r* was computed with confidence intervals from bootstrapping. In box plots, the centre line is the median, the box delineates the IQR and whiskers extend to 1.5× IQR.

concentration in the colon, gluconate levels remained high in the lower small intestine despite F18-mix-mediated Kp-2H7 suppression (Extended Data Fig. 9f). These results indicate an additional layer of complexity and suggest that gluconate availability may not be the sole determinant of *Klebsiella* colonization. Indeed, previous studies

implicate several other carbohydrates, including galactitol, cellobiose (a β-glucosidic sugar) and glucarate in controlling intestinal Enterobacteriaceae growth[33–37]. To deconvolute the roles of these sugars in F18-mix-mediated *Klebsiella* suppression, mice colonized with Kp-2H7 plus F18-mix were fed an AIN93G formula diet supplemented with

individual carbohydrates. Supplementation with mannose, xylose, cellobiose, glucarate or galacturonate did not affect Kp-2H7 sensitivity to F18-mix. By contrast, supplementation with glucosamine, galactitol or sorbitol significantly increased Kp-2H7 levels (Extended Data Fig. 10a), which aligns with prior findings[34–36,45] and suggests potential involvement of these carbohydrates. However, faecal levels of glucosamine and galactitol were very low or below the detection limit (Extended Data Figs. 9a,b and 10b), making it unlikely that competition for these carbohydrates has a major role in the F18-mix-mediated *Klebsiella* suppression. Sorbitol competition is also unlikely to be a primary mechanistic driver, as F18-mix decolonized Kp-2H7 despite having less-efficient sorbitol metabolism (Extended Data Fig. 9b). Additionally, sorbitol was effectively consumed by I41-mix (Extended Data Fig. 9b), which did not suppress Kp-2H7 (Fig. 1b). In sum, although competition for other carbohydrates may contribute to *Klebsiella* suppression, it is most likely that Kp-2H7 relies predominantly on gluconate and that F18-mix-mediated competition for this crucial resource inhibits growth in the gut. The importance of intestinal gluconate may extend beyond merely serving as a nutrient, and its presence and metabolism may have cascading effects on microbial interaction patterns (Supplementary Discussion 4). Nevertheless, these results indicate that the decolonization capacity of F18-mix is context-dependent and can be significantly influenced by dietary components, which must be kept in mind when assessing its clinical efficacy.

## Gluconate pathway genes in patients with IBD

Next, we sequenced the genomes of the F18 strains (Supplementary Table 1) and did not identify any prominent virulence factors or toxins (Supplementary Table 4). Although tetracycline-resistance genes were present in most of the genomes, none of the strains were multidrug-resistant (Supplementary Table 5). Notably, three *Blautia* strains, two *Enterocloster* strains and one *E. coli* strain were found to carry gene clusters encoding enzymes and transporters putatively involved in gluconate metabolism (Fig. 4a and Extended Data Fig. 11). In contrast to the 'classical' gluconate kinase-dependent metabolic pathway genes found in Enterobacteriaceae, the gene clusters identified in the *Blautia* and *Enterocloster* strains encode an alternative pathway that utilizes gluconate dehydratase (GAD). In this pathway, gluconate is first dehydrated to 2-keto-3-deoxygluconate (KDG) by GAD, then phosphorylated into KD6PG by KDG kinase (KDGK), and eventually cleaved into pyruvate and GA3P by Eda[46] (Fig. 4c). We thus queried the presence of alternative gluconate pathway genes in the genomes of our culture collections derived from donors F, K and I (comprising 101 isolates) (Supplementary Table 1). Classical gluconate operon genes were identified in several Enterobacteriaceae, *Bifidobacterium* and *Megasphaera* species, whereas alternative gene clusters encoding both gluconate transporter and GAD homologues were identified in *Blautia*, *Ruminococcus*, *Enterocloster* and *Faecalibacterium* species (Extended Data Fig. 11 and Supplementary Table 6). The carriage of gluconate pathway gene clusters, but not the transporter or dehydratase/kinase alone, was associated with effective gluconate consumption in vitro (Extended Data Fig. 11).

Finally, we examined the association between faecal gluconate levels and the abundance and prevalence of gluconate operon-carrying species in paediatric patients with ulcerative colitis from the Predicting Response to Standardized Colitis Therapy (PROTECT) cohort[47,48] (Extended Data Fig. 12a). Intensity of faecal gluconate, as measured and annotated by LC–MS using a chemical reference, was positively associated with levels of faecal calprotectin in PROTECT (Extended Data Fig. 12b). Among metagenomic species pangenomes (MSPs) containing gluconate-related genes, those annotated as *Citrobacter freundii*, *Klebsiella oxytoca*, *K. pneumoniae*, *E. coli*, *Megasphaera massiliensis* and *Megamonas funiformis*, which carry gluconate kinase and transporter genes, were significantly more prevalent and abundant in patients

with ulcerative colitis in active versus inactive disease states. By contrast, *Blautia*, *Clostridium* and *Faecalibacterium*, which encode the GAD operon, were more abundant in patients in inactive disease states (Fig. 4d,e). Given that expansion of Enterobacteriaceae in the setting of IBD has been associated with inflammation and biochemical processes beyond gluconate metabolism (such as nitrate respiration[1,8]), we used a mixed-effects model to explore the relationship between MSPs and gluconate abundance by implementing faecal calprotectin as fixed effect. This analysis revealed a positive association between gluconate abundance and MSPs that encode gluconate kinase, especially among Enterobacteriaceae (Fig. 4f,g, Supplementary Table 7 and Supplementary Discussion 3). In general, MSPs encoding both the gluconate kinase and transporter together had stronger associations with gluconate level (higher *t*-values and lower error rates) compared with species with other gene combinations. By contrast, MSPs encoding the GAD operon generally showed a weaker association with gluconate abundance, with the exception of *Ruminococcus gnavus*, which is often associated with IBD[49] and had a higher *t*-value than other dehydratase-encoding MSPs (Fig. 4g).

We also examined the adult IBD cohort from the Integrative Human Microbiome Project[14] (HMP2) (Extended Data Fig. 12a). Once again, IBD was associated with an expansion of gluconate kinase gene-carrying Enterobacteriaceae species (Extended Data Fig. 12c). In particular, *E. coli*, *C. freundii* and *K. pneumoniae* consistently emerged as significantly more prevalent in individuals with disease within the HMP2 cohort (Extended Data Fig. 12c), mirroring trends seen in PROTECT. Furthermore, even when faecal calprotectin concentration was controlled for in a mixed-effects model, gluconate level was still significantly associated with abundance of MSPs carrying gluconate kinase operon genes, including Enterobacteriaceae (Extended Data Fig. 12e,f and Supplementary Table 7). Conversely, the enrichment of GAD operons in commensal bacteria was associated with individuals without IBD (Extended Data Fig. 12c), suggesting that these genes may facilitate metabolic competition and suppress proinflammatory pathobionts, thereby maintaining gastrointestinal homeostasis.

## Discussion

Here we adapted a top-down gnotobiotic approach[50] to elaborate a defined microbial consortium consisting of 18 effector bacterial strains from a healthy individual, which is capable of effectively and selectively decolonizing Enterobacteriaceae strains. This F18-mix exerts potent anti-Enterobacteriaceae effects, probably through multiple mechanisms, although primarily by restricting nutrient availability and reshaping ecological niches within the intestine. Each microbiota member possesses a unique nutritional programme, which in turn determines local nutrient availability and thus niche definition. Our results, together with previous studies[32–39,51,52], suggest that Enterobacteriaceae has a hierarchy of carbon preferences, with gluconate being one of the most-preferred carbon sources for growth in the intestine. When faced with F18-mix-mediated gluconate restriction, *Klebsiella* compensates by metabolically switching to utilize other unpreferred carbon sources. However, it is likely that F18-mix can also effectively consume several of these alternative carbon sources, thereby further restricting nutrient availability to *Klebsiella*. Although more research is needed to fully untangle the rules governing effective competition—including other nutritional dependencies, the role of interspecies interactions, the regulation of carbohydrate metabolism by commensal bacteria in various sections of the gut and the influences of dietary components—our findings provide a solid foundation for developing microbiota-directed therapies aimed at suppressing Enterobacteriaceae pathobionts through ecological control. Overall, the F18 strains represent a promising candidate for clinical development as a live biotherapeutic product to treat prevalent infectious and inflammatory diseases.

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

## Methods

### Mice

C57BL/6 mice, maintained under GF conditions, were purchased from Sankyo Laboratories Japan, SLC Japan or CLEA Japan. GF and gnotobiotic mice were bred and maintained within the gnotobiotic facility of Keio University School of Medicine or the JSR-Keio University Medical and Chemical Innovation Center. $Il10^{-/-}$ and $Ifngr1^{-/-}$ mice were purchased from Jackson Laboratories. $Myd88^{-/-}Ticam1^{-/-}$ and $Rag2^{-/-}Il2rg^{-/-}$ mice were purchased from Oriental Bio Service. All mice were maintained under a 12-h light–dark cycle. A temperature of 20–24 °C and a humidity of 40–60% were used for the housing conditions. All animal experiments were approved by the Keio University Institutional Animal Care and Use Committee.

### Human faecal samples and isolation of bacterial strains

Human faecal samples were obtained from healthy human donors, patients with ulcerative colitis and patients with Crohn's disease following the protocol approved by the Institutional Review Board of Keio University School of Medicine (approval numbers 20150075, 20140211, and 20150075). Informed consent was obtained from each individual. Faecal samples were mixed with PBS (containing 20% glycerol) and stored at –80 °C. An aliquot of each sample was diluted with PBS in an anaerobic chamber (80% $N_2$, 10% $H_2$ and 10% $CO_2$; Coy Laboratory Products) and plated onto different agar plates (EG, mGAM, BHK, CM0151, MRS or BL media). After incubating for 2–7 days, colonies with different appearances were transferred to liquid media (EG, mGAM, HK or CM0149), incubated for 24–48 h, mixed with glycerol (final concentration 20% (v/v)), and stored at –80 °C. Bacterial genomic DNA was extracted from the isolated strains using the same protocol as DNA isolation from faecal samples. The 16S rRNA gene locus was amplified by PCR using the KOD plus Neo kit (TOYOBO) according to the manufacturer's protocol. DNA sequencing was performed by Eurofins. Sequences were aligned using the BLAST program of NCBI and the Ribosomal Database Project (RDP) databases. Primers used for DNA sequencing were as follows: F27 primer: 5′-AGRGTTTGATYMTGGCTCAG-3′; R1492 primer: 5′-TACGGYTACCTTGTTACGACTT-3′. Individual isolates in the culture collection were grouped as 'strains' if their 16S rRNA gene sequences shared >98.0% homology.

To prepare the bacterial mixture for inoculation, isolated strains were individually cultured in the appropriate broth at 37 °C for 1–2 days (mGAM broth was used for culturing the F18 strains). Bacterial density was adjusted based on absorbance at 600 nm, and equal volumes of the cultured strains were mixed and centrifuged at 3,000$g$ for 10 min at 4 °C to concentrate fivefold. Thereafter, GF mice were administered 200 µl of the bacterial mixture per mouse (approximately $1–2 × 10^9$ CFU of total bacteria) by oral gavage. The bacterial mixture was administered into GF mice (200 µl per mouse, approximately $1–2 × 10^9$ CFU of total bacteria) by oral gavage. In Extended Data Fig. 3e, f37_E. coli strain was swapped out for the E. coli Nissle1917 strain (Mutaflor, DSM 6601).

### Effect of defined consortia on pathogenic and commensal bacterial strains

To examine the effects of defined consortia on pathogenic bacteria, C57BL/6 GF mice (8–14 weeks of age, housed in separate GF isolators) were inoculated with K. pneumoniae 2H7 (Kp-2H7), carbapenem-resistant K. pneumoniae (CPM$^+$ Kp, ATCC BAA1705), K. aerogenes (strain Ka-11E12[17]), extended-spectrum-β-lactamase producing E. coli (ESBL$^+$ E. coli, ATCC BAA2777), adherent-invasive E. coli (AIEC, strain LF82, provided by N. Barnich[23]), P. aeruginosa (ATCC 10145), vancomycin-resistant E. faecium (VRE Ef, ATCC 700221), C. upsaliensis (ATCC BAA1059), or C. difficile (strain 630, ATCC BAA1382) by oral gavage ($2 × 10^8$ CFU per mouse). Seven days after colonization with pathogenic microbes, the mice were administered 200 µl of isolated bacterial strain mix (total $10^9$ CFU) or 200 µl of human faecal suspension by oral gavage.

Faecal samples were collected from mice every three or four days, suspended in PBS (containing 20% glycerol), and cultured on selective media (DHL agar with 30 mg l$^{-1}$ ampicillin and 30 mg l$^{-1}$ spectinomycin for Kp-2H7, CPM$^+$ Kp, Ka-11E12 and P. aeruginosa, MacConkey agar with 1 mg l$^{-1}$ cefotaxime, and VRE-selective agar plates (BD 251832) for VRE). After 24–48 h of incubation, the CFUs were counted. In cases where evaluation was not possible by counting CFUs, bacterial DNA extracted from faeces was evaluated by quantitative real-time PCR (qPCR). To evaluate colonization resistance activity against Kp-2H7, C57BL/6 GF mice were first colonized with either F18-mix or F13-mix and then inoculated with Kp-2H7 on day 7. Faecal samples were collected every three or four days to count Kp-2H7 CFUs. To examine the cumulative effect of the 18 strains on Kp-2H7, C57BL/6 GF mice were inoculated with Kp-2H7 ($2 × 10^8$ CFU per mouse) by oral gavage, followed by oral administration of each strain of the F18-mix one by one every five days for 95 days. Faecal samples were collected every five days to count the CFUs of Kp-2H7 as well as to quantify the levels of gluconate.

To investigate the influence of F18-mix on commensal strains, 7 strains (Dorea longicatena, Eubacterium rectale, C. scindens, Bacteroides thetaiotaomicron, Bacteroides uniformis, Bifidobacterium adlescentis and Collinsella aerofaciens) were selected from our culture collection. C57BL/6 GF mice were colonized with the 7 commensal strains together with Kp-2H7 (each $2 × 10^8$ CFUs). Fourteen days later, mice were administered with F18-mix (total $10^9$ CFUs in 200 µl) by oral gavage. Faecal samples were collected twice per week and subjected to quantification by CFU calculation and qPCR. To investigate the effect of F18-mix in the context of a more complex microbiota, C57BL/6 GF mice were colonized with either 41 strains from donor I, 46 strains from donor K, or a combination of both groups with C. scindens VE202-26 (totaling 88 strains). Subsequently, without any prior antibiotic treatment, these mice were orally inoculated with F18-mix. Faeces were collected for full-length 16S rRNA gene sequencing analysis.

To examine the effects of F18-mix in the context of dysbiotic microbiota, GF B6 or $Il10^{-/-}$ (B6 background) mice were administrated with 200 µl of faecal suspension from patients with ulcerative colitis or Crohn's disease containing high levels of ESBL$^+$ E. coli or K. pneumoniae. Ten days after IBD microbiota inoculation, mice were treated with 500 µl of 1 g l$^{-1}$ vancomycin by oral gavage. The mice were treated with or without F18-mix four times (at 4, 8, 24 and 48 h following the vancomycin treatment). Faeces were collected and subjected to 16S rRNA gene sequencing or counting CFUs of E. coli or K. pneumoniae.

### Dietary supplementation with carbohydrates

Unless otherwise indicated, mice were fed a nutrient-rich diet (CL-2; CLEA Japan), which is high in gluconate. To assess the impact of dietary carbohydrate supplementation, a formula diet (AIN93G; Oriental Yeast Co.) was used, which is high in glucose but lacks gluconate, sorbitol, mannose, xylose, glucarate, galacturonate and xylose. This diet was supplemented with varying percentages of gluconate (0%, 2.5% or 10% of total calories) or with 10% of other carbohydrates, including glucarate, galactitol, sorbitol, cellobiose, glucosamine, xylose, mannose or galacturonate (Supplementary Table 8). The diets were sterilized by γ-irradiation (50 Gy).

### Bacterial DNA extraction and quantitative real-time PCR

The frozen faecal samples were thawed and 50 µl of each sample was mixed with 350 µl TE10 (10 mM Tris-HCl, 10 mM EDTA) buffer containing RNase A (final concentration 100 µg ml$^{-1}$, Invitrogen) and lysozyme (final concentration 3.0 mg ml$^{-1}$, Sigma). The suspension was incubated for 1 h at 37 °C with gentle mixing. Purified achromopeptidase (Wako) was added to a final concentration of 2,000 units ml$^{-1}$, and the sample was further incubated for 30 min at 37 °C. Then, sodium dodecyl sulfate (final concentration 1%) and proteinase K (final concentration 1 mg ml$^{-1}$, Nacalai) were added to the suspension and the mixture was incubated for 1 h at 55 °C. Thereafter, purified DNA was obtained from

the samples using the Maxwell RSC cultured cell DNA kit, according to the manufacturer's protocol. For quantifying the amount of bacterial DNA, real-time qPCR was performed using the Thunderbird SYBR qPCR Mix (TOYOBO) and LightCycler 480 (Roche). The primer pairs used in this study are listed in Supplementary Table 9.

### Full-length 16S rRNA gene amplicon sequencing

Full-length 16S rRNA amplicon sequencing was performed according to the protocol prepared by PacBio with slight modifications. In brief, the full-length 16S rRNA gene (including hypervariable regions V1 to V9) was amplified using barcoded 27Fmod (5′-Phos-GCATCNNNNNNNNNNNAGRGTTYGATYMTGGCTCAG-3′) and barcoded 1492R (5′-Phos-GCATCNNNNNNNNNNNRGYTACCTTGTTA CGACTT-3′) primers; 'Phos' indicates a 5′-phosphate modification, and 'N' represents a unique PacBio barcode sequence for each sample. The PCR conditions were as follows: an initial denaturation at 95 °C for 3 min, followed by 20 cycles of denaturation at 95 °C for 30 s, annealing at 57 °C for 30 s, and extension at 72 °C for 60 s. The PCR products were purified using AMPure magnetic beads and pooled at equimolar concentrations. The pooled amplicons were further purified with AMPure beads, and 1 µg was used for library preparation. The library was prepared according to the PacBio SMRTbell Prep Kit 3.0 protocol and sequenced on the PacBio Revio system. The HiFi reads were automatically generated using SMRT Link software (version 13.0) with default settings and demultiplexed using lima application in SMRT Tools with HIFI-ASYMMETRIC presets.

### Amplicon sequence variants analysis

Full-length 16S rRNA gene amplicon sequence variants (FL16s-ASVs) were inferred from demultiplexed HiFi reads using the DADA2 package (version 1.30.0) in R (version 4.3.3) according to the previously described DADA2 for PacBio workflow[53] with slight modifications. The reads were subjected to quality filtering and trimming using the filterAndTrim function with the following parameters: minQ=3, minLen=1300, maxLen=1600, maxN=0, rm.phix=FALSE, maxEE=2. To learn error rates using the learnErrors function without the 'dada2:::PacBioErrfun' option, due to the binned quality value adopted by the Revio system, with a maximum of QV40. FL16s-ASVs were then subjected to a homology search against 16S rRNA gene sequences extracted from publicly available genomes (downloaded from GenBank on 12 September 2023) using BLASTN with a maximum e-value cut-off of $1 \times 10^{-10}$. Top hits were determined by the highest bitscore.

### Bacterial whole-genome sequencing

The Illumina MiSeq and PacBio Sequel platforms were used for bacterial whole-genome sequencing. For Illumina sequencing, the library was prepared using the TruSeq DNA PCR-free library prep kit (Illumina), with a target insert size of 550 bp. All the Illumina reads were trimmed and filtered using the FASTX-toolkit (version 0.0.13). For the PacBio sequencing, the library was prepared using the SMRTbell template prep kit 1.0. Sequence data for both types of sequencing were assembled using the hybrid assembler Unicycler. Taxonomic assignment of the genomes was determined by classify_wf of GTDB-tk[54] version 2.3.0 with GTDB[55] database R214. NCBI taxonomy of fastANI reference genome related to the genome of each strain was retrieved using NCBI-genome-download version 0.3.3 (ncbi-genome-download; https://doi.org/10.5281/zenodo.8192432) and rankedlineage.dmp from NCBI taxonomy database[56] (downloaded on 14 September 2023). The genes were predicted using Prokka version 1.14.0 with "--kingdom Bacteria --rnammer" options, and rnammer version 1.2. The homology search for the predicted genes was performed using diamond[57] version 2.0.15 with "blastp --evalue 0.00001 --id 30 --query-cover 60 --ultra-sensitive" options, with KEGG (downloaded on 19 April 2022)[58], COG (downloaded on 19 May 2021)[59], VFDB (downloaded on 10 September 2022)[60], and UniRef90 (downloaded on 24 May 2022;

https://www.uniprot.org/help/uniref) databases. For homology search against KEGG DB, a database was manually constructed from protein sequences with KEGG Ontology (K number) which were extracted from KEGG non-redundant datasets at the species level. We also added homology search for gluconate metabolism genes in our isolated strains with "blastp --evalue 0.00001 --id 20 --query-cover 60 --ultra-sensitive" options. The sequences of gluconate kinase (*gntK*, MKMCEHOJ_02531) and gluconate transporters (MKMCEHOJ_02530 and MKMCEHOJ_02505) from f37 *E. coli* strain, and gluconate dehydratase (*gad*, EAOGLLOI_00767), gluconate transporters (EAOGLLOI_00766 and EAOGLLOI_00912), 2-dehydro-3-deoxygluconokinase (*kdgK*, EAOGLLOI_00768), and 2-dehydro-3-deoxyphosphogluconate aldolase (*eda*, EAOGLLOI_00769) from f17 *Blautia caecimuris* strain were used as reference sequences.

### Ex vivo caecal suspension culture

Caecal contents from GF or F31-mix, F18-mix and F13-mix colonized mice were anaerobically resuspended in water at a concentration of 100 mg ml$^{-1}$. Caecal contents were either filtered through a 0.22-µm filter (Millex Millipore) after centrifuging at 10,000$g$ for 5 min, heat-killed at 105 °C for 30 min, or left untreated. Thereafter, a diluted overnight culture of Kp-2H7 ($10^3$ CFU in 10 µl) was added to 200 µl of each caecal suspension. After incubating at 37 °C for 48 h under aerobic or anaerobic conditions, samples were serially diluted and plated on a selection agar plate (DHL with 30 mg l$^{-1}$ ampicillin and 30 mg l$^{-1}$ spectinomycin) for counting Kp-2H7 CFU.

### Bacterial growth monitoring

The wild-type, Δ*gntK* or Δ*gntR* Kp-2H7 strain was cultured in M9 minimal medium for 24 h at 37 °C. Afterward, the culture was diluted 100-fold with sterile water. A 10-µl aliquot of the diluted culture was inoculated into 200 µl M9 medium supplemented with individual carbohydrates (final concentration of 2 mM) as the sole carbon source, or with a mock control. To examine the effect of metabolites on Kp-2H7 growth, 10 µl of wild-type Kp-2H7 culture dilutions were inoculated into 200 µl M9 medium containing varying concentrations of 4-HBA (100, 10, 1 or 0.1 mM), cholic acid (500, 100, 20 or 4 µM), and acetate or butyrate (100, 25, 6.25, 1.56 or 0.39 mM). The pH of acetate and butyrate was adjusted to either 5.0 or 7.0. Bacterial growth was monitored by measuring absorbance at 600 nm every 30 min using a microplate reader (Sunrise Thermo (Tecan) for anaerobic conditions and Infinite 200 PRO (Tecan) for aerobic conditions) at 37 °C with 100 s shaking before each time point.

### Transcriptome analysis of epithelial cells

Total RNA was isolated from colonic epithelial cells using NucleoSpin RNA (Macherey-Nagel), according to the manufacturer's instructions. Libraries for RNA sequencing were prepared using TruSeq Stranded mRNA Library Prep (Illumina), according to the manufacturer's instructions. The libraries were sequenced using NovaSeq 6000 (Illumina) with the mode of 150-bp paired-end. The sequenced paired-end reads were quality-controlled using Trimmomatic[61] version 0.39 with "2:30:10 LEADING:3 TRAILING:20 SLIDINGWINDOW:4:15 MINLEN:5" options and FASTX-Toolkit version 0.0.13 (https://github.com/agordon/fastx_toolkit) with "-q 20 -p 80" options. Unpaired reads and reads mapped to the PhiX reference genome using minimap2[62] version 2.17-r941 were excluded from further analyses. The remaining quality-controlled reads were mapped to the mouse reference genome (mm10) using STAR[63] version 2.7.2b. The mapped reads were counted for each gene using featureCounts[64] version 1.5.2 with "-t exon -p -B -Q 1" options. the transcripts per million (TPM) values of each gene in each sample were calculated. The differential expression analysis was performed using DESeq2[65] version 1.28.1, and the *P* values were corrected by the Benjamini−Hochberg method to maintain the false discovery rate (FDR) below 5%.

## Transcriptome analysis of Kp-2H7

To examine the transcriptome landscape of Kp-2H7 in vivo, GF mice were monocolonized with Kp-2H7, followed by oral administration of F18-mix or vehicle control. Two days after F18-mix administration, faecal samples were collected, and total RNA was extracted. For the in vitro examination of the transcriptomes of Kp-2H7 strains, wild-type, Δ*gntK*, and Δ*gntR* Kp-2H7 were cultured at 37 °C in M9 minimal medium supplemented with either glucose or gluconate. Bacteria were collected during the early log phase (absorbance at 600 nm = 0.35), and total RNA was extracted. Isolation of total RNA from in vivo faecal samples or in vitro culture samples was conducted using the NucleoSpin RNA kit (Macherey-Nagel), according to the manufacturer's instructions. Libraries for RNA sequencing were prepared using TruSeq Stranded mRNA Library Prep (Illumina) and sequenced using HiSeq X (Illumina) with the mode of 150-bp paired-end. To analyse the in vivo transcriptome profiles of Kp-2H7 in the presence or absence of F18-mix, a reference genome was created by concatenating the genome sequence of Kp-2H7 with the genome sequences of the F18-mix. The sequenced paired-end reads were quality-controlled using Trimmomatic[61] version 0.39 with "2:30:10 LEADING:3 TRAILING:20 SLIDINGWINDOW:4:15 MINLEN:5" options and FASTX-Toolkit version 0.0.13. Unpaired reads and reads mapped to the mouse (mm10) or PhiX reference genome using minimap2[62] version 2.17-r941 (in vivo) or 2.24-r1122 (in vitro) were excluded from further analyses. The quality-controlled reads were mapped to the concatenated or Kp-2H7 reference genome using bowtie2[66] version 2.3.4.1. (in vivo) or 2.4.4 (in vitro). For in vivo mice faecal samples, the read counts for each Kp-2H7 gene were obtained by counting uniquely mapped reads and then distributing and summing multi-hit read counts based on the number of uniquely mapped reads. For in vitro culture samples, the read counts for each Kp-2H7 gene were obtained using featureCounts[64] version 2.0.1 with "-t CDS -p -B -Q 10" options. The differential expression analysis was performed using DESeq2[65] version 1.28.1 (in vivo) or 1.30.1 (in vitro) with "fit-Type = local" option and Benjamini–Hochberg correction method to maintain the FDR below 5%. The heat map was obtained from the variance-stabilizing transformations values obtained from the DESeq2 output.

For real-time qPCR analysis, cDNA was synthesized using ReverTra Ace qPCR RT Master Mix (TOYOBO), and qPCR was performed using Thunderbird SYBR qPCR Mix (TOYOBO) on a LightCycler 480 (Roche).

## Construction of transposon mutant library

A transposon insertion library of Kp-2H7 was constructed using the EZ-Tn5TM<KAN-2> Tnp Transposome kit (Lucigen). In brief, 80 μl ($10^9$ CFU) of Kp-2H7 suspension was mixed with 0.5 μl of EZ-Tn5TM <KAN-2>, transferred to a 1-mm gap width electroporation cuvette, and subjected to electroporation using ELEPO21 (Nepa Gene Co.) with the following parameters: poring pulse; voltage: 1,800 V, pulse length: 5.0 ms, pulse interval: 50 ms, number of pulses: 1, and polarity: +, and transfer pulse; voltage: 150 V, pulse length: 50 ms, pulse interval: 50 ms, number of pulses: 5, and polarity: ±. Transformed Kp-2H7 cells were incubated in 1 ml LB broth for 3 h at 37 °C, and then selected on LB agar plates containing kanamycin (90 mg l⁻¹) at 37 °C. Thereafter, approximately $8 × 10^5$ transposon mutant colonies were collected and stored at –80 °C in LB containing 20% glycerol.

## Transposon sequencing

GF mice were colonized with the pool of $8 × 10^5$ Kp-2H7 transposon mutants. Faecal samples were collected on days 0, 4, 10 and 28 following colonization, suspended in PBS (50 mg ml⁻¹) containing 20% glycerol, and cultured overnight at 37 °C on LB agar plates containing kanamycin (90 mg l⁻¹). Kp-2H7 mutant colonies were scraped together and DNA was extracted by the method described above. Transposon sequencing was carried out according to the method described by

Kazi et al.[67]. In brief, genomic DNA was fragmented via sonication. Then, a poly-C tail was added to the 3′ end of the DNA fragment by terminal deoxynucleotidyl transferase. The transposon junctions were amplified using a biotinylated primer, which was then enriched using streptavidin beads. By performing a second nested PCR, a single barcode was added to each sample. The libraries were sequenced using HiSeq 2500 (Illumina) with the mode of 50-bp single-end. The first 24 bases of each sequenced read were trimmed to exclude primer and mosaic end sequences. The trimmed reads were quality-controlled using Trimmomatic[61] version 0.39 and FASTX-Toolkit version 0.0.13. The remaining reads were mapped to the PhiX reference genome (mm10) using minimap2 version 2.17-r941 to exclude those that align with the PhiX genome. Then, the analysis-ready reads were mapped to the Kp-2H7 genome using bowtie2 version 2.4.2[66]. The mapped reads were counted for each gene using featureCounts[64] version 1.5.2 with "-t CDS -p -B -Q 1" options, and the TPM of each gene was calculated as the relative abundance of a gene mutant in a sample by assuming that each transposon mutant has a single insertion. The differential abundance mutants were detected by Welch's *t*-test for log-scaled TPM with Benjamini–Hochberg correction method to maintain the FDR below 5%.

## Generation of Kp-2H7 mutants

The Kp-2H7 deletion mutants were generated as shown in Supplementary Fig. 3 using the Quick and Easy *E. coli* Gene Deletion Kit (Gene Bridges) according to the manufacturer's protocol. In brief, Kp-2H7 cells were transformed with the pRED/ET plasmid harbouring the tetracycline-resistant gene by electroporation. Bacteria with pRED/ET were selected on LB plates containing tetracycline (30 mg l⁻¹) at 30 °C. Thereafter, these cells were incubated in LB broth with appropriate antibiotics at 30 °C until absorbance at 600 nm reached 0.2, followed by an additional 1 h of incubation with 0.3% L-arabinose at 37 °C to induce the expression of the recombinant proteins. These cells were used to prepare electrocompetent cells and were transformed with the linear DNA fragment (the FRT-PGK-gb2-neo-FRT cassette)-flanked homology arms. The functional cassettes were generated by PCR, according to the manufacturer's protocol. The primers with homology arms are listed in Supplementary Table 9. The electroporated cells were incubated in 1 ml LB broth for 3 h at 37 °C. Gene deletion strains were selected on LB agar plates with kanamycin (90 mg l⁻¹) after overnight growth at 37 °C. The double-knockout strains were generated by removing the kanamycin selection marker through electroporation of the FLP expression plasmid (707-FLPe) and repeating the above-mentioned protocol. The deletions were confirmed by DNA sequencing.

## Isolation of lymphocytes and flow cytometry

Lymphocytes were collected from the large intestines and analysed according to previously described protocols[17,68]. In brief, the intestines were dissected longitudinally and washed with PBS to remove all luminal contents. All samples were incubated in 15 ml Hanks' balanced salt solution (HBSS) containing 5 mM EDTA for 20 min at 37 °C in a shaking water bath to remove epithelial cells. Thereafter, after removal of any remaining epithelial cells, muscular layers and fat tissues using forceps, the samples were cut into small pieces and incubated in 10 ml RPMI1640 containing 4% foetal bovine serum (FBS), 0.5 mg ml⁻¹ collagenase D (Roche Diagnostics), 0.5 mg ml⁻¹ dispase II (Roche Diagnostics), and 40 μg ml⁻¹ DNase I (Roche Diagnostics) for 50 min at 37 °C in a shaking water bath. Thereafter, the resultant digested tissues were washed with 10 ml HBSS containing 5 mM EDTA, resuspended in 5 ml of 40% Percoll (GE Healthcare), and underlaid with 2.5 ml of 80% Percoll in a 15-ml Falcon tube. Percoll gradient separation was performed by centrifugation at 850*g* for 25 min at 25 °C. Lymphocytes were collected from the interface of the Percoll gradient and washed with RPMI1640 containing 10% FBS, and then stimulated with 50 ng ml⁻¹ PMA and 750 ng ml⁻¹

ionomycin (both from Sigma) in the presence of Golgistop (BD Biosciences) at 37 °C for 4 h. After labelling of the dead cells with Ghost Dye Red 780 Viability Dye (Cell Signaling Technology), the cells were permeabilized and stained with anti-CD3e (BUV395; BD Biosciences), anti-CD4 (BUV737; BD Biosciences), anti-TCRβ (BV421; Biolegend) and anti-IFNγ (FITC; Biolegend) at 1:1,000 dilution using the Foxp3/Transcription Factor Staining Buffer Kit (Tonbo Biosciences), according to the manufacturer's instructions. All data were collected on a BD LSRFortessa (BD Biosciences) and analysed using Flowjo software (TreeStar). CD4+ T cells were defined as a CD4+TCRβ+CD3e+ subset within the live lymphocyte gate.

## Measurement of lipocalin-2 and calprotectin

The faecal pellets from $Il10^{-/-}$ mice were vortexed, suspended in PBS (5% w/v) with Complete Protease Inhibitor Cocktail (1 tablet dissolved in 50 ml PBS; Roche) and centrifuged, and supernatants were collected. The concentration of lipocalin-2 and calprotectin in faecal supernatants was measured by ELISA (Mouse Lipocalin-2 Matched Antibody Pair Kit; Abcam, Mouse S100A8/S100A9 Heterodimer DuoSet; R&D), according to the manufacturer's protocol.

## Histological analysis

Colon tissue samples were dissected longitudinally and swiss-rolled, fixed with 4% paraformaldehyde, embedded in paraffin, sliced to 5-μm sections and stained with hematoxylin and eosin. The degrees of colitis were graded by the mouse colitis histology index[69]. The histological slides were evaluated blind by two investigators.

## Non-targeted metabolomics analysis

C57BL/6 GF mice were monocolonized with Kp-2H7, followed by oral administration of bacterial mix. Caecal contents were collected on day 28 after administration of isolated bacterial mix and stored at −80 °C until use. Frozen caecal contents were homogenized by shaking with metal corn using a multi beads shocker as previously described[70]. Then, the samples were suspended in 400 μl of methanol per 100 mg caecal contents, and a 40 μl aliquot was subjected to the single layer extraction and untargeted LC–QTOF/MS analysis[70]. SCFAs were simultaneously extracted and derivatized from 20 μl of the suspension by using pentafluorobenzyl bromide alkylation reagent (Thermo Fischer Scientific), and analysed by gas chromatography–mass spectrometry (GC–MS) as previously described[71]. Water-soluble metabolites were extracted by first mixing 4 μl of the suspension, 196 μl of methanol, 200 μl of chloroform, 70 μl of water, and 10 μl of internal standards mix (100 μM cycloleucine, 500 μM citric acid-d4, and 1.0 mM ornithine-d7 (Cambridge Isotope Laboratories)). After vortexing for 1 min and centrifugation at 15,000g for 5 min at 4 °C, 100 μl of supernatant was evaporated to dryness. The dried samples were derivatized via methoxyamination, trimethylsilylation, or *tert*-butyldimethylsilylation, and then analysed by GC–MS/MS using Smart Metabolite Database (Shimadzu) or GC–MS operated in selected ion monitoring mode, as described previously[72]. Bile acids were extracted from 4 μl of the suspension mixed with deuterium-labelled internal standard mix (1.0 μM cholic acid-d4, 1.0 μM lithocholic acid-d4, 1.0 μM deoxycholic acid-d4, 1.0 μM taurocholic acid-d4, and 1.0 μM glycocholic acid-d4 (Cayman Chemical)) using the Monospin C18 column (GL science). The column was washed with 300 μl water (×2) and 300 μl of hexane (×1). Bile acids were eluted with 100 μl methanol, then subjected to LC–MS/MS analysis using an UPLC I class (Waters) with a linear ion-trap quadrupole mass spectrometer (QTRAP 6500; AB SCIEX) equipped with an Acquity UPLC BEH C18 column (50 mm, 2.1 mm, and 1.7 μm; Waters). Samples were analysed with a mobile phase consisting of water:methanol:acetonitrile (14:3:3 (vol:vol:vol)) and acetonitrile, both containing 5 mM ammonium acetate, for 4 min, which was changed to 40:60 after 12 min, to 5:95 after 2 min, and then held for 2 min; with flow rates of 300 μl min⁻¹. Bile acids were detected by multiple-reaction monitoring in negative mode. Ions of [M·H]⁻, taurine ($m/z = 124$), and glycine ($m/z = 74$), generated from the precursor ion, were monitored as product ions for non-conjugated, taurine-conjugated, and glycine-conjugated bile acids, respectively. MS/MS settings were as follows: ion source, turbo spray; curtain gas, 30 psi; collision gas, 9 psi; ion spray voltage, −4,500 V; source temperature, 600 °C; ion source gas 1, 50 psi; and ion source gas 2, 60 psi.

## Measurement of carbohydrate levels

To evaluate bacterial gluconate utilization in vitro, isolated strains were cultured in mGAM broth or RCM containing 300 μM gluconate for 48 h at 37 °C under anaerobic conditions. Supernatant of each culture broth was collected, and the concentration of gluconate was measured by the ExionLC AD and SCIEX Triple Quad 6500+ LC–MS/MS system. To evaluate carbohydrate levels in faeces or intestinal contents, each sample was suspended in water (50 mg ml⁻¹), and the carbohydrate levels in the supernatant were measured by LC–MS/MS. The measurement conditions for gluconate, glucuronate, and galacturonate were as follows: chromatographic separation was performed using the Intrada Organic Acid column, 150 × 2 mm (Imtakt); column temperature was 40 °C; and the volume of each injection was 5 μl. The mobile phase comprising A (acetonitrile/water/formic acid, 10/90/0.1) and B (acetonitrile/100mM ammonium formate, 10/90) was used under gradient conditions: 0–1.5 min, A 100%, B 0%; 1.6–7 min, A 70%, B 30%; 10–13 min, A 0%, B 100%; and 13.1–18 min, A 100%, B 0%); and the flow rate was 0.2 ml min⁻¹. Detailed MS conditions were as follows: curtain gas, 30 psi; collision gas, 6; ion spray voltage, −4,500 V; temperature, 550 °C; ion source gas 1, 50 psi; and ion source gas 2, 60 psi. The retention time and multiple-reaction monitoring transitions are listed in Supplementary Table 10. The measurement conditions for other carbohydrates were as follows: chromatographic separation was performed using the UK-Amino column (UKA26), 250 × 2 mm, (Imtakt); column temperature was 65 °C and the volume of each injection was 2 μl. The mobile phase comprising A (5 mM ammonium acetate, 0.05% formic acid) and B (acetonitrile) was used under gradient conditions: 0–10 min, A 5%, B 95%; 35 min, A 15%, B 85%; 50 min, A 40%, B 60%; 50.1–55 min, A 80%, B 20%; 55.1–60 min, and A 5%, B 95%); and the flow rate was 0.25 ml min⁻¹. Detailed MS conditions were as follows: curtain gas, 25 psi; collision gas, 9; ion spray voltage, −4,500 V in negative mode and 5,500 V in positive mode; temperature, 250 °C, ion source gas 1, 50 psi; and ion source gas 2, 70 psi. Multiple-reaction monitoring parameters are listed in Supplementary Table 10. Data were obtained using Analyst software version 1.7.1 and analysed using SCIEX OS-MQ software version 2.1.0.55343.

## Metagenomic analysis of IBD cohorts

To explore established and novel microbial taxa possessing gluconate operon genes, gene catalogues were acquired from two cohorts with IBD aetiology: the paediatric PROTECT and adult HMP2 cohorts, comprising 240 and 1,638 longitudinal metagenomic samples from 94 and 91 individuals, respectively. MSPs were constructed via the co-abundant gene binning (MSPminer[73]), followed by quality assessment (CheckM[74]), as described by Schirmer et al.[3] (PROTECT) and Kenny et al.[75] (HMP2). A targeted screening of these bins with DIAMOND BLASTP version 0.9.14[76] was conducted to identify putative gluconate transport and metabolism genes, retaining hits with an e-value <0.01 and sequence identity ≥60%. MSPs were categorized based on the combinations of gluconate-related genes detected. A differential abundance analysis was performed on TPM-normalized and centred log-ratio-transformed MSP counts to control for sequencing depth, gene length, and compositional biases. Statistical significance was ascertained through a non-parametric, two-sided Mann–Whitney U test with Benjamini–Hochberg correction. Effect sizes ($r$), calculated as the test statistic divided by the square root of the sample size, along with bootstrapped confidence intervals, were computed to account for unbalanced group

sizes, offering insights into the robustness and directionality of the observed effects.

For PROTECT, comparative analyses were iteratively repeated with varying seed values for random sample selection from longitudinal data pools of mild ($n = 64$), moderate/severe ($n = 57$), and non-IBD samples ($n = 119$). Within HMP2, inclusion was limited to cross-sectional samples accompanied by calprotectin data. In response to the attenuated disease signal observed in the study cohort[77], a targeted inflammation-specific selection approach was utilized. For IBD cases the sample with maximal calprotectin value per patient was included (Crohn's disease, $n = 41$; ulcerative colitis, $n = 26$). Conversely, for the non-IBD control group, the sample with the minimal calprotectin value per patient was chosen ($n = 24$). Statistical analyses were conducted using R software version 4.2.1 (Ubuntu 20.04.5 LTS).

## Untargeted stool metabolomics and gluconate intensity estimation

Untargeted stool metabolomics of faecal samples from the PROTECT cohort was performed using LC–MS in negative mode, and calprotectin was measured by ELISA. In brief, hydrophilic interaction liquid chromatography (HILIC) analyses of water-soluble metabolites in the negative ionization mode were conducted using Shimadzu Nexera X2 U-HPLC (Shimadzu) coupled to a Q Exactive Plus mass spectrometer (Thermo Fisher Scientific). Metabolites were extracted from plasma or stool (30 µl) using 120 µl of 80% methanol containing inosine-15N4, thymine-d4, and glycocholate-d4 internal standards (Cambridge Isotope Laboratories). The samples were centrifuged (10 min, 9,000$g$, 4 °C), and the supernatants were injected directly onto a 150 × 2.0 mm Luna NH2 column (Phenomenex). All masses detected in HILIC negative mode were matched via adduct subtraction and molecular formula match to compounds downloaded from the Human Metabolome Database (HMDB) on 10 October 2022. The measured $m/z$ values were adjusted for [M-H]- adducts, and molecular formulae matching to within 5 ppm were selected as candidate identifiers. In cases where multiple molecular formulae matched the adduct-adjusted mass (as a result of multiple potential adducts), the one with a minimal ppm difference was selected. Out of 4,461 detected features ($m/z$, retention time pairs), a single feature 195.0512 $m/z$ at 4.34 min resolved to the formula $C_6H_{12}O_7$ (delta ppm = 0.89), related to a group of 5 compounds with canonical structure O=C(O)C(O)C(O)C(O)C(O)CO, which includes L-gluconic acid (HMDB0000625). The metabolic feature was subsequently validated with a reference standard (Sigma Aldrich, S2054) via retention time and MS/MS match against iHMP pooled stool samples, and aligned via global $m/z$ and retention time matching with the PROTECT stool samples using Eclipse[78]. This led to annotating HNs_QI1923 (HILIC-neg 195.0512 $m/z$ at 4.34 min) from PROTECT and QI11027 (HILIC-neg 195.0512 $m/z$ at 4.48 min) from HMP2.

## Statistical analyses

Statistical analyses were performed using GraphPad Prism software. Kruskal–Wallis test and the FDR method of Benjamini and Hochberg were used for multiple comparisons during CFU comparisons. Mann–Whitney U test with Welch's correction was used for comparisons between the two groups. Spearman's rank correlation was used to investigate the correlation between the relative abundance of Kp-2H7 and isolated strains.

## Reporting summary

Further information on research design is available in the Nature Portfolio Reporting Summary linked to this article.

## Data availability

The data of Tn-Seq data and RNA sequence data of host, Kp-2H7 in vivo and Kp-2H7 in vitro are deposited in the DNA Data Bank of Japan under BioProject PRJDB17114. Genome sequences of the 31, 41 and 46 strains isolated from donors F, I and K, respectively, are deposited in the DNA Data Bank of Japan under BioProject PRJDB17661. Source data are provided with this paper.

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

**Acknowledgements** K.H. is funded through the Japan Agency for Medical Research and Development (AMED) Moonshot Research and Development Program (JP22zf0127007), AMED NEDDTrim programme (JP21ae0121041), AMED LEAP programme (JP20gm0010003), Grant-in-Aid for Specially Promoted Research from JSPS (20H05627), and a Stand Up to Cancer Grant (SU2C Convergence 3.1416 Research Team). R.J.X. is supported by grants from the Center for Microbiome Informatics and Therapeutics at MIT and NIH (DK043351, AI172147, DK 127171, HL157717). The authors thank S. Narushima for bacterial isolation, K. Takeshita for collecting clinical samples, and M. Kumamoto for animal maintenance. Figs. 1a,b,e,f, 2b–d, 3a,e,h and 4b, Extended Data Figs. 1a, 3a–c,e–g, 4a–c, 5a,b, 6a,d, 7a, 8b, 9c,d, 10a and 12a and Supplementary Fig. 1 were created with BioRender.com.

**Author contributions** K.H. and M.F. planned experiments, analysed data and wrote the paper together with T.K., M.-M.P., K.Y.-M., D.R.P., S.S., Y.A., N.O., Y.K., M.A., A.N.S., D.R.P. and R.J.X. K.Y.-M. designed the in vivo Kp-2H7 decolonization experiment. M.F., T.K. and K.Y.-M. conducted bacterial and animal experiments supported by N.H., T.O., M.Y., T. Tuganbaev, K. Amafuji, K.S., N.N., J.A.-P., K.P., C.B.C., S.C., J.M.N., B.O., T. Tanoue and K. Atarashi. S.S., M.U., N.O. and M.A. conducted metabolomic analyses. M.-M.P., D.R.P., Y.A., Y.K., W.S., M.H., M.S., V.B. and R.J.X. performed microbiome and bioinformatic analysis.

**Competing interests** K.H. is a scientific advisory board member of Vedanta Biosciences and 4BIO CAPITAL. R.J.X. is co-founder of Jnana Therapeutics and Celsius Therapeutics, scientific advisory board member at Nestlé, and board director at MoonLake Immunotherapeutics. Y.A., M.U., K. Amafuji and Y.K. are employees of JSR corporation. J.M.N. and B.O. are employees of Vedanta Biosciences. S.C. was an employee of Vedanta Biosciences at the time of her contributions. D.R.P. is currently an employee of Novonesis. K.H., M.F., N.H., S.S., K. Atarashi, T.O. and Y.A. have filed international patent application PCT/JP2024/008014. The other authors declare no competing interests.

**Additional information**
**Correspondence and requests for materials** should be addressed to Ramnik J. Xavier or Kenya Honda.

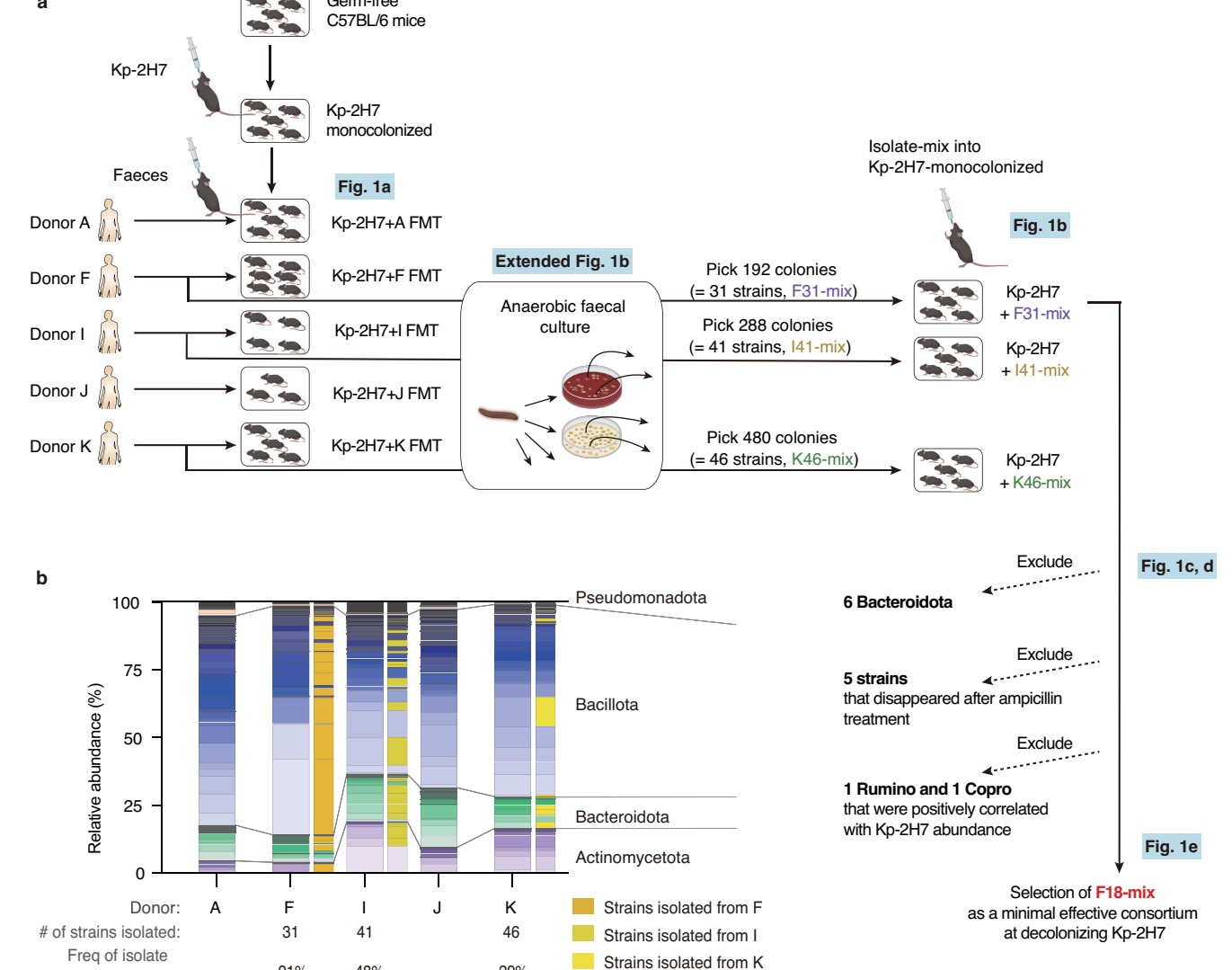

**Extended Data Fig. 1 | Isolation of bacterial strains from healthy human gut microbiota that are capable of decolonizing *Klebsiella pneumoniae*.**
**a**, Schematic representation of the strategy for isolating *Klebsiella*-decolonizing commensals from healthy human gut microbiota. Faeces from donors F, I, and K were cultured anaerobically on various types of agar with different growth media, including EG, mGAM, BHK, CM0151, MRS, and BL. A total of 192, 288, and 480 bacterial colonies were picked and sequenced from donors F, I, and K microbiota, respectively. From these, 31, 41, and 46 strains were identified from

donors F, I, and K, respectively, and subjected to gnotobiotic screening. The 31 strains derived from donor F were further evaluated until a minimal effector consortium of 18 strains, referred to as F18-mix, was identified. **b**, Microbiome compositions of faeces from healthy human donors were determined by PacBio-based full-length 16S rRNA gene sequencing. Dark to light yellows represent the sets of amplicon sequence variants (ASVs) corresponding to the 31 strains from donor F, 41 strains from donor I, and 46 strains from donor K that account for 91%, 48%, and 29% of the total sequences, respectively.

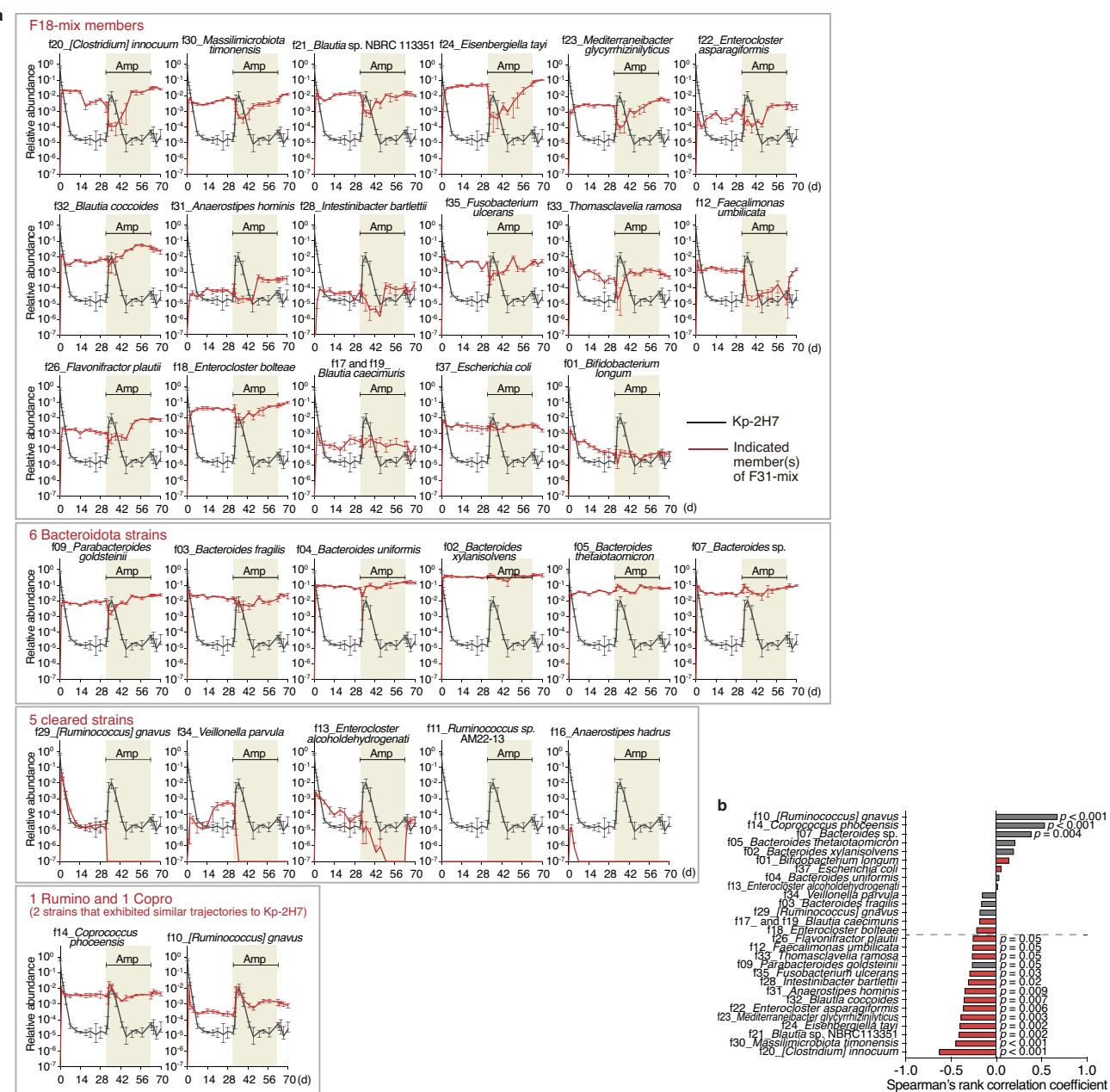

**Extended Data Fig. 2 | Selection of F18 members based on differential response to ampicillin treatment. a**, GF B6 mice (n = 5) monocolonized with Kp-2H7 were orally administered F31-mix. Ampicillin (200 mg/L) was added to the drinking water from day 32 to 63. Faeces were collected longitudinally and relative abundance of Kp-2H7 and each member of the F31 consortium was determined by qPCR using strain-specific primer sets (with the exception of f17 and f19 *Blautia caecimuris* strains, which could not be distinguished). Average data from two independent experiments are shown. F31 members that exhibited inverse trajectories to Kp-2H7 were thought to be necessary for Kp-2H7 decolonization, whereas those that behaved similarly to or independently of Kp-2H7 were thought to be dispensable. Data are shown as median ± IQR. **b**, Spearman's rank correlation coefficient (two-sided) quantifying the association between Kp-2H7 relative abundance and each member of the F31 consortium during ampicillin treatment. Significant negative correlations were found between Kp-2H7 abundance and most of the F18 strains (red).

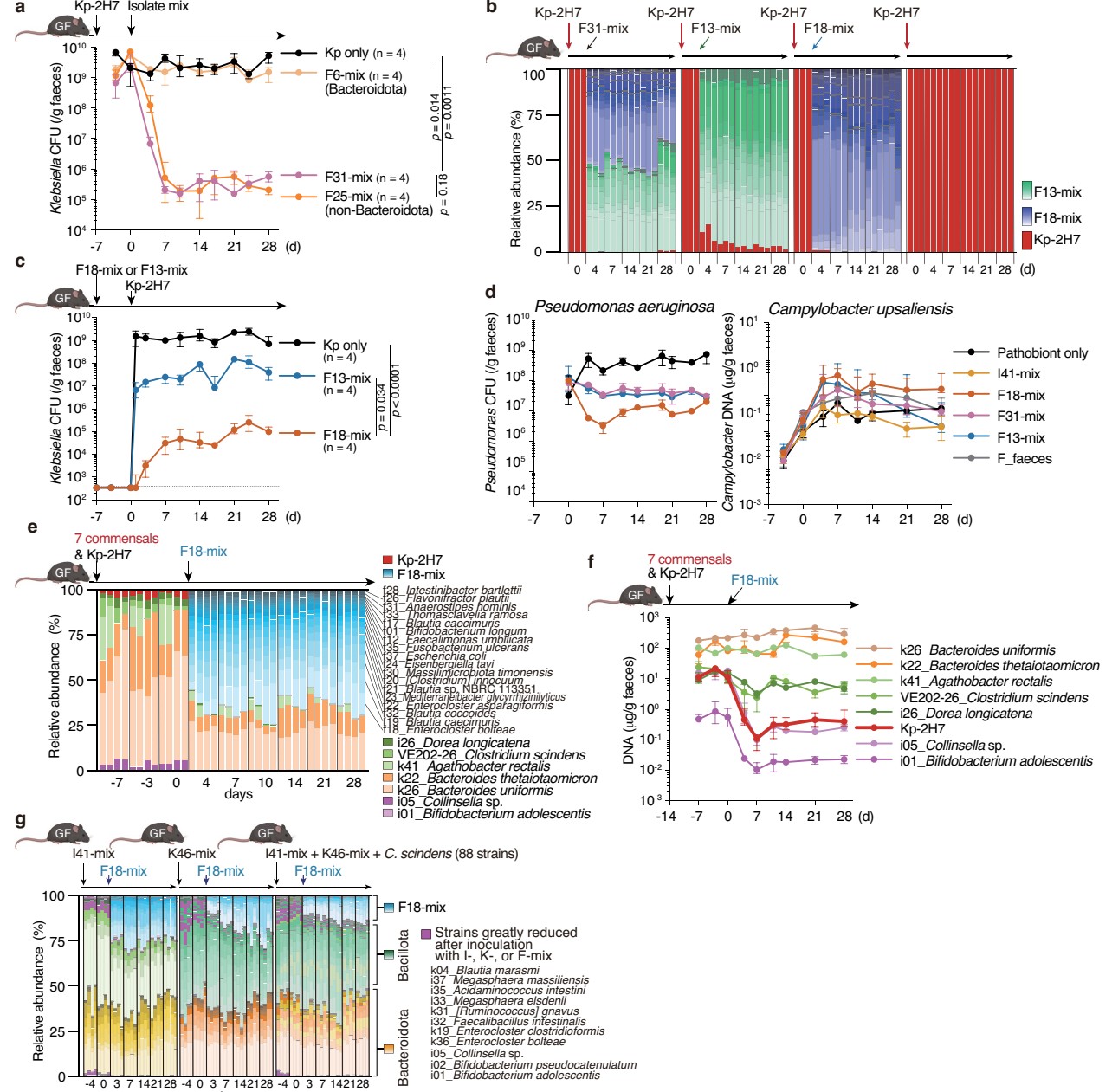

**Extended Data Fig. 3 | Effects of F18-mix on pathogenic and commensal bacteria. a, b**, GF B6 mice were monocolonized with Kp-2H7 and then treated with the indicated bacterial mix including F6 Bacteroidota-mix and F25 non-Bacteroidota-mix (n = 4 per group). Kp-2H7 faecal CFUs were counted over time (**a**), and full-length 16S rRNA gene sequencing was performed on longitudinally-collected faecal samples from each mouse (**b**). **c**, To examine the colonization resistance effect of F18-mix, GF B6 mice were first colonized with either F18-mix or F13-mix and then inoculated with Kp-2H7 on day 7. Kp-2H7 faecal CFUs were counted over time. **d**, GF B6 mice (n = 4 per group) were monocolonized with *Pseudomonas aeruginosa* or *Campylobacter upsaliensis*, followed by oral administration of the indicated bacterial mix. CFUs (*P. aeruginosa*) or level of bacterial DNA (*C. upsaliensis*) in longitudinal faecal samples was determined by culture or qPCR. **e, f**, GF B6 mice were colonized

with Kp-2H7 along with 7 commensal strains chosen from our culture collection, then treated with F18-mix. Faecal abundance of each bacterial strain was quantified by qPCR using strain-specific primer sets. The relative abundance (**e**) and DNA concentration (**f**) of each strain is shown. **g**, GF B6 mice (n = 4 per group) were colonized with either I41-mix, K46-mix, or both of these consortia together along with a *C. scindens* strain (totaling 88 strains), followed by oral inoculation with F18-mix without prior antibiotic treatment. Full-length 16S rRNA gene sequencing was performed on faecal samples. Strains substantially reduced after F18-mix treatment are displayed in purple, including *Bifidobacterium*, *Collinsella*, and *Megasphaera* strains. Data are expressed as median ± IQR, representative of two independent experiments, and compared by Kruskal-Wallis test using the Benjamini-Hochberg correction for multiple comparisons at day 28 compared to F31-mix (**a**) or F18-mix (**c**).

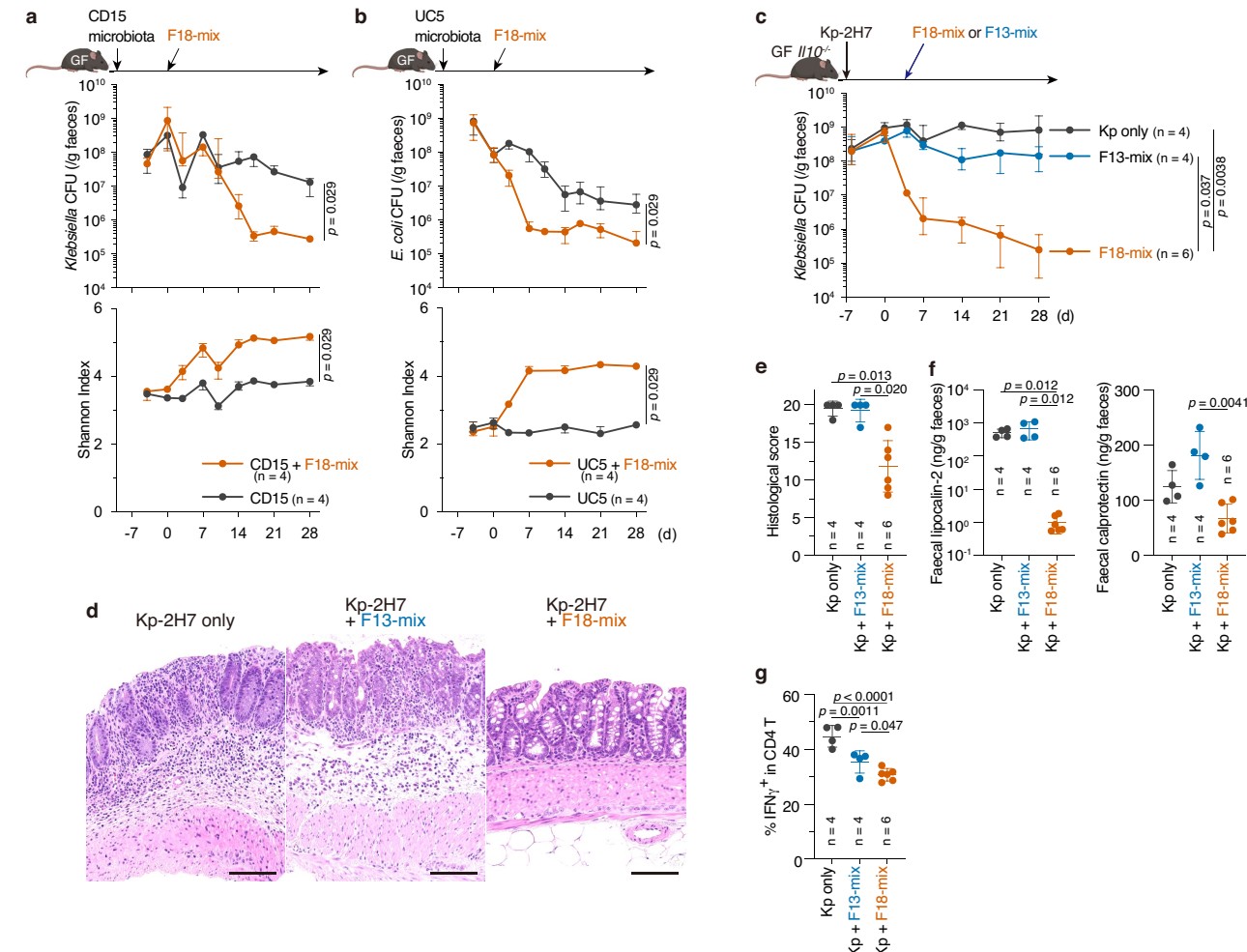

**Extended Data Fig. 4 | Effects of F18-mix in an IBD model. a, b,** GF B6 mice were colonized with faecal microbiota from either a patient with Crohn's disease (CD#15) containing a high level of *K. pneumoniae* (**a**) or from a patient with ulcerative colitis (UC#5) containing ESBL⁺ *E. coli* (**b**). All mice were subsequently treated with vancomycin, and half received oral F18-mix administration four times over two days. CFUs of *K. pneumoniae* and *E. coli* (upper panels) and Shannon index (lower panels) of the faecal microbiota were examined longitudinally and compared by Mann-Whitney U test at day 28 (two-sided).

**c-g,** GF *Il10*⁻/⁻ mice were monocolonized with Kp-2H7, then orally administered the indicated bacterial mix seven days later. Representative haematoxylin and eosin staining of the colon (scale bar = 100 μm) (**d**), histological colitis scores (**e**), faecal lipocalin-2 and calprotectin levels (**f**), and frequency of IFNγ⁺ cells among colonic lamina propria CD4⁺TCRβ⁺ T cells (**g**) are shown. In panels **a-c** and **e-g**, median ± IQR are shown, representative of two independent experiments. Statistical analysis was performed using the Kruskal-Wallis test with the Benjamini-Hochberg correction for multiple comparisons.

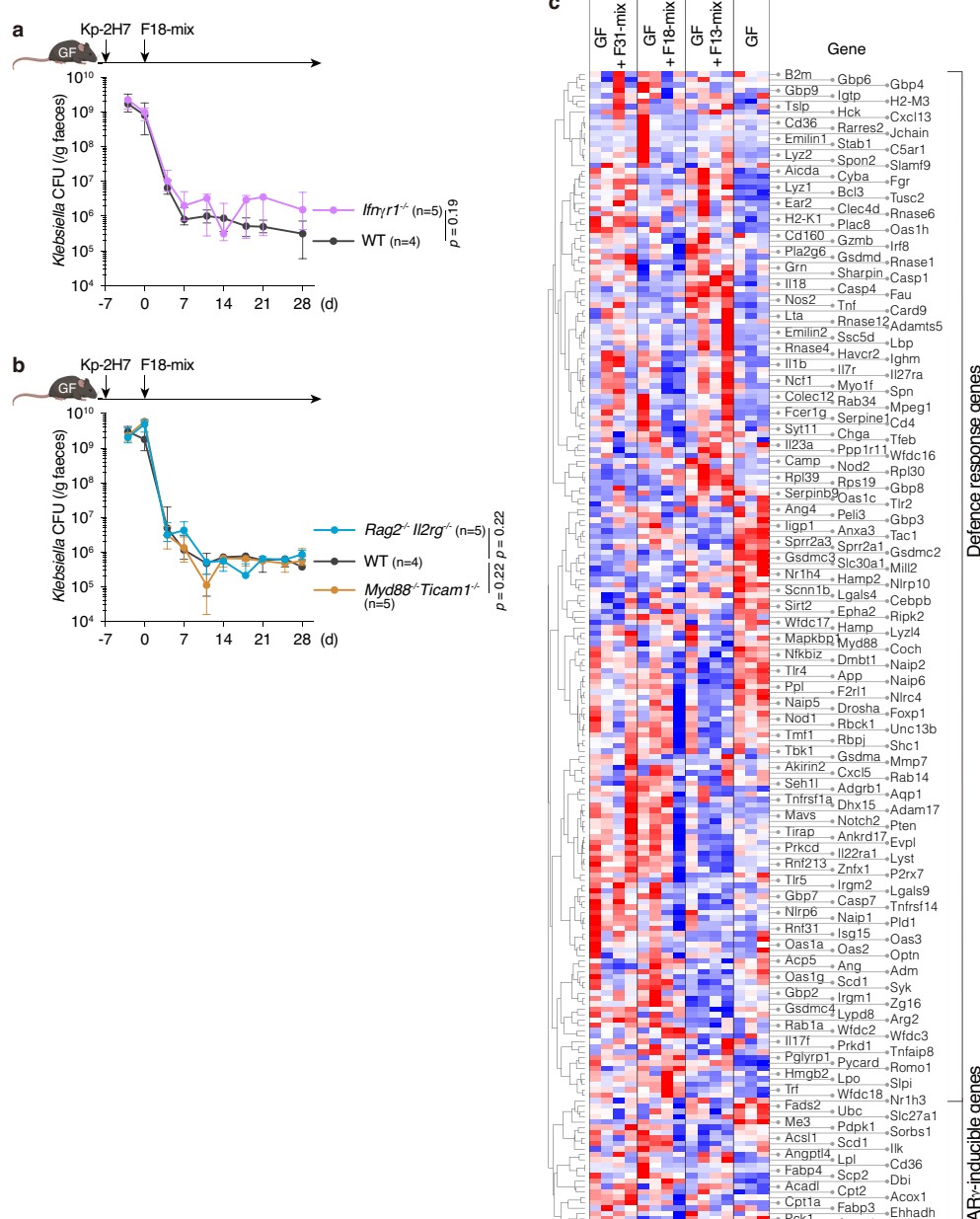

**Extended Data Fig. 5 | F18-mix decolonizes Kp-2H7 by a mechanism independent of the host's major immune system. a, b,** GF *Ifngr1⁻/⁻* (**a**), *Myd88⁻/⁻Ticam1⁻/⁻*, *Rag2⁻/⁻Il2rg⁻/⁻* (**b**), or wild-type (WT) B6 mice were monocolonized with Kp-2H7, followed by oral administration of F18-mix. Faecal Kp-2H7 CFUs were counted until day 28 post-F18-mix administration. Data are expressed as median ± IQR and compared by Mann-Whitney U test (two-sided) (**a**) or Kruskal-Wallis using the Benjamini-Hochberg correction for multiple comparisons (**b**) on samples at day 28. **c,** Heatmap depicts the expression of genes associated with defence response and peroxisome proliferator-activated receptor γ (PPARγ)-inducible genes in colonic epithelial cells from mice colonized with the indicated bacterial mix. Defence response genes and PPARγ-inducible genes were selected based on gene ontology term.

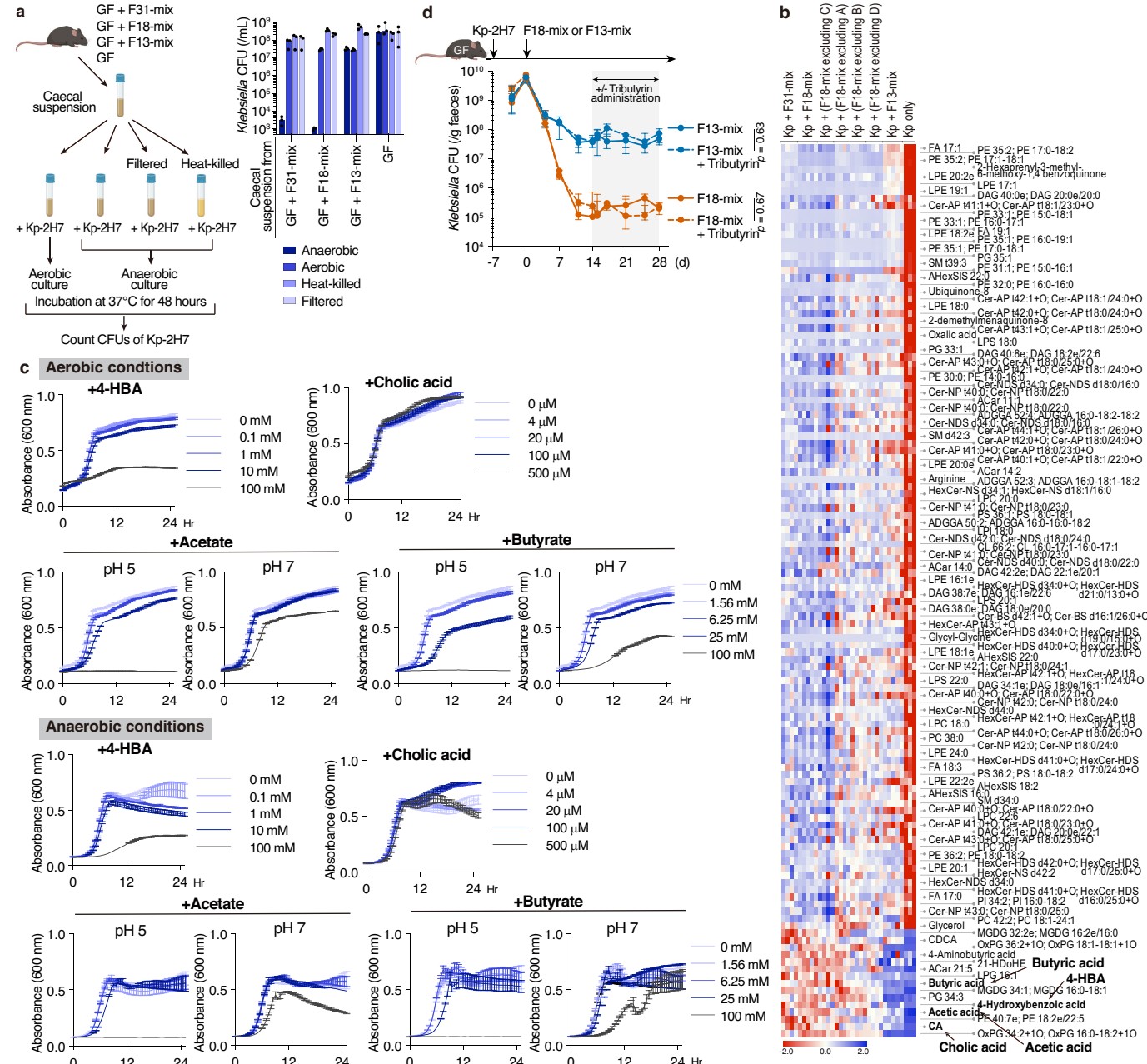

**Extended Data Fig. 6 | Exploration of mechanisms involved in *Klebsiella* reduction. a**, Kp-2H7 was incubated *in vitro* under aerobic or anaerobic conditions with caecal suspensions from uncolonized GF mice or GF mice colonized with F31-, F18-, or F13-mix with or without prior heat-inactivation or filtration (0.22 μm). Kp-2H7 CFUs were counted after a 48 hr incubation at 37 °C in n = 3 biological replicates. Data are expressed as median ± IQR. **b**, GF mice (n = 3-5 per group) were colonized with Kp-2H7 and then treated with the indicated bacterial mix. Strains designated as F18-mix excluding A, B, C, or D are detailed in Fig. 1c. Caecal contents were collected on day 28 and subjected to targeted and non-targeted LC-MS/MS, GC-MS, or LC-QTOF/MS analyses. Heatmap depicts the z-score of each metabolite that showed a correlation with faecal Kp-2H7 CFUs (Pearson's coefficient −0.6 > r > 0.6). 4-HBA, 4-hydroxybenzoic acid. **c**, Kp-2H7

was incubated with various chemical compounds at different concentrations in M9 medium under both aerobic and anaerobic conditions at 37 °C. Media supplemented with acetate or butyrate were adjusted to a final pH of 5.0 or 7.0 using NaOH. Bacterial growth was monitored by measuring absorbance at 600 nm every 0.5 hr using a microplate reader. Data are mean ± SEM from n = 3 biological replicates per condition. **d**, GF B6 mice were colonized with Kp-2H7 and subsequently treated with either F18-mix or F13-mix on day 0. From day 14, tributyrin (5 g/kg body weight) or a vehicle control was administered orally once daily for two weeks. Faecal CFUs of Kp-2H7 were counted through day 28 and are presented as median ± IQR. The day 28 data were compared using the Mann-Whitney U test (two-sided).

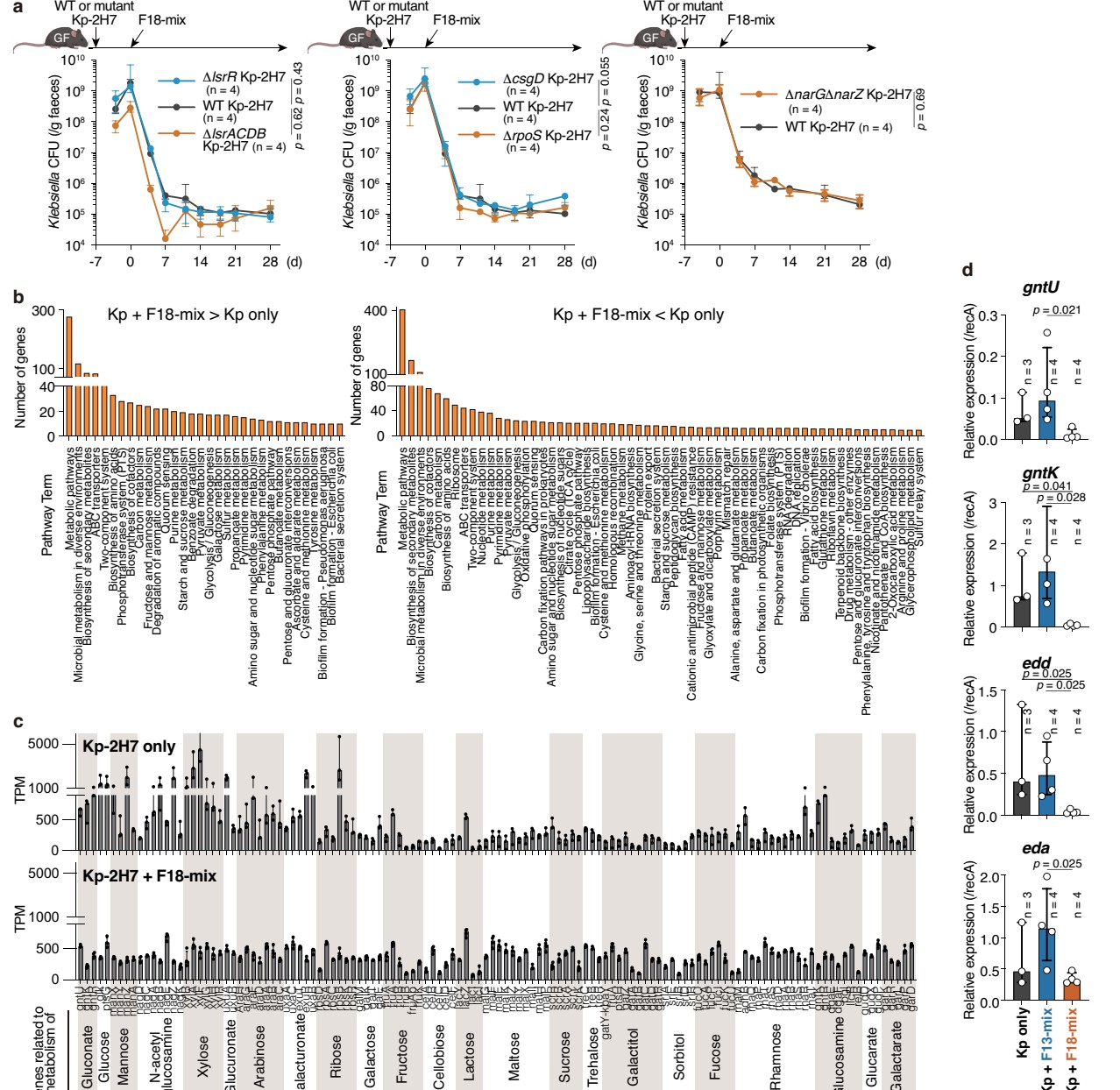

**Extended Data Fig. 7 | F18-mix suppresses *Klebsiella* through nutrient competition, rather than by affecting quorum sensing, biofilm formation, stress response, or nitrate respiration. a**, GF B6 mice were colonized with Kp-2H7 (wild-type or the indicated mutant strain), followed by oral administration of F18-mix. Faecal Kp-2H7 CFUs were counted. Data are presented as median ± IQR. The day 28 data were compared using the Mann-Whitney U test (two-sided) or the Kruskal-Wallis test with the Benjamini-Hochberg correction for multiple comparisons. **b, c**, GF mice were inoculated with either Kp-2H7 alone (n = 3) or Kp-2H7+F18-mix (n = 4). Two days after F18-mix administration, faecal samples were collected and subjected to bacterial RNA extraction and sequencing. KEGG pathway enrichment analysis was performed on Kp-2H7 genes with significantly different expression (q<0.001) between the groups. The number of genes within each pathway (consisting of >10 genes) that were up- or down-regulated in the Kp-2H7+F18-mix group compared to the Kp-2H7 only group are shown in (**b**). The expression levels of genes involved in carbohydrate metabolism, measured in transcripts per million (TPM), are displayed in (**c**). **d**, Expression of gluconate metabolism genes of Kp-2H7 in the faeces of mice colonized with Kp-2H7 only, Kp-2H7+F13-mix, or Kp-2H7+F18-mix was examined by qPCR in two technical replicates. Each dot represents data from an individual mouse. Data are presented as median ± IQR (**c, d**) and were analysed using the Kruskal-Wallis test with the Benjamini-Hochberg correction for multiple comparisons (**d**).

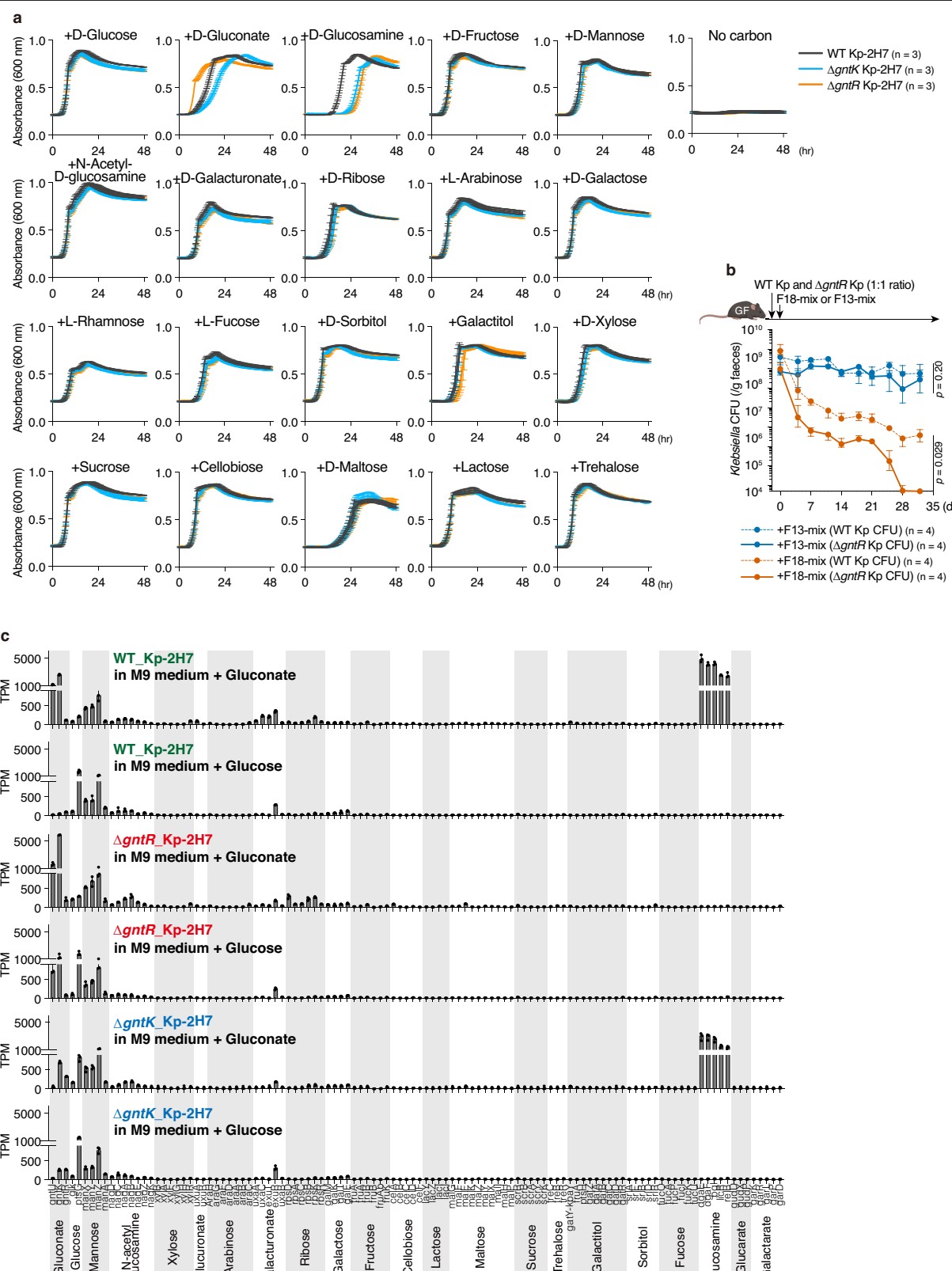

**Extended Data Fig. 8 | The role of gluconate metabolism genes in *Klebsiella* growth. a**, Wild-type (WT), Δ*gntK*, or Δ*gntR* Kp-2H7 strains were cultured at 37 °C for 48 hr in M9 minimal medium supplemented with the indicated carbohydrate (final concentration: 2mM). Bacterial growth was assessed by measuring absorbance at 600 nm. **b**, GF mice were inoculated with a 1:1 mixture of WT and Δ*gntR* Kp-2H7 and then treated with F18-mix or F13-mix. Faecal CFUs of Kp-2H7 were counted over time. Data are presented as median ± IQR, representative of two independent experiments, and were compared using the two-sided Mann-Whitney U test. **c**, WT, Δ*gntK*, or Δ*gntR* Kp-2H7 were cultured at 37 °C in M9 minimal medium supplemented with either glucose or gluconate (n = 3 biological replicates). Bacteria were collected during the early log phase (absorbance at 600 nm = 0.35) and subjected to RNA extraction and sequencing. The expression levels of genes involved in carbohydrate metabolism, expressed in transcripts per million (TPM), are presented as median ± IQR.

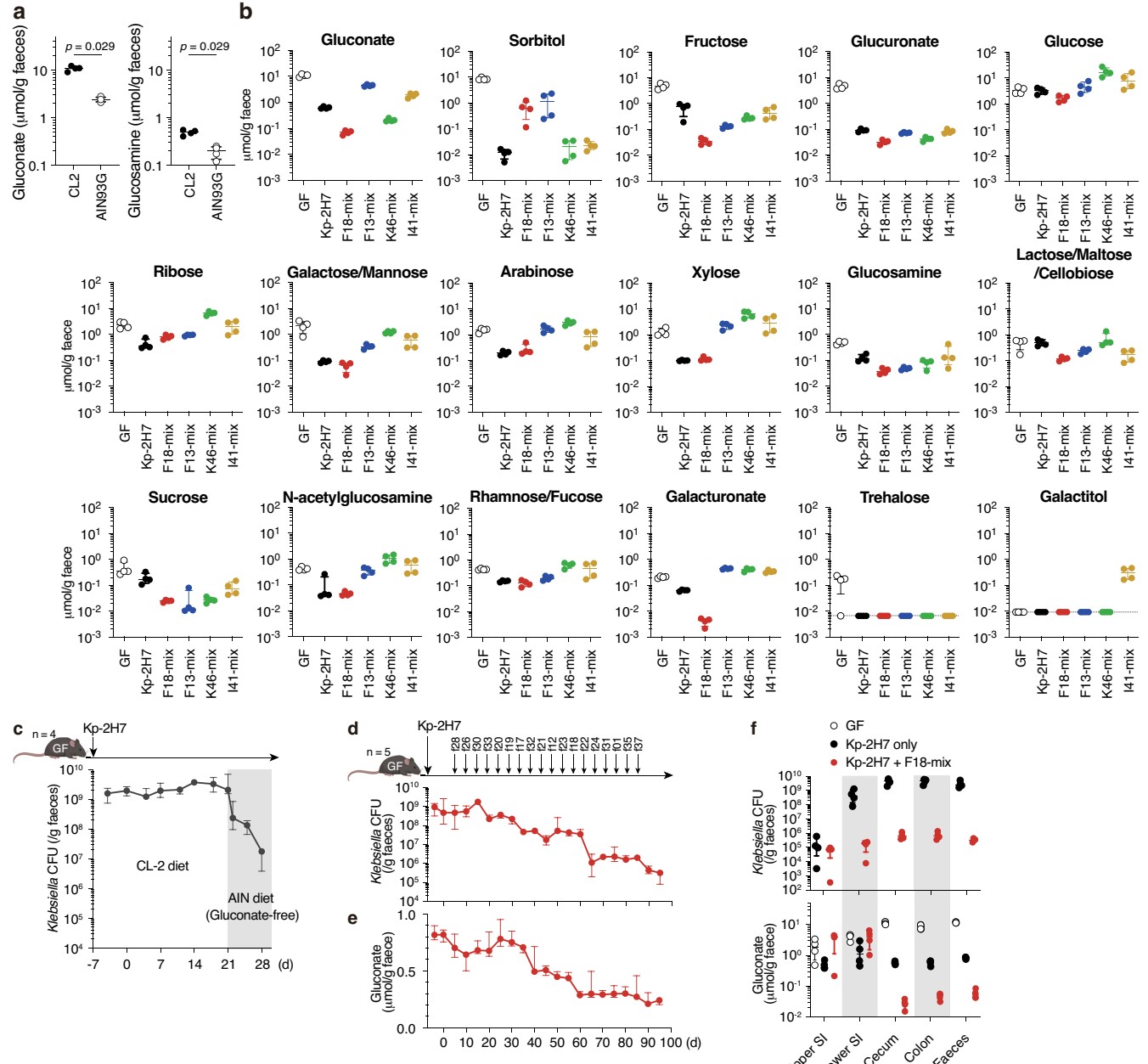

**Extended Data Fig. 9 | Utilization of gluconate by Kp-2H7 and F18-mix.**
**a**, Gluconate and glucosamine levels in the faeces of GF B6 mice (n = 4 per group) fed either a standard (CL-2) or a defined (AIN93G, gluconate- and glucosamine-free) diet were measured by LC-MS/MS. High levels of gluconate were detected in faeces from GF mice on the CL-2 diet. Substantial levels of faecal gluconate were also found in mice fed the gluconate-deficient AIN93G diet, implying that sources of gut luminal gluconate include both dietary intake and host production. In contrast, faecal glucosamine levels were very low in mice fed the CL-2 diet and became almost undetectable in those on the AIN93G diet, suggesting that glucosamine is primarily derived from dietary sources. Data are presented as median ± IQR, representative of two independent experiments, and were analysed using the two-sided Mann-Whitney U test. **b**, Faecal carbohydrate levels in GF mice colonized with Kp-2H7 or the indicated

bacterial mix (F18-mix, F13-mix, K46-mix, or I41-mix) fed a CL-2 diet were measured using LC-MS/MS (n = 4 per group), and the results are presented as median ± IQR. The data are representative of two independent experiments. **c**, GF mice on a CL-2 diet were monocolonized with Kp-2H7, and then switched to a defined, gluconate-free AIN93G diet on day 21. Kp-2H7 faecal CFUs are shown as median ± IQR. **d, e**, GF mice were monocolonized with Kp-2H7 and subsequently inoculated with individual members of F18-mix at 5-day intervals over a total period of 95 days. Faecal CFUs of Kp-2H7 (**d**) and gluconate levels (**e**) were measured throughout the study. Data are expressed as median ± IQR. **f**, *Klebsiella* CFUs and gluconate levels in the upper and lower intestinal lumen of GF mice colonized with Kp-2H7 or Kp-2H7+F18-mix (n = 4 per group). SI, small intestine. Data are shown as median ± IQR.

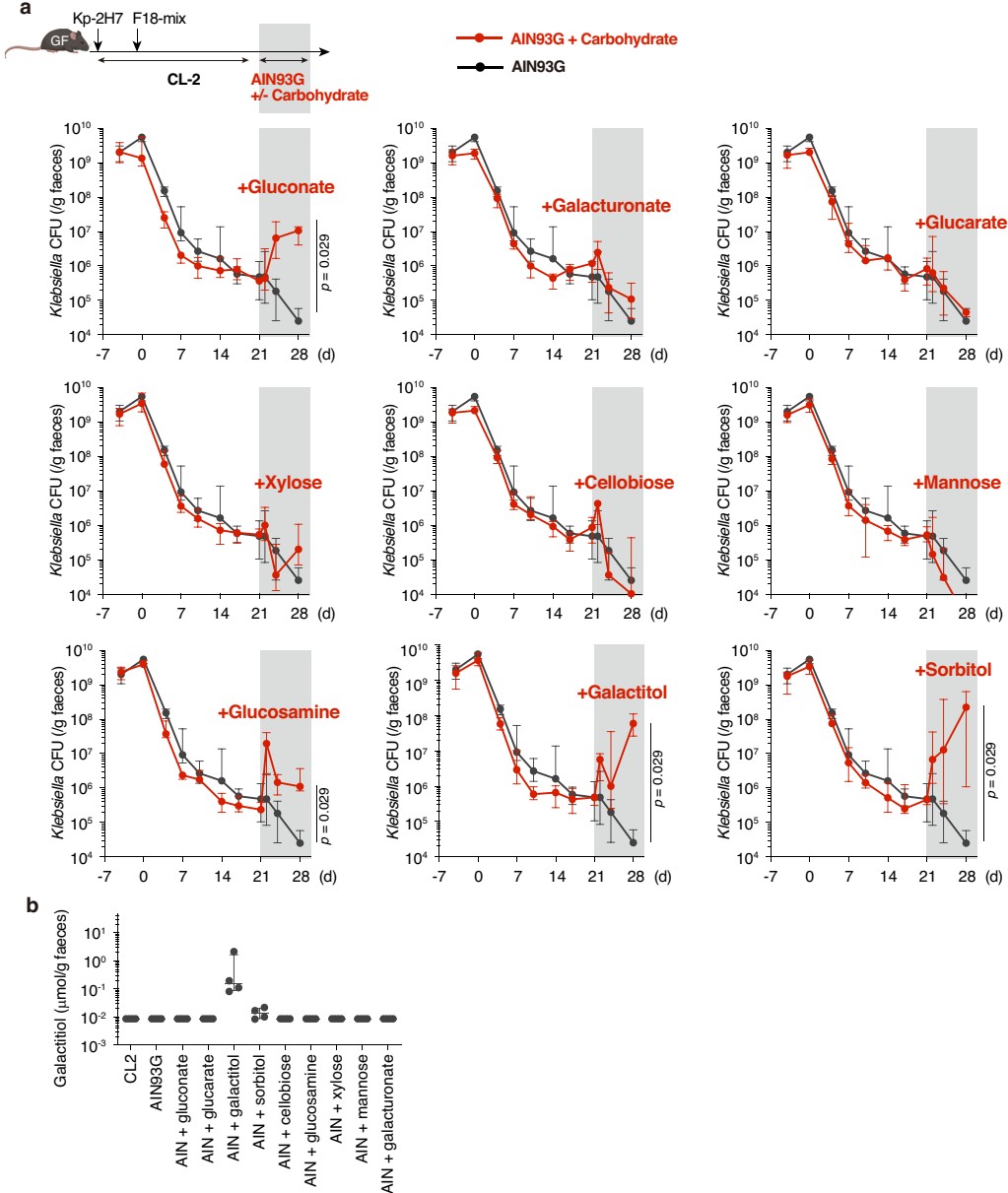

**Extended Data Fig. 10 | Impact of carbohydrate supplementation on the efficacy of F18-mix-mediated Kp-2H7 suppression. a**, GF B6 mice (n = 4 per group) were monocolonized with Kp-2H7 and then orally administered F18-mix seven days later. On day 21, their diet was switched from the standard CL-2 diet to a defined, gluconate-deficient (but glucose-rich) AIN93G diet supplemented with the indicated carbohydrate at 10% of total calories. Faecal Kp-2H7 CFUs are displayed as median ± IQR, representative of two independent experiments, and were analysed using the Kruskal-Wallis test with the Benjamini-Hochberg correction for multiple comparisons. **b**, Faecal galactitol levels were measured by LC-MS/MS (n = 4 per group). Faecal galactitol levels were below the detection limit in mice on either the CL-2 or AIN93G diet. However, faecal galactitol was successfully detected in mice fed an AIN93G diet supplemented with 10% galactitol, validating our galactitol detection assay. Data are shown as median ± IQR.

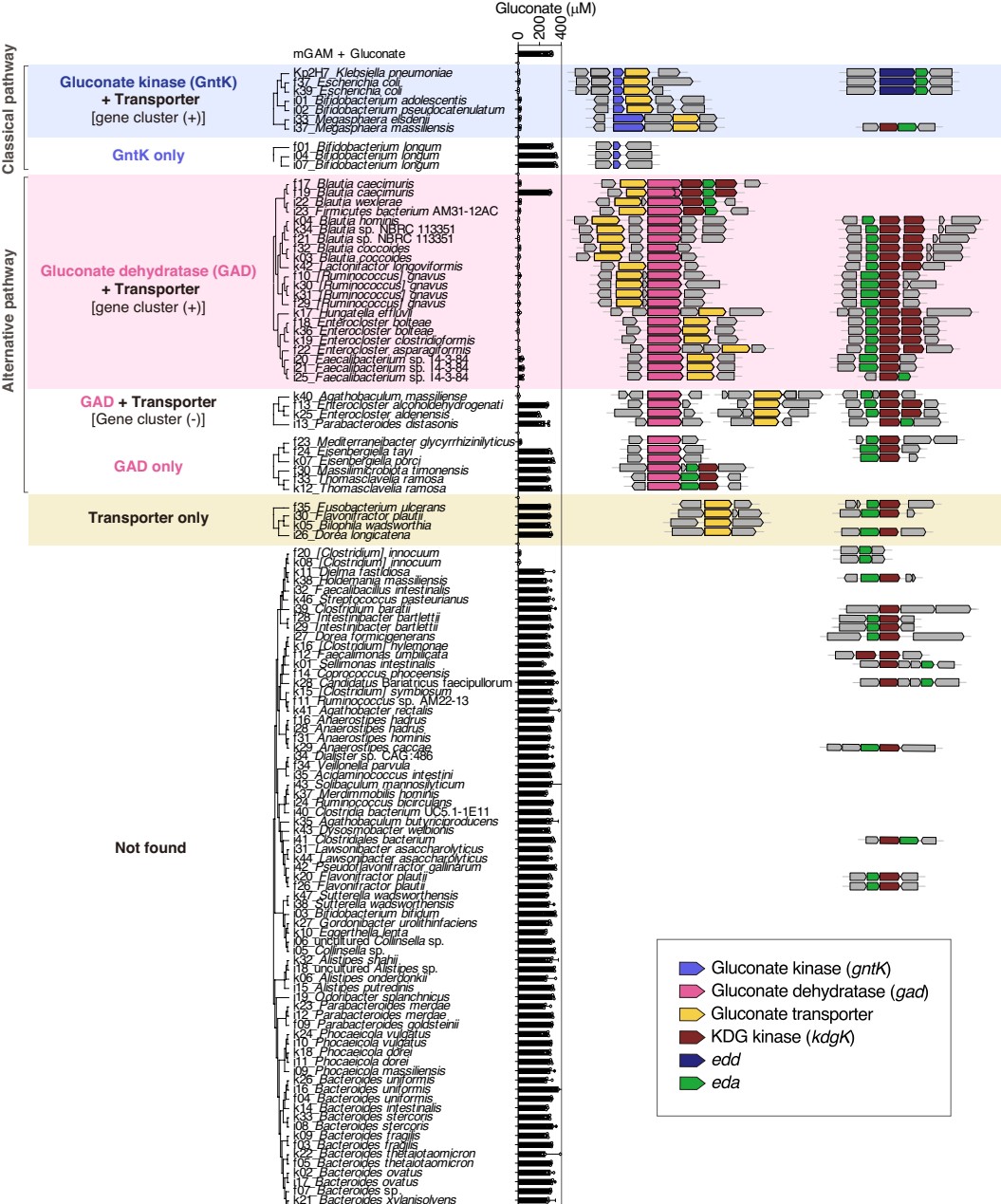

**Extended Data Fig. 11 | Carriage of classical and alternative gluconate metabolism genes by commensal strains.** Bacterial strains isolated from donors F, K, or I were cultured in mGAM broth containing 300 μM gluconate for 48 hr at 37 °C (n = 3 biological replicates). Gluconate concentration in the culture supernatant was measured by LC-MS/MS and is depicted in the middle bar graph. Data are shown as median ± IQR. Genomes of cultured strains were sequenced and examined for carriage of genes putatively involved in gluconate metabolism. For classical pathway genes, gluconate kinase (*gntK*, MKMCEHOJ_02531) and gluconate transporter sequences (MKMCEHOJ_02530,

MKMCEHOJ_02505) from the f37 *E. coli* strain were used as the reference. For alternative pathway genes, gluconate dehydratase (*gad*, EAOGLLOI_00767), gluconate transporter sequences (EAOGLLOI_00766, EAOGLLOI_00912), 2-dehydro-3-deoxygluconokinase (*kdgK*, EAOGLLOI_00768), and 2-dehydro-3-deoxyphosphogluconate aldolase (*eda*, EAOGLLOI_00769) from the f17 *Blautia caecimuris* strain were used as the reference. Asterisk indicates that the gluconate dehydratase in the f19 *Blautia caecimuris* strain is nonfunctional due to a frameshift mutation. GntK, gluconate kinase. GAD, gluconate dehydratase.

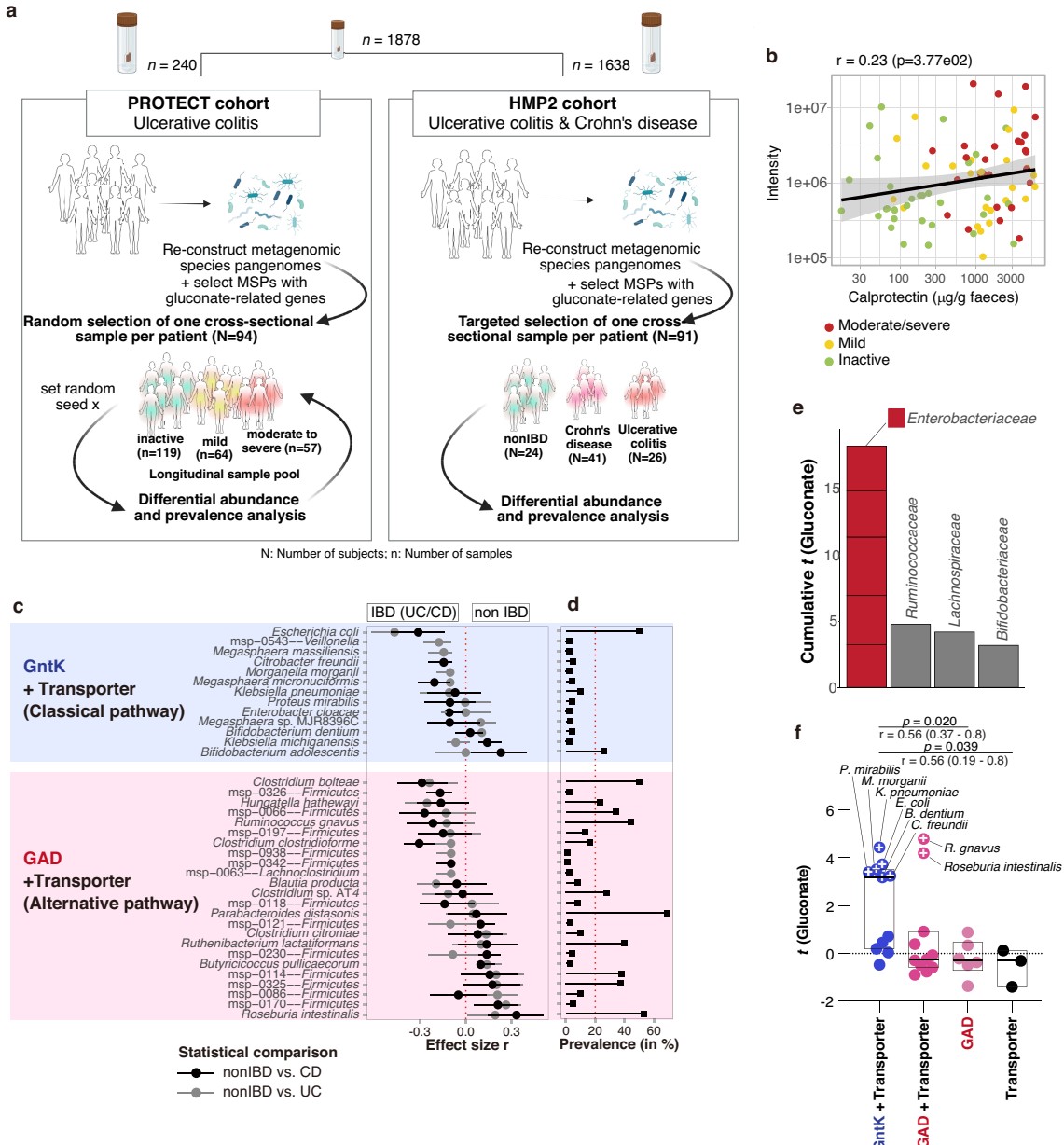

**Extended Data Fig. 12 | Association between disease state and species carrying gluconate kinase operon genes in patients with IBD. a**, Cohort details. For PROTECT (N = 94), analysis was performed iteratively with varied cross-sectional sample selections from mild (n = 64), moderate/severe (n = 57), and non-IBD (n = 119) longitudinal sample pools. For HMP2, samples were selected based on most extreme calprotectin value from a given patient: CD (N = 41), UC (N = 26), and non-IBD (N = 24). N: Number of subjects; n: Number of samples. **b**, Spearman correlation analysis of faecal calprotectin (in μg/g) versus gluconate mass intensity in a subset of 84 PROTECT samples with varying degrees of UC severity: inactive (n = 33, green), mild (n = 24, yellow), and moderate/severe (n = 27, red). Standard errors of model estimates are highlighted in grey. **c**, Comparative abundance analysis of MSPs with gluconate genes in HMP2 samples between CD (black, N = 41) or UC (grey, N = 26) vs. non-IBD (N = 24). Dots and extending line segments represent r effect sizes and CIs obtained by bootstrapping. Taxa were categorized by gluconate gene presence and combination in MSP bins. **d**, MSP prevalence (in %) in the HMP2

cohort (N = 91), as percentage. Taxa without species annotation and reference genome remain 'msp-' labelled, so gene combinations could not be verified in complete assemblies. **e, f**, The mixed-effects model quantifies the relationship between species abundance and gluconate in stool samples (n = 223), while controlling for calprotectin and subject ID. Cumulative *t*-values (model coefficients corrected by standard error) highlight the predominantly positive associations of gluconate with the Enterobacteriaceae clade in HMP2 (**e**). Circles in (**f**) indicate MSPs having gluconate metabolism genes, and those with plus marks represent associations between species and gluconate with BH adjusted *p*-values < 0.05. MSPs with gluconate kinase + transporter genes had stronger gluconate associations than those with only transporter (two-sided Mann-Whitney U, p-0.04) or gluconate dehydratase (p-0.02). The effect size r was computed with CIs from bootstrapping. Boxplots show median (center line) and IQR (box); whiskers extend to 1.5 x IQR. GntK, gluconate kinase. GAD, gluconate dehydratase.

| | |
|---|---|

# Reporting Summary

## Statistics

For all statistical analyses, confirm that the following items are present in the figure legend, table legend, main text, or Methods section.

| n/a | Confirmed | |
|---|---|---|
| ☐ | ☒ | The exact sample size (*n*) for each experimental group/condition, given as a discrete number and unit of measurement |
| ☐ | ☒ | A statement on whether measurements were taken from distinct samples or whether the same sample was measured repeatedly |
| ☐ | ☒ | The statistical test(s) used AND whether they are one- or two-sided *Only common tests should be described solely by name; describe more complex techniques in the Methods section.* |
| ☐ | ☒ | A description of all covariates tested |
| ☐ | ☒ | A description of any assumptions or corrections, such as tests of normality and adjustment for multiple comparisons |
| ☐ | ☒ | A full description of the statistical parameters including central tendency (e.g. means) or other basic estimates (e.g. regression coefficient) AND variation (e.g. standard deviation) or associated estimates of uncertainty (e.g. confidence intervals) |
| ☐ | ☒ | For null hypothesis testing, the test statistic (e.g. *F*, *t*, *r*) with confidence intervals, effect sizes, degrees of freedom and *P* value noted *Give P values as exact values whenever suitable.* |
| ☒ | ☐ | For Bayesian analysis, information on the choice of priors and Markov chain Monte Carlo settings |
| ☒ | ☐ | For hierarchical and complex designs, identification of the appropriate level for tests and full reporting of outcomes |
| ☐ | ☒ | Estimates of effect sizes (e.g. Cohen's *d*, Pearson's *r*), indicating how they were calculated |

*Our web collection on statistics for biologists contains articles on many of the points above.*

## Software and code

Policy information about availability of computer code

| Data collection | LC-MS/MS data were obtained using Analyst software version 1.7.1.<br>Light Cycler 480 1.5.1 software was used for collecting qPCR data.<br>16S rRNA gene sequencing was sequenced using PacBio Revio system. Bacterial whole-genome was sequenced using the Illumina MiSeq and PacBio Sequel.<br>RNA sequence was performed using NovaSeq 6000 (Illumina Inc.) HiSeq X (Illumina Inc.).<br>Tn-seq was performed using HiSeq 2500 (Illumina Inc.). |
|---|---|
| Data analysis | For 16S rRNA gene sequencing, the sequenced reads were uploaded to the DADA2 (version 1.30.0) in R (version 4.3.3) to construct ASVs using the filterAndTrim function. FL16s-ASVs were then subjected to a homology search against 16S rRNA gene sequences extracted from publicly available genomes (downloaded from GenBank on September 12, 2023) using BLASTN with a maximum e-value cut-off of 1e-10. Top hits were determined by the highest bitscore.<br>Bacterial whole-genome sequencing were assembled using Unicycler. Taxonomic assignment of the genomes was determined by classify_wf of GTDB-tk version 2.3.0 with GTDB database R214. NCBI taxonomy of fastANI reference genome related to the genome of each strain was retrieved using NCBI-genome-download version 0.3.3 (ncbi-genome-download; DOI: 10.5281/zenodo.8192432) and rankedlineage.dmp from NCBI taxonomy database (downloaded on 14/09/2023).<br>Transcriptome analysis was performed using Trimmomatic version 0.39, FASTX-Toolkit version 0.0.13, bowtie2 version 2.3.4.1. (in vivo) or 2.4.4 (in vitro), featureCounts3 version 2.0.1., STAR version 2.7.2b, and DESeq274 version 1.28.1 (in vivo) or 1.30.1 (in vitro).<br>Tn-seq was performed using Trimmomatic version 0.39, FASTX-Toolkit version 0.0.13, minimap2 version 2.17-r941, bowtie2 version 2.4.2, and featureCounts3 version 1.5.2. |

For manuscripts utilizing custom algorithms or software that are central to the research but not yet described in published literature, software must be made available to editors and reviewers. We strongly encourage code deposition in a community repository (e.g. GitHub). See the Nature Portfolio guidelines for submitting code & software for further information.

## Data

Policy information about [availability of data](availability of data)

All manuscripts must include a [data availability statement](data availability statement). This statement should provide the following information, where applicable:

- Accession codes, unique identifiers, or web links for publicly available datasets
- A description of any restrictions on data availability
- For clinical datasets or third party data, please ensure that the statement adheres to our [policy](policy)

> The data of Tn-Seq data and RNA sequence data of host, Kp in vivo, and Kp in vitro are deposited in the DNA Data Bank of Japan under BioProject PRJDB17114. Genome sequences of the 31, 41, and 46 strains isolated from donor F, I and K, respectively are deposited in the DNA Data Bank of Japan under BioProject PRJDB17661.

## Research involving human participants, their data, or biological material

Policy information about studies with [human participants or human data](human participants or human data). See also policy information about [sex, gender (identity/presentation), and sexual orientation](sex, gender) and [race, ethnicity and racism](race, ethnicity and racism).

| Reporting on sex and gender | The outcomes were consistent across sexes. Therefore, we have chosen not to delineate the results by these factors to avoid undue emphasis on distinctions that were not meaningful in the context of our study. |
| --- | --- |
| Reporting on race, ethnicity, or other socially relevant groupings | In this study, no distinction of race or ethnicity was made at any point in the collection of human faecal samples or any other analysis. |
| Population characteristics | Human faecal samples were obtained from healthy human donors (n=5). The median age of healthy human was 33 years (range: 30-41), and all lf them were Japanese. |
| Recruitment | Human faecal samples were collected at Keio University and Keio University Hospital. Informed consent was obtained from each subject. |
| Ethics oversight | Human faecal samples were obtained from healthy human donors following the protocol approved by the Institutional Review Board of Keio University School of Medicine (approval numbers #20150075, #20140211, and #20150075). |

Note that full information on the approval of the study protocol must also be provided in the manuscript.

# Field-specific reporting

Please select the one below that is the best fit for your research. If you are not sure, read the appropriate sections before making your selection.

☒ Life sciences    ☐ Behavioural & social sciences    ☐ Ecological, evolutionary & environmental sciences

For a reference copy of the document with all sections, see [nature.com/documents/nr-reporting-summary-flat.pdf](nature.com/documents/nr-reporting-summary-flat.pdf)

# Life sciences study design

All studies must disclose on these points even when the disclosure is negative.

| Sample size | No statistical methods were used to predetermine sample size. Sample size for animal experiments were chosen according to preliminary pilot studies, balancing statistical robustness and animal welfare. |
| --- | --- |
| Data exclusions | No data was excluded |
| Replication | For all the experiments, reproducibility was verified by conducting the experiment at least twice, which yielded comparable results. |
| Randomization | In animal experiments, samples were randomly allocated into experimental groups. Sex-matched littermates were used and the experiments were designed to test a single variable. |
| Blinding | The CFU counting and LC-MS/MS analysis were performed by technicians who were blinded to expected outcomes. The histological analysis were evaluated blind by two investigators. All other studies were not strictly blinded because the measurements were based on objective analysis methods. |

# Reporting for specific materials, systems and methods

We require information from authors about some types of materials, experimental systems and methods used in many studies. Here, indicate whether each material, system or method listed is relevant to your study. If you are not sure if a list item applies to your research, read the appropriate section before selecting a response.

## Materials & experimental systems

| n/a | Involved in the study |
|---|---|
| ☐ | ☒ Antibodies |
| ☒ | ☐ Eukaryotic cell lines |
| ☒ | ☐ Palaeontology and archaeology |
| ☐ | ☒ Animals and other organisms |
| ☒ | ☐ Clinical data |
| ☒ | ☐ Dual use research of concern |
| ☒ | ☐ Plants |

## Methods

| n/a | Involved in the study |
|---|---|
| ☒ | ☐ ChIP-seq |
| ☐ | ☒ Flow cytometry |
| ☒ | ☐ MRI-based neuroimaging |

## Antibodies

| | |
|---|---|
| Antibodies used | anti-CD3e (BUV395; BD Biosciences, 1:1000), CD4 (BUV737; BD Biosciences, 1:1000), TCRβ (BV421; Biolegend, 1:1000), and IFN-γ (FITC; Biolegend, 1:1000) |
| Validation | All the antibodies are commercially available. Quality validations were performed by each manufacturer. Validation statements are provided on the manufacturer's website.<br>anti-CD3e (BUV395; BD Biosciences) (https://www.bdbiosciences.com/en-us/products/reagents/flow-cytometry-reagents/research-reagents/single-color-antibodies-ruo/buv395-hamster-anti-mouse-cd3e.563565)<br>CD4 (BUV737; BD Biosciences) (https://www.bdbiosciences.com/en-us/products/reagents/flow-cytometry-reagents/research-reagents/single-color-antibodies-ruo/buv737-rat-anti-mouse-cd4.612843)<br>TCRβ (BV421; Biolegend) (https://www.biolegend.com/ja-jp/products/brilliant-violet-421-anti-mouse-tcr-beta-chain-antibody-7251)<br>IFN-γ (FITC; Biolegend) (https://www.biolegend.com/ja-jp/products/fitc-anti-mouse-ifn-gamma-antibody-995) |

## Animals and other research organisms

Policy information about studies involving animals; ARRIVE guidelines recommended for reporting animal research, and Sex and Gender in Research

| | |
|---|---|
| Laboratory animals | C57BL/6 mice, maintained under specific-pathogen-free or germ-free (GF) conditions, were purchased from Sankyo Laboratories Japan, SLC Japan, or CLEA Japan. GF and gnotobiotic mice were bred and maintained within the gnotobiotic facility of Keio University School of Medicine or the JSR-Keio University Medical and Chemical Innovation Center. Il10-/- and Ifngr1-/- mice were purchased from Jackson Laboratories. Myd88-/-Trif-/- and Rag2-/-γc-/- mice were purchased from Oriental Bio Service, Japan. All animals were maintained under a 12-h light–dark cycle. A temperature of 20–24 °C and a humidity of 40–60% were used for the housing conditions. |
| Wild animals | The study did not involve wild animals. |
| Reporting on sex | We did not categorize the sexes because the differences between the sexes did not affect the results of the experiment. |
| Field-collected samples | The study did not involve samples collected from the field. |
| Ethics oversight | All animal experiments were approved by the Keio University Institutional Animal Care and Use Committee. |

Note that full information on the approval of the study protocol must also be provided in the manuscript.

## Plants

| | |
|---|---|
| Seed stocks | *Report on the source of all seed stocks or other plant material used. If applicable, state the seed stock centre and catalogue number. If plant specimens were collected from the field, describe the collection location, date and sampling procedures.* |
| Novel plant genotypes | *Describe the methods by which all novel plant genotypes were produced. This includes those generated by transgenic approaches, gene editing, chemical/radiation-based mutagenesis and hybridization. For transgenic lines, describe the transformation method, the number of independent lines analyzed and the generation upon which experiments were performed. For gene-edited lines, describe the editor used, the endogenous sequence targeted for editing, the targeting guide RNA sequence (if applicable) and how the editor was applied.* |
| Authentication | *Describe any authentication procedures for each seed stock used or novel genotype generated. Describe any experiments used to assess the effect of a mutation and, where applicable, how potential secondary effects (e.g. second site T-DNA insertions, mosiacism, off-target gene editing) were examined.* |

# Flow Cytometry

## Plots

Confirm that:

☒ The axis labels state the marker and fluorochrome used (e.g. CD4-FITC).

☒ The axis scales are clearly visible. Include numbers along axes only for bottom left plot of group (a 'group' is an analysis of identical markers).

☒ All plots are contour plots with outliers or pseudocolor plots.

☒ A numerical value for number of cells or percentage (with statistics) is provided.

## Methodology

**Sample preparation**

Lymphocytes were collected from the large intestines using the method described below. The intestines were dissected longitudinally and washed with PBS to remove all luminal contents. All samples were incubated in 15 mL of Hanks' balanced salt solution (HBSS) containing 5 mM EDTA for 20 min at 37°C in a shaking water bath to remove epithelial cells. Thereafter, after removal of any remaining epithelial cells, muscular layers and fat tissues using forceps, the samples were cut into small pieces and incubated in 10 mL of RPMI1640 containing 4% foetal bovine serum (FBS), 0.5 mg/mL collagenase D (Roche Diagnostics), 0.5 mg/mL dispase II (Roche Diagnostics), and 40 μg/mL DNase I (Roche Diagnostics) for 50 min at 37°C in a shaking water bath. Thereafter, the resultant digested tissues were washed with 10 mL of HBSS containing 5 mM EDTA, resuspended in 5 mL of 40% Percoll (GE Healthcare), and underlaid with 2.5 mL of 80% Percoll in a 15-mL Falcon tube. Percoll gradient separation was performed by centrifugation at 850 × g for 25 min at 25°C. Lymphocytes were collected from the interface of the Percoll gradient and washed with RPMI1640 containing 10% FBS, and then stimulated with 50 ng/mL PMA and 750 ng/mL ionomycin (both from Sigma) in the presence of Golgistop (BD Biosciences) at 37°C for four hours.

**Instrument**

BD LSRFortessa (BD Biosciences)

**Software**

Flowjo software (TreeStar)

**Cell population abundance**

The Purity was determined by running a purity check of the sorted populations after the sort was completed.

**Gating strategy**

All samples were initially gated using forward scatter and side scatter to identify events corresponding to cells, and then using forward scatter height vs. area to enrich for single cells, next alive cells were selected by negativity for viability dye. TCD4+ T cells were defined as a CD4+ TCRβ+ CD3e+ subset within the live lymphocyte gate.

☒ Tick this box to confirm that a figure exemplifying the gating strategy is provided in the Supplementary Information.

