## [Peer Review file · Nature]

Manuscript Title: Commensal consortia decolonize Enterobacteriaceae via ecological control

Reviewer Comments & Author Rebuttals

Reviewer Reports on the Initial Version:

Referees' comments:

Referee #1 (Remarks to the Author):

Furuichi and colleagues have assembled commensal bacterial consortia and tested their ability to suppress in vivo colonization with Enterobacteriaceae. The investigators use the strategy developed and perfected by the Honda laboratory of transferring human fecal samples into germ free mice, in this case testing for the establishment of colonization resistance, and then cultivating commensal bacteria from the most effective fecal samples and testing them, in various combinations, for their ability to suppress in vivo growth of *Klebsiella pneumoniae* and other Enterobacteriaceae. The authors were able to reduce the consortia composition from 31 to 18 strains, however further reduction resulted in reduced colonization resistance. One member of the consortium which contributed to the consortium's effectiveness is a strain of *Escherichia coli*. Of note, the 18-member consortium was not highly effective at reducing colonization with VRE or providing resistance to *C. difficile* infection. Testing the F18-mix's ability to suppress Enterobacteriaceae colonization in a range of immunocompromised mouse strains indicated that the resistance mechanism did not depend on the stimulation of innate or adaptive immune responses and thus reflected more direct inter-bacterial interactions. Characterization of the *K. pneumoniae* transcriptome demonstrated that transcription of genes involved with carbon and amino acid metabolism were impacted by administration of the F18-mix. Using a transposon mutant library, the authors identified genes involved in carbohydrate metabolism that markedly reduce *K. pneumoniae* fitness in colonized mice. In particular, deficiency of *gntR*, which regulates gluconate metabolism, resulted in enhanced F18-mediated clearance of *K. pneumoniae* compared to wild type Kp over the course of 28 days. The authors demonstrate that patients with active IBD have reduced commensal species encoding genes related to gluconate metabolism.

Suppression of Enterobacteriaceae in the gut is an important challenge and this manuscript uses a beautifully designed and methodical approach to identify commensal bacterial consortia that markedly reduce intestinal colonization with these potential pathogens. Furthermore, the use of transcriptomic analyses and a transposon mutant library to identify the gluconate operon as a contributor to microbiota-mediated colonization resistance is a nice step forward. This manuscript complements recent work published by the Kamada, Stecher, Xavier, Ubeda and Strowig laboratories demonstrating the contributions of amino acid (Kamada), galactitol (Stecher, Xavier), fructose (Ubeda) and diverse carbohydrate (Strowig) depletion on suppression of different Enterobacteriaceae species. It remains unclear from the work presented in this manuscript whether gluconate is just one of many potential carbon sources that contributes to colonization resistance or if it stands out as the most critical. The impact and importance of this manuscript would be significantly enhanced by additional studies that contrast the presented results with recently

published work on this topic.

Additional issues that should be addressed:

1.) The kinetics of Kp clearance shown in this manuscript are quite prolonged, with persistence of Kp for up to 49 days in mice receiving the F18-mix. Although the authors describe V1-V2 16S rRNA gene amplification and MiSeq sequencing of fecal pellets in the methods section, the figure legends state that strain engraftment was determined by qPCR with strain specific primers. Demonstrating engraftment and persistence of administered strains over the course of Kp clearance might provide insights into the prolonged clearance kinetics and whether the administered consortia compositions change over time. Importantly, pan-16S sequencing as opposed to strain-specific PCR would also demonstrate that mice had not acquired contaminating bacterial strains that might influence Kp clearance.

2.) While the authors comment that enhanced clearance of Kp lacking *gntR* indicates that regulation of gluconate is involved in colonization resistance, the mechanism remains largely unexplained. Does continued transcription of the gluconate operon in the absence of gluconate reduce Kp growth and render it more easily cleared? This idea, however, is not supported by the unimpaired *in vitro* growth of these mutant strains in media with glucose. Is it possible that *gntR* is regulating the expression of genes involved in the metabolism of other carbon sources?

3.) The authors demonstrate that *gntK*-deficient Kp clearance is enhanced by the F18 and F13-mix while *gntR*-deficient Kp are depleted by the F18-mix but unaffected by the F13-mix. It is not clear why this would be the case. Can the authors speculate on this interesting finding?

4.) The experiment shown in Extended Data Figure 6d involves administration of Tributyrin to mice that have already suppressed colonization with *Klebsiella pneumoniae*. The result that tributyrin does not further suppress Kp is not surprising since the F18-mix is likely already producing high levels of butyrate in the cecum/colon. The authors could, perhaps, determine whether tributyrin administration to mice colonized with consortia that do not produce butyrate enhances *K. pneumoniae* clearance.

5.) The authors measure a wide range of carbon sources in feces from GF mice and demonstrate that gluconate is present at the highest concentrations. Presumably fecal concentrations of gluconate are impacted by the diet. The authors administered the AIN diet, which is gluconate deficient, and demonstrate reduced colonization with Kp. The authors should provide fecal gluconate concentrations in fecal samples from mice on the different diets. Furthermore, since Kp colonizes the length of the gut, the authors should determine whether colonization of the small intestine is impacted by gluconate and measure gluconate concentrations in jejunum and ileum.

6.) In the final figure, the authors demonstrate that patients with active IBD have higher representation of bacterial strains encoding gluconate related genes. Given that expansion of Enterobacteriaceae in the presence of bowel inflammation is known to occur, this result is predictable. Furthermore, demonstrating greater representation of gluconate-related genes in the microbiota of patients with active IBD does not implicate regulation of gluconate metabolism as a driver of IBD flares. It might be interesting, however, to measure fecal concentration of gluconate in this patient population and to determine whether lower densities of Enterobacteriaceae correlate

with reduced gluconate concentrations.

Referee #2 (Remarks to the Author):

The manuscript by Furuichi et al. identified microbial consortia that suppress the growth of *Klebsiella* and other Enterobacteriaceae in monocolonized ex-germ-free mice. After identifying the consortium, the authors search for a mechanism by which these microbes can displace the pathobionts. The follow-up comprehensive experiments explore the role of host immunity and bacterial metabolites. Finally, a genetic screen using transposon mutagenesis in *Klebsiella* and follow-up studies led to the identification of the availability of some carbohydrates, and in particular gluconate, as a key factor for *Klebsiella* growth. The community of 18 strains reduced the availability of gluconate from *Klebsiella*, and excess gluconate administration rescued *Klebsiella* colonization.

Overall, this is an important study that identified a microbial community that can suppress the growth of *Klebsiella* and other Enterobacteriaceae. In the past decade, it has been appreciated how bacterial metabolism and carbohydrate utilization are essential for bacterial growth and colonization resistance. For example, see studies showing that *Salmonella* uses galactarate and glucarate (PMID: 27309805), that *E. coli* depletes galactilol to compete with *Salmonella* (PMID: 34610296), and that *Klebsiella oxytoca* and other commensals provide colonization resistance against *Klebsiella pneumoniae* via carbohydrate utilization, particularly of beta-glucosidic sugars (PMID: 34610293). Importantly, the latter study identified *Blautia* and *Enterocloster* as key organisms for competition with *Klebsiella pneumoniae*, and these bacteria were also present in the F18-mix of this manuscript. Prior manuscripts showing the importance of carbohydrate competition in colonization resistance against Enterobacteriaceae need to be acknowledged and discussed, and in particular, the *Klebsiella* study, which is particularly relevant to this manuscript.

The results presented are overall convincing. The data presentation is excellent, and the conclusions are supported by the data. I suggest a couple of experiments to strengthen the authors' conclusions further:

1. All experiments were done in ex-germ-free mice colonized with the pathobiont and displaced by the consortia (e.g. F18-mix). While this is essential for understanding the mechanism, it would be important to repeat this study in SPF mice. For example, do the consortia (e.g. F18-mix) displace *Klebsiella* also in SPF IL-10KO mice?
2. In all experiments, mice were colonized with *Klebsiella* first, then *Klebsiella* was displaced by the F18-mix. What about the reverse experiment? Does the F18-mix prevent *Klebsiella* colonization?
3. Administration of gluconate partly rescues *Klebsiella* intestinal growth in the presence of the F18-mix (Fig. 3h) What happens if other carbohydrates are administered to mice? I understand the rationale of focusing on gluconate, but it seems that other carbohydrates may also be important.

Referee #3 (Remarks to the Author):

The study, titled "Rationally-defined microbial consortia suppress multidrug-resistant proinflammatory Enterobacteriaceae via ecological control," submitted by Furuichi et al., describes a minimal microbial community that suppresses Enterobacteriaceae abundance in gnotobiotic mice, along with a potential mechanism underlying this interaction. The authors initially identified complex communities from healthy donor stool samples that were most efficient in competing with *Klebsiella pneumoniae* Kp77 2H7. They then employed an ecosystem deconstruction approach to determine which community members are required to suppress the growth of Kp77 2H7. The authors found that a community of 18 strains (F18) was sufficient and essential for suppressing growth. This approach was complemented by generating a transposon mutant library of *K. pneumoniae* to identify potential mechanisms contributing to its suppression by F18. The authors identified gluconate as an important nutrient for *K. pneumoniae* in the intestine, which is depleted by the F18 community but not by less effective communities. A high gluconate diet also increased *K. pneumoniae* abundance. Finally, the authors present correlative data showing a high abundance of species containing gluconate-related genes in patients with inflammatory bowel disease (IBD).

Conceptually, identifying defined microbial intestinal communities that suppress specific community members, such as Enterobacteriaceae, is relevant given the overall negative impact of Enterobacteriaceae overgrowth on human health. The study is based on the concept that *K. pneumoniae* contributes to IBD, but unlike established pathogens like *C. difficile*/VRE that cause human disease, the potential adverse events of *K. pneumoniae* are still under investigation, and its relevance in humans remains to be shown. The study has several strengths, including a rigorous study design, a labor-intensive systematic interrogation of potential mechanisms using the transposon mutant library, transparent presentation of data, including experiments that did not lead to finding a mechanism, and identification of a potential pathway that can be targeted to suppress overgrowth of Enterobacteriaceae. However, there are some concerns outlined below.

Major Concerns:

1. The study primarily relies on microbial consortia suppressing *K. pneumoniae* in germ-free mice colonized with *K. pneumoniae*. It remains unclear if the same effect and mechanism would apply in the context of a complex community harboring high levels of *K. pneumoniae*. While the entire study does not need to be done in a complex community, the effect of F18 and verification of competition for gluconate should be demonstrated in the context of germ-free mice colonized with stool samples from IBD patients with high levels of *K. pneumoniae*.
2. One limitation with this approach that needs to be acknowledged is that while F18 reduces gluconate availability, it is not the only difference between F18 and other communities tested by the authors. There could be other mechanisms that are concurrently affected by the F18 community, contributing to the suppression of *K. pneumoniae*. Ideally, this would require knocking down the gluconate pathway in members of the F18 community, but that can be very tedious.
3. In the absence of experiments using *K. pneumoniae* in the context of complex communities mentioned above in point #1, it remains unclear if there are any off-target positive or negative

effects of the F18 community on the rest of the microbial community.

4. The authors use human data from a published cohort to corroborate some of the findings. The correlation of gluconate and calprotectin is nominally significant ($p=0.0377$) with a very small effect size ($r=0.23$). As Enterobacteriaceae are known to be elevated in IBD, the high abundance of species containing gluconate-related genes is difficult to interpret, as Enterobacteriaceae harbor these genes. The authors should consider comparing patients with and without high levels of *K. pneumoniae* to determine if gluconate pathways correlate with their prevalence.

Minor Concerns:

1. While more semantics, I would not consider these as rationally defined communities, as they were identified after trying all combinations of isolated strains to determine which ones would suppress *K. pneumoniae* and then finding a potential mechanism. Rational selection would indicate that communities were developed based on a mechanism known to be important for suppressing *K. pneumoniae*.

2. In line 123, *E. coli* and *Fusobacterium* are included, as well as antibiotic-resistant taxa such as *Enterocloster* sp. Are these strains relevant to give to people since they are considered pathobionts? Please add this to the discussion.

3. In line 233, instead of mutants, please say "transposon insertion mutants" for clarity.

4. In line 266, the header should start at the subsequent paragraph.

5. In line 296, there is a typo: "Clostiridium" should be corrected to "Clostridium."

6. Figure 4e and Ext. data Figure 12C are too small; adjust for legibility.

7. In Ext. Data Figure 6, there is a typo: "suspention" should be corrected to "suspension."

Author Rebuttals to Initial Comments:

We are very grateful to the editor and reviewers for their invaluable comments, which were thorough, thoughtful, and constructive. Based on these comments, we have performed additional experiments and made the necessary revisions to the manuscript. All new changes to the text since the last review are marked in blue font. We believe that the new experimental data have sufficiently addressed the reviewers' comments and further increased the value of this work. For each specific point below, the referee's comment is in bold-italics and our response is in plain type.

Referee #1:

Furuichi and colleagues have assembled commensal bacterial consortia and tested their ability to suppress in vivo colonization with Enterobacteriaceae. The investigators use the strategy developed and perfected by the Honda laboratory of transferring human fecal samples into germ free mice, in this case testing for the establishment of colonization resistance, and then cultivating commensal bacteria from the most effective fecal samples and testing them, in various combinations, for their ability to suppress in vivo growth of Klebsiella pneumoniae and other Enterobacteriaceae. The authors were able to reduce the consortia composition from 31 to 18 strains, however further reduction resulted in reduced colonization resistance. One member of the consortium which contributed to the consortium's effectiveness is a strain of Escherichia coli. Of note, the 18-member consortium was not highly effective at reducing colonization with VRE or providing resistance to C. difficile infection. Testing the F18-mix's ability to suppress Enterobacteriaceae colonization in a range of immunocompromised mouse strains indicated that the resistance mechanism did not depend on the stimulation of innate or adaptive immune responses and thus reflected more direct inter-bacterial interactions. Characterization of the K. pneumoniae transcriptome demonstrated that transcription of genes involved with carbon and amino acid metabolism were impacted by administration of the F18-mix. Using a transposon mutant library, the authors identified genes involved in carbohydrate metabolism that markedly reduce K. pneumoniae fitness in colonized mice. In particular, deficiency of gntR, which regulates gluconate metabolism, resulted in enhanced F18-mediated clearance of K. pneumoniae compared to wild type Kp over the course of 28 days. The authors demonstrate that patients with active IBD have reduced commensal species encoding genes related to gluconate metabolism.

Suppression of Enterobacteriaceae in the gut is an important challenge and this manuscript uses a beautifully designed and methodical approach to identify commensal bacterial consortia that markedly reduce intestinal colonization with these potential pathogens. Furthermore, the use of transcriptomic analyses and a transposon mutant library to identify the gluconate operon as a contributor to microbiota-mediated colonization resistance is a nice step forward. This manuscript complements recent work published by the Kamada, Stecher, Xavier, Ubeda and Strowig laboratories demonstrating the contributions of amino acid (Kamada), galactitol (Stecher, Xavier), fructose (Ubeda) and diverse carbohydrate (Strowig) depletion on suppression of different Enterobacteriaceae species. It remains unclear from the work presented in this manuscript whether gluconate is just one of many potential carbon sources that contributes to colonization resistance or if it stands out as the most critical. The impact and importance of this manuscript would be significantly enhanced by additional studies that contrast the presented results with recently published work on this topic.

We are grateful for the reviewer's thoughtful assessment and the acknowledgment of our efforts. We greatly appreciate the insightful feedback provided. We agree with the reviewer's suggestion to provide additional insights into the significance and complementarity of our study in relation to previous reports.

To examine the contribution of a diverse group of carbohydrates (including those mentioned by the reviewer) to the mechanism of F18-mix-mediated *Enterobacteriaceae* decolonization, we performed LC-MS-based faecal carbohydrate profiling of GF mice colonized with Kp-2H7, F18-mix, F13-mix, K46-mix, or I41-mix. This analysis revealed that Kp-2H7 monocolonization resulted in marked reductions in gluconate, galacturonate, glucuronate, N-acetylglucosamine, sorbitol, galactose/mannose, ribose, arabinose, and xylose (new **Extended Data Fig. 11b**), indicating that Kp-2H7 may have a preference for these carbon sources. Importantly, F18-mix efficiently utilized an overlapping set of carbohydrates (including gluconate), thereby potentially limiting nutrient availability and reshaping ecological niches within the intestine (new **Extended Data Fig. 11b**). In contrast, F13-mix, K46-mix, and I41-mix demonstrated distinct carbohydrate utilization profiles. Notably, only gluconate consumption exhibited a clear correlation with the consortia's ability to suppress Kp-2H7 (new **Extended Data Fig. 11b**). These findings support the hypothesis that gluconate consumption/competition plays a crucial role in F18-mix-mediated *Enterobacteriaceae* decolonization.

As the reviewer highlighted, galactitol, cellobiose (a type of β -glucosidic sugar), and glucarate have been reported to be involved in the growth of *Klebsiella* in the intestine and may play a mechanistic role in F18-mix-mediated Kp-2H7 suppression (we have now included these important references in the revised manuscript). We therefore investigated whether providing these carbohydrates in excess would affect the ability of F18-mix to decolonize Kp-2H7. To this end, GF mice first monocolonized with Kp-2H7 and then treated with F18-mix. These mice were initially fed a nutrient-rich (CL-2) diet and were subsequently switched to a defined (AIN93G) diet. The AIN93G diet contains high levels of sucrose but lacks gluconate, sorbitol, glucosamine, and xylose. We supplemented the AIN93G diet with each of the following carbohydrates individually (final concentration, 10% of total calories): gluconate, galactitol, sorbitol, glucosamine, glucarate, galacturonate, cellobiose, mannose, xylose, or vehicle control. We found that supplementation with mannose, xylose, cellobiose, glucarate, or galacturonate did not affect Kp-2H7 sensitivity to F18-mix (new **Extended Data Fig. 12a**). In contrast, supplementation with galactitol, sorbitol, and glucosamine significantly increased Kp-2H7 levels (new **Extended Data Fig. 12a**), suggesting that these sugars may contribute to *Klebsiella* fitness *in vivo*.

Importantly, however, faecal galactitol levels were below the limit of detection regardless of whether mice were on the CL-2 or AIN93G diet using our validated LC-MS-based galactitol detection method (new **Extended Data Fig. 11b** and **12b**). Therefore, it is unlikely that competition for galactitol contributes to the observed suppression of *Klebsiella* by F18-mix. While intestinal sorbitol is abundant (**Fig. 3f** and new **Extended Data Fig. 11b**), it was not able to compensate for the restricted availability of other carbohydrates like gluconate, as Kp-2H7 was decolonized by F18-mix despite exhibiting more efficient sorbitol metabolism (new **Extended Data Fig. 11b**). Additionally, sorbitol was effectively consumed by K46-mix and I41-mix (new **Extended Data Fig. 11b**), both of which failed to fully suppress Kp-2H7 (**Fig. 1b**). Moreover, the expression of Kp-2H7 sorbitol metabolism genes was minimally perturbed by F18-mix treatment (**Extended Data Fig. 9b**). These results suggest that, although sorbitol competition cannot be entirely ruled out, it is unlikely to be a primary driver of F18-mix-mediated *Klebsiella* suppression. On the other hand, it is possible that glucosamine regulation

may contribute to *Klebsiella* suppression, especially given that GntR and GntK are involved in glucosamine metabolism (new **Extended Data Fig. 10b** and **10d**, see our response to points 2, 3 below) which, much like gluconate metabolism, uses the ED pathway. However, unlike gluconate, which is derived from both dietary sources and the host, glucosamine originates almost exclusively from dietary intake and is present at levels approximately 20 times lower than gluconate in the intestine (new **Extended Data Fig. 11a**). Therefore, while competition for glucosamine may contribute, it is more plausible that *Klebsiella* predominantly relies on gluconate as a crucial energy source for growth in the gut, and that F18-mix competes for this resource.

The above results are intriguing. In particular, even when a large amount of sorbitol is available for Kp-2H7 to utilize almost freely, it does not significantly affect the dynamics when there is competition for gluconate. However, in the absence of gluconate, if a substantial amount of sorbitol is present, Kp-2H7 can increase even in the presence of F18-mix. These new data suggest that while gluconate is clearly one of several carbon sources that can affect *Klebsiella* growth, its significance may extend beyond serving as a mere nutritional source. The presence and metabolism of gluconate may have cascading effects on microbial interaction patterns in the intestine. Gluconate metabolism may induce phenotypic changes in *Klebsiella* or alter the intestinal environment by, for instance, promoting changes in pH levels, producing inhibitory substances, or modulating signaling pathways. Such changes could potentially impact *Klebsiella*'s ability to utilize other carbohydrates, such as sorbitol and glucosamine, as effective energy sources for growth and survival. We have included these hypotheses in **Supplemental Discussion** in the revised text.

Although more research is needed to fully elucidate the rules governing competition, our new findings—thanks to the reviewer's suggestion—indicate that the decolonization capacity of F18-mix is context-dependent and can be significantly influenced by dietary components. This consideration must be kept in mind when assessing its clinical efficacy. We have included this important point in the revised text.

Additional issues that should be addressed:

1) The kinetics of Kp clearance shown in this manuscript are quite prolonged, with persistence of Kp for up to 49 days in mice receiving the F18-mix. Although the authors describe V1-V2 16S rRNA gene amplification and MiSeq sequencing of fecal pellets in the methods section, the figure legends state that strain engraftment was determined by qPCR with strain specific primers. Demonstrating engraftment and persistence of administered strains over the course of Kp clearance might provide insights into the prolonged clearance kinetics and whether the administered consortia compositions change over time. Importantly, pan-16S sequencing as opposed to strain-specific PCR would also demonstrate that mice had not acquired contaminating bacterial strains that might influence Kp clearance.

We appreciate the reviewer's valuable point. Accordingly, we performed PacBio-based full-length 16S rRNA sequencing on faecal samples from the F18-mix-mediated Kp-2H7 decolonization experiment (Indeed, we have replaced all the previous 16S rRNA sequencing data with new full-length 16S rRNA sequencing data). The results were consistent with our original submission, revealing absence of contaminating species and a persistent suppression of Kp-2H7 following F18-mix treatment (new **Extended Data Fig. 3b**).

2) While the authors comment that enhanced clearance of *Kp* lacking *gntR* indicates that regulation of gluconate is involved in colonization resistance, the mechanism remains largely unexplained. Does continued transcription of the gluconate operon in the absence of gluconate reduce *Kp* growth and render it more easily cleared? This idea, however, is not supported by the unimpaired *in vitro* growth of these mutant strains in media with glucose. Is it possible that *gntR* is regulating the expression of genes involved in the metabolism of other carbon sources?

3) The authors demonstrate that *gntK*-deficient *Kp* clearance is enhanced by the F18 and F13-mix while *gntR*-deficient *Kp* are depleted by the F18-mix but unaffected by the F13-mix. It is not clear why this would be the case. Can the authors speculate on this interesting finding?

We thank the reviewer for the insightful comments and agree that the enhanced clearance of *Kp*-2H7 lacking *gntR* in the presence of F18-mix might be related to GntR's role in regulating the metabolism of other carbon sources.

To address these comments, we performed RNA-seq analysis on wild-type (WT), $\Delta gntR$, and $\Delta gntK$ *Kp*-2H7 strains cultured in minimal medium supplemented with either glucose or gluconate. Unexpectedly, the WT strain grown in gluconate-supplemented medium exhibited marked upregulation of genes predicted to be involved in glucosamine metabolism, whereas the $\Delta gntR$ mutant did not (new **Extended Data Fig. 10d**). These new findings suggest that, in addition to its role in negatively regulating gluconate operon genes, GntR is also involved in positively regulating glucosamine metabolism genes. These new results are consistent with prior studies showing that GntR functions as a positive regulator of genes involved in citrate fermentation (J Bacteriol. 2008, 190, 7419-30), glycoside hydrolase family genes, β -phosphoglucomutase, and unlinked sugar phosphotransferase (Sci Trans Med 2019, 11, aat8418) in *Enterococcus* species.

Interestingly and consistently, the $\Delta gntK$ and $\Delta gntR$ strains grew poorly compared to WT in glucosamine-supplemented media (new **Extended Data Fig. 10b**), indicating that both genes are involved in glucosamine utilization. While the complete glucosamine metabolic pathway in *Klebsiella* has yet to be characterized, several key steps have been inferred based on predicted gene functions within the glucosamine operon: conversion of glucosamine to glucosamine-6P, then glucosamine-6P, followed by 2-keto-3-deoxygluconate-6P (KDG-6P), and ultimately pyruvate. It is likely that GntR plays a key role in positively regulating this pathway. Notably, the enzyme responsible for the initial glucosamine phosphorylation is not found in the *Klebsiella* glucosamine operon, suggesting that GntK might perform this function.

Returning to the transposon data, *gntR* mutant strains demonstrated dominance *in vivo* in mice colonized with *Kp*-TPs and *Kp*-TPs+F13-mix, rapidly outcompeting other mutant strains. In contrast, their abundance gradually declined and were eventually eliminated when co-colonized with F18-mix (**Fig. 3c** and **Extended Data Fig. 10c**). This phenotype may be linked to a hierarchy of *Klebsiella* carbohydrate preferences. Gluconate is a preferred carbon source for *Klebsiella*, and when it is abundant (such as in *Kp*-2H7 monocolonization or F13-mix co-colonization conditions) the lack of GntR-mediated repression causes gluconate operon gene upregulation, thereby enhancing gluconate utilization and conferring a competitive growth advantage. Under such conditions, there is a reduced

need to rely on other carbons such as glucosamine as an energy source, allowing robust growth of *gntR* mutant strains. However, in the presence of species capable of efficiently utilizing gluconate like F18-mix, upregulation of gluconate operon genes in *gntR* mutant Kp-2H7 becomes futile. As gluconate levels dwindle due to concurrent consumption by *Klebsiella* and F18-mix, *Klebsiella* may be forced to shift toward utilizing less-preferred carbon sources such as glucosamine. In these circumstances, GntR-mediated enhancement of glucosamine utilization becomes crucial to growth and survival, and as such *gntR* mutant strains are outcompeted and suppressed. While further research will be needed to fully elucidate these complex dynamics, these hypotheses have been carefully considered in the **Supplemental Discussion** of the revised manuscript.

4) The experiment shown in Extended Data Figure 6d involves administration of Tributyrin to mice that have already suppressed colonization with Klebsiella pneumoniae. The result that tributyrin does not further suppress Kp is not surprising since the F18-mix is likely already producing high levels of butyrate in the cecum/colon. The authors could, perhaps, determine whether tributyrin administration to mice colonized with consortia that do not produce butyrate enhances K. pneumoniae clearance.

In accordance with this comment, we evaluated the effect of butyrate supplementation on F13-mix- and F18-mix-mediated suppression of *Klebsiella*. Butyrate supplementation via tributyrin administration did not significantly alter the *in vivo* effectiveness of either F13-mix or F18-mix (new **Extended Data Fig. 7d**).

5) The authors measure a wide range of carbon sources in feces from GF mice and demonstrate that gluconate is present at the highest concentrations. Presumably fecal concentrations of gluconate are impacted by the diet. The authors administered the AIN diet, which is gluconate deficient, and demonstrate reduced colonization with Kp. The authors should provide fecal gluconate concentrations in fecal samples from mice on the different diets. Furthermore, since Kp colonizes the length of the gut, the authors should determine whether colonization of the small intestine is impacted by gluconate and measure gluconate concentrations in jejunum and ileum.

We appreciate the reviewer's insightful suggestion, which inspired a series of important experiments. We examined faecal gluconate levels in GF mice fed either the CL-2 diet or the AIN93G diet, and found that gluconate levels vary with diet (new **Extended Data Fig. 11a**). Interestingly, substantial levels of faecal gluconate were observed even in GF mice fed a gluconate-deficient AIN93G diet, implying that both dietary intake and host production contribute to the pool of intestinal gluconate.

We also assessed the levels of both gluconate and Kp-2H7 in the small intestine and colon. Kp-2H7 distribution correlated well with gluconate concentration in the colon. However, in the lower small intestine, despite F18-mix treatment suppressing Kp-2H7, gluconate concentrations remained high (new **Extended Data Fig. 11f**). This suggests that gluconate availability may not be the sole determinant of *Klebsiella* colonization, particularly in the small intestine. Therefore, as discussed in our response to the first comment, although *Klebsiella* relies on gluconate as a key energy source for growth in the gut, the impact of commensal-mediated gluconate competition may be context-dependent. This context-based variability is a crucial factor to consider when evaluating the clinical efficacy of F18-mix. We have incorporated these additional data and corresponding discussions into

the revised manuscript.

6.) *In the final figure, the authors demonstrate that patients with active IBD have higher representation of bacterial strains encoding gluconate related genes. Given that expansion of Enterobacteriaceae in the presence of bowel inflammation is known to occur, this result is predictable. Furthermore, demonstrating greater representation of gluconate-related genes in the microbiota of patients with active IBD does not implicate regulation of gluconate metabolism as a driver of IBD flares. It might be interesting, however, to measure fecal concentration of gluconate in this patient population and to determine whether lower densities of Enterobacteriaceae correlate with reduced gluconate concentrations.*

We agree with the reviewer that *Enterobacteriaceae* expansion in the IBD context is predictable and has been linked to biochemical mechanisms beyond gluconate metabolism (such as nitrate respiration). We did not intend to imply a causal relationship between the regulation of gluconate metabolism and IBD flares in the human cohort data and apologize if our description was unclear. The point of the *in silico* analysis was to assess whether our experimental findings apply to humans by (a) highlighting the positive association between faecal calprotectin and gluconate abundance in stool metabolomic data from a pediatric UC cohort, and (b) confirming the enrichment of bacterial species containing gluconate kinase operons in both pediatric UC (PROTECT) and adult IBD (HMP2) cohorts. To underline the association context of our statistical frameworks, we carefully rephrased the statements in our manuscript.

In addition, we further evaluated the existence of an association between gluconate abundance and *Enterobacteriaceae* enrichment by accounting for the extent of inflammation within the statistical framework. We first validated the gluconate candidate in our stool metabolomics data by using the following chemical reference: Sigma Aldrich S2054 (details in **Methods**). This led to annotating HN_s_QI1923 (HILIC-neg 195.0512 *m/z* at 4.34 min) from PROTECT and QI11027 (HILIC-neg 195.0512 *m/z* at 4.48 min) from HMP2. Then, we utilized a general linear mixed-effects model to assess the association between species (MSPs) and gluconate abundance, factoring in faecal calprotectin and subject ID as fixed and random effects, respectively. In the PROTECT cohort, our analysis revealed a significant positive association between gluconate and *Enterobacteriaceae* abundance, even after adjusting for faecal calprotectin levels (new **Fig. 4f**). This association was notably strong and achieved statistical significance specifically in both *Enterobacteriaceae* and non-*Enterobacteriaceae* that possess gluconate kinase operon genes (new **Fig. 4g**). In contrast, MSPs encoding gluconate dehydratase generally showed a weaker association with gluconate abundance, with the exception of *Ruminococcus gnavus*, which is often associated with IBD and had a higher *t*-value than other dehydratase-encoding MSPs (new **Fig. 4g**).

We also analyzed HMP2, an adult IBD cohort characterized by a low disease signal and marginal differences in calprotectin scores between IBD patients and controls. In this dataset, no correlation was found between calprotectin levels and gluconate abundance. However, we observed a notable link with strong effect sizes between high gluconate abundance in subjects and an increased presence of *Enterobacteriaceae* and other MSPs encoding gluconate kinase operons (**Extended Data Fig. 14e, f**).

Conversely, the enrichment of alternative gluconate pathway genes in commensal bacteria was associated with non-IBD (**Extended Data Fig. 14c**).

Without establishing causality, but acknowledging the complex nature of the human cohorts and IBD pathogenesis, we additionally ran a simplified computational flux-based model analysis on a subset of microbial pangenomes assembled from the PROTECT cohort. By applying the *gapseq* [1] software with its default, well-defined healthy gut medium and parameters, we found that *Klebsiella pneumoniae* maintained a lower abundance in the mock community in the absence of gluconate (see below **Figure for reviewer only, left panel**). The introduction of gluconate resulted in a marked increase in *Klebsiella* abundance (**Figure for reviewer only, center panel**). The addition of another gluconate-consuming species (*Bifidobacterium adolescentis*) reduced the expansion of *Klebsiella*, computationally supporting the experimental results presented in the study (**Figure for reviewer only, right panel**).

Figure for reviewer only. Flux-based analysis with *gapseq* [1] for simulating gluconate metabolism in a mock community of assembled pangenomes from the PROTECT cohort. (Left) Baseline genome-scale reconstruction simulations with *gapseq*'s default gut medium, without gluconate indicates low abundance of *Klebsiella*. (Center) Upon addition of gluconate with diffusion speed of 0.04, *Klebsiella* becomes the most abundant species in the community. (Right) Introducing another gluconate consumer (*B. adolescentis*) into the gluconate-supplemented arena results in reduced *Klebsiella* abundance and an increase in *Bifidobacterium*.

Reference

[1] Zimmermann J, Kaleta C, Waschina S. *gapseq*: informed prediction of bacterial metabolic pathways and reconstruction of accurate metabolic models. *Genome Biol.* 2021 Mar 10;22(1):81. doi: 10.1186/s13059-021-02295-1.

Referee #2 (Remarks to the Author):

The manuscript by Furuichi et al. identified microbial consortia that suppress the growth of Klebsiella and other Enterobacteriaceae in monocolonized ex-germ-free mice. After identifying the consortium, the authors search for a mechanism by which these microbes can displace the pathobionts. The follow-up comprehensive experiments explore the role of host immunity and bacterial metabolites. Finally, a genetic screen using transposon mutagenesis in Klebsiella and follow-up studies led to the identification of the availability of some carbohydrates, and in particular gluconate, as a key factor for Klebsiella growth. The community of 18 strains reduced the availability of gluconate from Klebsiella, and excess gluconate administration rescued Klebsiella colonization.

Overall, this is an important study that identified a microbial community that can suppress the growth of Klebsiella and other Enterobacteriaceae. In the past decade, it has been appreciated how bacterial metabolism and carbohydrate utilization are essential for bacterial growth and colonization resistance. For example, see studies showing that Salmonella uses galactarate and glucarate (PMID: 27309805), that E. coli depletes galactitol to compete with Salmonella (PMID: 34610296), and that Klebsiella oxytoca and other commensals provide colonization resistance against Klebsiella pneumoniae via carbohydrate utilization, particularly of beta-glucosidic sugars (PMID: 34610293). Importantly, the latter study identified Blautia and Enterocloster as key organisms for competition with Klebsiella pneumoniae, and these bacteria were also present in the F18-mix of this manuscript. Prior manuscripts showing the importance of carbohydrate competition in colonization resistance against Enterobacteriaceae need to be acknowledged and discussed, and in particular, the Klebsiella study, which is particularly relevant to this manuscript.

The results presented are overall convincing. The data presentation is excellent, and the conclusions are supported by the data. I suggest a couple of experiments to strengthen the authors' conclusions further:

We thank the reviewer for highlighting the findings and importance of our study and for the insightful comments. We have addressed each comment individually below by conducting additional experiments and making necessary amendments. We hope the reviewer will agree that these changes greatly strengthen the manuscript.

1. All experiments were done in ex-germ-free mice colonized with the pathobiont and displaced by the consortia (e.g. F18-mix). While this is essential for understanding the mechanism, it would be important to repeat this study in SPF mice. For example, do the consortia (e.g. F18-mix) displace Klebsiella also in SPF IL-10KO mice?

We greatly appreciate the insightful feedback provided.

To test the effect of F18-mix in the context of a complex microbiota and to provide additional clinical relevance, we first explored the impact of F18-mix on other human-associated commensals. We selected seven representative commensal strains from our culture collection, comprising both gram-positive and gram-negative species reported to be prevalent in the healthy human gut microbiota, and simultaneously inoculated these strains and Kp-2H7 into GF mice. Seven days later, we administered F18-mix by oral gavage and monitored the faecal abundance of each strain over time using qPCR. All F18-mix-derived strains successfully colonized and persisted (new **Extended Data Fig. 4b**). Upon F18-mix administration, there was a notable decrease in Kp-2H7 abundance. The levels of Bacillota (*Dorea longicatena*, *Eubacterium rectale*, and *Clostridium scindens*) and Bacteroidetes strains (*Bacteroides thetaiotaomicron* and *Bacteroides uniformis*) remained largely stable, though there was a reduction in faecal levels of low-abundance strains (*Bifidobacterium adolescentis* and *Collinsella aerofaciens*) (new **Extended Data Fig. 4b, c**). Of note, both *B. adolescentis* and F18-mix utilizes

gluconate as a carbon source, and as such the reduction in *B. adolescentis* abundance may be secondary to metabolic competition (new **Extended Data Fig. 4d**).

To examine the impact of F18-mix on other commensals in the setting of a more complex microbiota, GF mice were colonized with either I41-mix, K46-mix, or a combination of these consortia together with a *Clostridium scindens* strain (totaling 88 strains), followed by oral administration of F18-mix. Overall, the commensal strains from donors I and K remained largely stable, although some low-abundance members, including *Bifidobacterium*, *Collinsella*, and *Megasphaera*, showed reductions (new **Extended Data Figure 4e** and new **Table S2**) [it is noteworthy that *Bifidobacterium* and *Megasphaera* carry gluconate kinase operon genes (see **Extended Data Fig. 13**) and may compete with F18-mix for gluconate]. Together, these data suggest that F18-mix can specifically reduce *Klebsiella* levels in the intestine without significantly affecting commensal communities.

To further probe clinical translatability, we tested the effect of F18-mix on *Enterobacteriaceae* decolonization in the context of an IBD-associated complex microbiota. GF mice were inoculated with faecal microbiota from patients with Crohn's disease (CD#15) or ulcerative colitis (UC#5) that exhibited enrichment of either *K. pneumoniae* or ESBL⁺ *E. coli*, respectively (new **Fig. 2b, c**). The mice were then treated with vancomycin to generate ecological niches amenable to F18-mix engraftment, followed by F18-mix gavage, mimicking a potential clinical treatment regimen for live biotherapeutic products. Gut microbiome composition was examined by full-length 16S rRNA gene sequencing (new **Fig. 2b, c**) and *K. pneumoniae* or *E. coli* burden was also determined by counting faecal CFUs (new **Extended Data Fig. 5a, 5b**). All F18 strains engrafted successfully within the IBD-associated microbial communities, resulting in a concomitant increase in microbial diversity and suppression of *K. pneumoniae* and *E. coli*. In contrast, vancomycin treatment alone failed to suppress these pathobionts (new **Fig. 2b, c** and **Extended Data Fig. 5a, 5b**). Thus, F18-mix can exert anti-*Enterobacteriaceae* activity in the context of several clinically relevant, complex microbiota.

We also examined whether F18-mix-mediated *Enterobacteriaceae* decolonization could prevent the induction of colitis. GF *I110*^{-/-} mice were colonized with UC#5 microbiota, and either F18-mix or F13-mix was orally administered 7 days later. F18-mix treatment successfully decolonized ESBL⁺ *E. coli* from the UC#5 microbiota and prevented intestinal inflammation (new **Fig. 2d-f**). Therefore, F18-mix is capable of reducing intestinal *Enterobacteriaceae* burden and alleviating IBD-like inflammation without disrupting the gut commensal community, suggesting high translational potential.

As suggested by the reviewer, we also examined the effects of F18-mix in SPF mice. SPF mice were pre-treated with antibiotics (ampicillin) and orally inoculated with Kp-2H7, followed by treatment

with either F13-mix, F18-mix, or F31-mix 7 days later. F18-mix and F31-mix tended to be efficacious at decolonizing Kp-2H7 (see **Figure for reviewer only**, to the right). However, spontaneous clearance of *Klebsiella* was often observed in SPF conditions due to recovery of the endogenous gut microbiota following antibiotic washout, making it difficult to clearly discern and interpret the effects of F18-mix treatment. Given that F18-mix comprises human-derived bacteria, we believe the results obtained from mice colonized with human faecal microbiota from IBD patients (new **Fig. 2b, c** and **Extended Data Fig. 5a, 5b**) to be more clinically relevant than those from SPF mice. Therefore, we have opted to omit the results of the SPF mouse experiments from the current manuscript.

2. In all experiments, mice were colonized with *Klebsiella* first, then *Klebsiella* was displaced by the F18-mix. What about the reverse experiment? Does the F18-mix prevent *Klebsiella* colonization?

In response to this comment, we designed an *in vivo* mouse model to assess the “colonization resistance” capacity F18-mix exerts against *Klebsiella*. GF mice were colonized with F18-mix, F13-mix, or left un-colonized, and were orally challenged 7 days later with 10⁴-10⁵ CFU of Kp-2H7. Faecal pellets from each mouse were collected longitudinally post-challenge to monitor the extent of Kp-2H7 colonization. In contrast to the untreated and F13-mix pre-treated groups, pre-treatment with F18-mix yielded robust Kp-2H7 suppression (new **Extended Data Fig. 3c**). Thus, F18-mix is capable of exerting colonization resistance activity against *Klebsiella*.

3. Administration of gluconate partly rescues *Klebsiella* intestinal growth in the presence of the F18-mix (Fig. 3h) What happens if other carbohydrates are administered to mice? I understand the rationale of focusing on gluconate, but it seems that other carbohydrates may also be important.

We agree with the reviewer’s suggestion to provide additional insights into the significance and complementarity of our study in relation to previous reports demonstrating the role of various carbohydrates in determining intestinal niches of *Enterobacteriaceae* species.

To examine the contribution of a diverse group of carbohydrates (including those mentioned by the reviewer) to the mechanism of F18-mix-mediated *Enterobacteriaceae* decolonization, we performed LC-MS-based faecal carbohydrate profiling of GF mice colonized with Kp-2H7, F18-mix, F13-mix, K46-mix, or I41-mix. This analysis revealed that Kp-2H7 monocolonization resulted in marked reductions in gluconate, galacturonate, glucuronate, N-acetylglucosamine, sorbitol, galactose/mannose, ribose, arabinose, and xylose (new **Extended Data Fig. 11b**), indicating that Kp-2H7 may have a preference for these carbon sources. Importantly, F18-mix efficiently utilized an overlapping set of carbohydrates (including gluconate), thereby potentially limiting nutrient availability and reshaping ecological niches within the intestine (new **Extended Data Fig. 11b**). In

contrast, F13-mix, K46-mix, and I41-mix demonstrated distinct carbohydrate utilization profiles. Notably, only gluconate consumption exhibited a clear correlation with the consortia's ability to suppress Kp-2H7 (new **Extended Data Fig. 11b**). These findings support the hypothesis that gluconate consumption/competition plays a crucial role in F18-mix-mediated *Enterobacteriaceae* decolonization.

As the reviewer highlighted, galactitol, cellobiose (a type of β -glucosidic sugar), and glucarate have been reported to be involved in the growth of *Klebsiella* in the intestine and may play a mechanistic role in F18-mix-mediated Kp-2H7 suppression (we have now included these important references in the revised manuscript). We therefore investigated whether providing these carbohydrates in excess would affect the ability of F18-mix to decolonize Kp-2H7. To this end, GF mice first monocolonized with Kp-2H7 and then treated with F18-mix. These mice were initially fed a nutrient-rich (CL-2) diet and were subsequently switched to a defined (AIN93G) diet. The AIN93G diet contains high levels of sucrose but lacks gluconate, sorbitol, glucosamine, and xylose. We supplemented the AIN93G diet with each of the following carbohydrates individually (final concentration, 10% of total calories): gluconate, galactitol, sorbitol, glucosamine, glucarate, galacturonate, cellobiose, mannose, xylose, or vehicle control. We found that supplementation with mannose, xylose, cellobiose, glucarate, or galacturonate did not affect Kp-2H7 sensitivity to F18-mix (new **Extended Data Fig. 12a**). In contrast, supplementation with galactitol, sorbitol, and glucosamine significantly increased Kp-2H7 levels (new **Extended Data Fig. 12a**), suggesting that these sugars may contribute to *Klebsiella* fitness *in vivo*.

Importantly, however, faecal galactitol levels were below the limit of detection regardless of whether mice were on the CL-2 or AIN93G diet using our validated LC-MS-based galactitol detection method (new **Extended Data Fig. 11b** and **12b**). Therefore, it is unlikely that competition for galactitol contributes to the observed suppression of *Klebsiella* by F18-mix. While intestinal sorbitol is abundant (**Fig. 3f** and new **Extended Data Fig. 11b**), it was not able to compensate for the restricted availability of other carbohydrates like gluconate, as Kp-2H7 was decolonized by F18-mix despite exhibiting more efficient sorbitol metabolism (new **Extended Data Fig. 11b**). Additionally, sorbitol was effectively consumed by K46-mix and I41-mix (new **Extended Data Fig. 11b**), both of which failed to fully suppress Kp-2H7 (**Fig. 1b**). Moreover, the expression of Kp-2H7 sorbitol metabolism genes was minimally perturbed by F18-mix treatment (new **Extended Data Fig. 9b**). These results suggest that, although sorbitol competition cannot be entirely ruled out, it is unlikely to be a primary driver of F18-mix-mediated *Klebsiella* suppression. On the other hand, it is possible that glucosamine regulation may contribute to *Klebsiella* suppression, especially given that GntR and GntK are involved in glucosamine metabolism (new **Extended Data Fig. 10b** and **10d**, see also **Supplemental Discussion**) which, much like gluconate metabolism, uses the ED pathway. However, unlike gluconate, which is derived from both dietary sources and the host, glucosamine originates almost exclusively from dietary intake and is present at levels approximately 20 times lower than gluconate in the intestine (new **Extended Data Fig. 11a**). Therefore, while competition for glucosamine may contribute, it is more plausible that *Klebsiella* predominantly relies on gluconate as a crucial energy source for growth in the gut, and that F18-mix competes for this resource.

The above results are intriguing. In particular, even when a large amount of sorbitol is available for

Kp-2H7 to utilize almost freely, it does not significantly affect the dynamics when there is competition for gluconate. However, in the absence of gluconate, if a substantial amount of sorbitol is present, Kp-2H7 can increase even in the presence of F18-mix. These new data suggest that while gluconate is clearly one of several carbon sources that can affect *Klebsiella* growth, its significance may extend beyond serving as a mere nutritional source. The presence and metabolism of gluconate may have cascading effects on microbial interaction patterns in the intestine. Gluconate metabolism may induce phenotypic changes in *Klebsiella* or alter the intestinal environment by, for instance, promoting changes in pH levels, producing inhibitory substances, or modulating signaling pathways. Such changes could potentially impact *Klebsiella*'s ability to utilize other carbohydrates, such as sorbitol and glucosamine, as effective energy sources for growth and survival. We have included these hypotheses in **Supplemental discussion** in the revised text.

Although more research is needed to fully elucidate the rules governing competition, our new findings—thanks to the reviewer's suggestion—indicate that the decolonization capacity of F18-mix is context-dependent and can be significantly influenced by dietary components. This consideration must be kept in mind when assessing its clinical efficacy. We have included this important point in the revised text.

Referee #3 (Remarks to the Author):

The study, titled "Rationally-defined microbial consortia suppress multidrug-resistant proinflammatory Enterobacteriaceae via ecological control," submitted by Furuichi et al., describes a minimal microbial community that suppresses Enterobacteriaceae abundance in gnotobiotic mice, along with a potential mechanism underlying this interaction. The authors initially identified complex communities from healthy donor stool samples that were most efficient in competing with Klebsiella pneumoniae Kp-2H7. They then employed an ecosystem deconstruction approach to determine which community members are required to suppress the growth of Kp-2H7. The authors found that a community of 18 strains (F18) was sufficient and essential for suppressing growth. This approach was complemented by generating a transposon mutant library of K. pneumoniae to identify potential mechanisms contributing to its suppression by F18. The authors identified gluconate as an important nutrient for K. pneumoniae in the intestine, which is depleted by the F18 community but not by less effective communities. A high gluconate diet also increased K. pneumoniae abundance. Finally, the authors present correlative data showing a high abundance of species containing gluconate-related genes in patients with inflammatory bowel disease (IBD).

Conceptually, identifying defined microbial intestinal communities that suppress specific community members, such as Enterobacteriaceae, is relevant given the overall negative impact of Enterobacteriaceae overgrowth on human health. The study is based on the concept that K. pneumoniae contributes to IBD, but unlike established pathogens like C. difficile/VRE that cause human disease, the potential adverse events of K. pneumoniae are still under investigation, and its relevance in humans remains to be shown. The study has several strengths, including a rigorous study design, a labor-intensive systematic interrogation of potential mechanisms using the transposon mutant library, transparent presentation of data, including experiments that did not lead to finding a mechanism, and identification of a potential pathway that can be targeted to suppress overgrowth of Enterobacteriaceae. However, there are some concerns outlined below.

We thank the reviewer for acknowledging the strength of our study and for helping point out sections that could be more thoroughly explored. We have conducted additional experiments to address these important points, and we hope the reviewer will find that these new data not only reinforce the conclusions we had drawn in the previous manuscript draft, but also offer novel insights into the commensal-mediated *Klebsiella* suppression.

Major Concerns:

1. The study primarily relies on microbial consortia suppressing K. pneumoniae in germ-free mice colonized with K. pneumoniae. It remains unclear if the same effect and mechanism would apply in the context of a complex community harboring high levels of K. pneumoniae. While the entire study does not need to be done in a complex community, the effect of F18 and verification of competition for gluconate should be demonstrated in the context of germ-free mice colonized with stool samples from IBD patients with high levels of K. pneumoniae.

As the reviewer suggests, we tested the effect of F18-mix on *Enterobacteriaceae* decolonization in the context of IBD-associated complex microbiotas. GF mice were inoculated with faecal microbiota from patients with Crohn's disease (CD#15) or ulcerative colitis (UC#5) that exhibited enrichment of either *K. pneumoniae* or ESBL⁺ *E. coli*, respectively (new **Fig. 2b, c**). The mice were then treated with vancomycin to generate ecological niches amenable to F18-mix engraftment, followed by F18-mix gavage, mimicking a potential clinical treatment regimen for live biotherapeutic products. Gut microbiome composition was examined by full-length 16S rRNA gene sequencing (new **Fig. 2b, c**) and *K. pneumoniae* or *E. coli* burden was also determined by counting faecal CFUs (new **Extended**

Data Fig. 5a, 5b). All F18 strains engrafted successfully within the IBD-associated microbial communities, resulting in a concomitant increase in microbial diversity and suppression of *K. pneumoniae* and *E. coli*. In contrast, vancomycin treatment alone failed to suppress these pathobionts (new **Fig. 2b, 2c** and **Extended Data Fig. 5a, 5b**). Thus, F18-mix can exert anti-*Enterobacteriaceae* activity in the context of clinically relevant, complex microbiota.

We also examined whether F18-mix-mediated *Enterobacteriaceae* decolonization could prevent the induction of colitis. GF *I110*^{-/-} mice were colonized with UC#5 microbiota, and either F18-mix or F13-mix was orally administered 7 days later. F18-mix treatment successfully decolonized ESBL⁺ *E. coli* from the UC#5 microbiota and prevented intestinal inflammation (new **Fig. 2d-f**). Therefore, F18-mix is capable of reducing intestinal *Enterobacteriaceae* burden and alleviating IBD-like inflammation, suggesting high translational potential.

2. One limitation with this approach that needs to be acknowledged is that while F18 reduces gluconate availability, it is not the only difference between F18 and other communities tested by the authors. There could be other mechanisms that are concurrently affected by the F18 community, contributing to the suppression of K. pneumoniae. Ideally, this would require knocking down the gluconate pathway in members of the F18 community, but that can be very tedious.

We agree with this reviewer's suggestion to acknowledge the limitations of our study and to provide additional insights into other potential factors contributing to intestinal control and niche dynamics of *Enterobacteriaceae* species.

To examine the contribution of a diverse group of carbohydrates to the mechanism of F18-mix-mediated *Enterobacteriaceae* decolonization, we performed LC-MS-based faecal carbohydrate profiling of GF mice colonized with Kp-2H7, F18-mix, F13-mix, K46-mix, or I41-mix. This analysis revealed that Kp-2H7 monocolonization resulted in marked reductions in gluconate, galacturonate, glucuronate, N-acetylglucosamine, sorbitol, galactose/mannose, ribose, arabinose, and xylose (new **Extended Data Fig. 11b**), indicating that Kp-2H7 may have a preference for these carbon sources. Importantly, F18-mix efficiently utilized an overlapping set of carbohydrates (including gluconate), thereby potentially limiting nutrient availability and reshaping ecological niches within the intestine (new **Extended Data Fig. 11b**). In contrast, F13-mix, K46-mix, and I41-mix demonstrated distinct carbohydrate utilization profiles. Notably, only gluconate consumption exhibited a clear correlation with the consortia's ability to suppress Kp-2H7 (new **Extended Data Fig. 11b**). These findings support the hypothesis that gluconate consumption/competition plays a crucial role in F18-mix-mediated *Enterobacteriaceae* decolonization.

Of note, galactitol, cellobiose (a type of β -glucosidic sugar), and glucarate have been reported to be involved in the growth of *Klebsiella* in the intestine and may play a mechanistic role in F18-mix-mediated Kp-2H7 suppression (we have now included these important references in the revised manuscript). We therefore investigated whether providing these carbohydrates in excess would affect the ability of F18-mix to decolonize Kp-2H7. To this end, GF mice first monocolonized with Kp-2H7 and then treated with F18-mix. These mice were initially fed a nutrient-rich (CL-2) diet and were

subsequently switched to a defined (AIN93G) diet. The AIN93G diet contains high levels of sucrose but lacks gluconate, sorbitol, glucosamine, and xylose. We supplemented the AIN93G diet with each of the following carbohydrates individually (final concentration, 10% of total calories): gluconate, galactitol, sorbitol, glucosamine, glucarate, galacturonate, cellobiose, mannose, xylose, or vehicle control. We found that supplementation with mannose, xylose, cellobiose, glucarate, or galacturonate did not affect Kp-2H7 sensitivity to F18-mix (new **Extended Data Fig. 12a**). In contrast, supplementation with galactitol, sorbitol, and glucosamine significantly increased Kp-2H7 levels (new **Extended Data Fig. 12a**), suggesting that these sugars may contribute to *Klebsiella* fitness *in vivo*.

Importantly, however, faecal galactitol levels were below the limit of detection regardless of whether mice were on the CL-2 or AIN93G diet using our validated LC-MS-based galactitol detection method (new **Extended Data Fig. 11b** and **12b**). Therefore, it is unlikely that competition for galactitol contributes to the observed suppression of *Klebsiella* by F18-mix. While intestinal sorbitol is abundant (**Fig. 3f** and new **Extended Data Fig. 11b**), it was not able to compensate for the restricted availability of other carbohydrates like gluconate, as Kp-2H7 was decolonized by F18-mix despite exhibiting more efficient sorbitol metabolism (new **Extended Data Fig. 11b**). Additionally, sorbitol was effectively consumed by K46-mix and I41-mix (new **Extended Data Fig. 11b**), both of which failed to fully suppress Kp-2H7 (**Fig. 1b**). Moreover, the expression of Kp-2H7 sorbitol metabolism genes was minimally perturbed by F18-mix treatment (new **Extended Data Fig. 9b**). These results suggest that, although sorbitol competition cannot be entirely ruled out, it is unlikely to be a primary driver of F18-mix-mediated *Klebsiella* suppression. On the other hand, it is possible that glucosamine regulation may contribute to *Klebsiella* suppression, especially given that GntR and GntK are involved in glucosamine metabolism (new **Extended Data Fig. 10b** and **10d**, see also **Supplemental Discussion**) which, much like gluconate metabolism, uses the ED pathway. However, unlike gluconate, which is derived from both dietary sources and the host, glucosamine originates almost exclusively from dietary intake and is present at levels approximately 20 times lower than gluconate in the intestine (new **Extended Data Fig. 11a**). Therefore, while competition for glucosamine may contribute, it is more plausible that *Klebsiella* predominantly relies on gluconate as a crucial energy source for growth in the gut, and that F18-mix competes for this resource.

The above results are intriguing. In particular, even when a large amount of sorbitol is available for Kp-2H7 to utilize almost freely, it does not significantly affect the dynamics when there is competition for gluconate. However, in the absence of gluconate, if a substantial amount of sorbitol is present, Kp-2H7 can increase even in the presence of F18-mix. While gluconate is clearly one of several carbon sources that can affect *Klebsiella* growth, its significance may extend beyond serving as a mere nutritional source. The presence and metabolism of gluconate may have cascading effects on microbial interaction patterns in the intestine. Gluconate metabolism may induce phenotypic changes in *Klebsiella* or alter the intestinal environment by, for instance, promoting changes in pH levels, producing inhibitory substances, or modulating signaling pathways. Such changes could potentially impact *Klebsiella*'s ability to utilize other carbohydrates, such as sorbitol and glucosamine, as effective energy sources for growth and survival. We have included these hypotheses in **Supplemental**

discussion in the revised text.

Although more research is needed to fully elucidate the rules governing competition—for instance, by testing the effect of gene disruption of gluconate dehydratase operon genes from F18-mix, as the reviewer suggested but is currently not feasible—our new findings indicate that the decolonization capacity of F18-mix is context-dependent and can be significantly influenced by dietary components. This consideration must be kept in mind when assessing its clinical efficacy. We have included this important point in the revised text.

3. In the absence of experiments using *K. pneumoniae* in the context of complex communities mentioned above in point #1, it remains unclear if there are any off-target positive or negative effects of the F18 community on the rest of the microbial community.

We agree with the reviewer that it is important to investigate whether the F18 community has any off-target effects on the rest of the gut microbiota community. Accordingly, we tested the effect of F18-mix on other human-associated commensals. We first selected seven representative commensal strains from our culture collection, comprising both gram-positive and gram-negative species reported to be prevalent in the healthy human gut microbiota, and simultaneously inoculated these strains and Kp-2H7 into GF mice. Seven days later, we administered F18-mix by oral gavage and monitored the faecal abundance of each strain over time using qPCR. All F18-mix-derived strains successfully colonized and persisted (new **Extended Data Fig. 4b**). Upon F18-mix administration, there was a notable decrease in Kp-2H7 abundance. The levels of Bacillota (*Dorea longicatena*, *Eubacterium rectale*, and *Clostridium scindens*) and Bacteroidetes strains (*Bacteroides thetaiotaomicron* and *Bacteroides uniformis*) remained largely stable, though there was a reduction in faecal levels of low-abundance strains (*Bifidobacterium adolescentis* and *Collinsella aerofaciens*) (new **Extended Data Fig. 4b, c**). Of note, both *B. adolescentis* and F18-mix utilizes gluconate as a carbon source, and as such the reduction in *B. adolescentis* abundance may be secondary to metabolic competition (new **Extended Data Fig. 4d**).

To examine the impact of F18-mix on other commensals in the setting of a more complex microbiota, GF mice were colonized with either I41-mix, K46-mix, or a combination of these consortia together with a *Clostridium scindens* strain (totaling 88 strains), followed by oral administration of F18-mix. Overall, the commensal strains from donors I and K remained largely stable, although some low-abundance members, including *Bifidobacterium*, *Collinsella*, and *Megasphaera*, showed reductions (new **Extended Data Figure 4e** and new **Table S2**) [it is noteworthy that *Bifidobacterium* and *Megasphaera* carry gluconate kinase operon genes (see **Extended Data Fig. 13**) and may compete with F18-mix for gluconate]. Together, these data suggest that F18-mix can specifically reduce *Klebsiella* levels in the intestine without significantly affecting commensal communities.

4. The authors use human data from a published cohort to corroborate some of the findings. The correlation of gluconate and calprotectin is nominally significant ($p=0.0377$) with a very small effect size ($r=0.23$). As Enterobacteriaceae are known to be elevated in IBD, the high abundance of species containing gluconate-related genes is difficult to interpret, as Enterobacteriaceae harbor these genes. The authors should consider comparing patients with

and without high levels of K. pneumoniae to determine if gluconate pathways correlate with their prevalence.

We acknowledge the reviewer's concern that the correlation between gluconate and calprotectin is modest in our PROTECT cohort analysis and that it does not achieve statistical significance within the HMP2 cohort. We also recognize that the expansion of *Enterobacteriaceae* in inflammatory conditions is somewhat expected and has been associated with biochemical mechanisms other than gluconate metabolism, such as nitrate respiration.

Accordingly, to evaluate the existence of an association between gluconate abundance and *Enterobacteriaceae* enrichment by accounting for the extent of inflammation within the statistical framework, we first validated the gluconate candidate in our stool metabolomics data by using the following chemical reference: Sigma Aldrich S2054 (details in **Methods**). This led to annotating HN_s_QI1923 (HILIC-neg 195.0512 *m/z* at 4.34 min) from PROTECT and QI11027 (HILIC-neg 195.0512 *m/z* at 4.48 min) from HMP2. Then, we utilized a general linear mixed-effects model to assess the association between species (MSPs) and gluconate abundance, factoring in faecal calprotectin and subject ID as fixed and random effects, respectively. In the PROTECT cohort, our analysis revealed a significant positive association between gluconate and *Enterobacteriaceae* abundance, even after adjusting for faecal calprotectin levels (new **Fig. 4f**). This association was notably strong and achieved statistical significance specifically in both *Enterobacteriaceae* and non-*Enterobacteriaceae* that possess gluconate kinase operon genes (new **Fig. 4g**). In general, MSPs encoding both the gluconate kinase and transporter together had higher *t*-values, and thus stronger associations with gluconate and lower error rates, compared to species with other gene combinations. In contrast, MSPs encoding gluconate dehydratase generally showed a weaker association with gluconate abundance, with the exception of *Ruminococcus gnavus*, which is often associated with IBD and had a higher *t*-value than other dehydratase-encoding MSPs.

We also examined the adult IBD cohort HMP2 (**Extended Data Fig. 14a**). Once again, IBD was associated with an expansion of gluconate kinase operon-carrying *Enterobacteriaceae* species (new **Extended Data Fig. 14c-f**). In particular, *E. coli*, *C. freundii*, and *K. pneumoniae* consistently emerged as significantly more prevalent in individuals with disease within the HMP2 cohort (**Extended Data Fig. 14c**), mirroring trends seen in the PROTECT cohort. Furthermore, even when faecal calprotectin levels were controlled for in a linear mixed-effects model, gluconate level was still significantly associated with abundance of MSPs carrying gluconate kinase operon genes, including *Enterobacteriaceae* (new **Extended Data Fig. 14e, f**). Conversely, the enrichment of gluconate dehydratase operons in commensal bacteria was associated with non-IBD (**Extended Data Fig. 14c**), suggesting that these genes may facilitate metabolic competition and suppress proinflammatory pathobionts, thereby maintaining gastrointestinal homeostasis.

Minor Concerns:

1. While more semantics, I would not consider these as rationally defined communities, as they were identified after trying all combinations of isolated strains to determine which ones would suppress *K. pneumoniae* and then finding a potential mechanism. Rational selection would indicate that communities were developed based on a mechanism known to be important for suppressing *K. pneumoniae*.

We appreciate this comment. Accordingly, we have removed the word “rationally” from the title, Introduction, and Discussion.

2. In line 123, *E. coli* and *Fusobacterium* are included, as well as antibiotic-resistant taxa such as *Enterocloster* sp. Are these strains relevant to give to people since they are considered pathobionts? Please add this to the discussion.

We appreciate the reviewer’s concerns regarding the safety of our consortium, which includes *E. coli*, *Fusobacterium*, and *Enterocloster* strains. Accordingly, we analysed the genomes of all F18 strains for virulence factors [using the Virulence Factor Databases (VFDB) with identity >70% and coverage length >70%] and antibiotic resistance genes. Although tetracycline-resistance genes were present in most of the genomes, none of the strains were multidrug-resistant (new **Table S5**). Furthermore, our investigation did not detect any prominent virulence factors or toxins (new **Table S4**). Although some of the F18 strains carry genes encoding potentially virulent genes such as capsular polysaccharide synthesis enzymes, catalases, and transporter genes, most of these genes are encoded by other commensal species as well. For example, in the genome of our f37_ *E. coli* strain, 128 potential virulence genes were identified; however, these were highly similar to the potential virulence genes identified in *E. coli* Nissle1917 strain. Furthermore, our f37_ *E. coli* strain does not carry the *pks* island. Because the F18 strains originate from a healthy microbiota, do not carry known toxins, and are not multidrug-resistant, we believe they could serve as the promising foundation for a live biotherapeutic product.

Furthermore, we explored the possibility of swapping out our f37_ *E. coli* strain for the Nissle1917 strain, which has been clinically proven to be safe. We confirmed that *E. coli* Nissle1917 efficiently utilizes gluconate *in vitro* (data not shown) and observed no difference in F18-mix-mediated *Klebsiella* decolonization capacity regardless of which *E. coli* strain was included (new **Extended data Fig. 3e**). Thus, it appears that the therapeutically-active consortium is not necessarily limited to the strains isolated in this study and that safety concerns related to clinical translation can potentially be mitigated by selecting functionally similar strains with established safety profiles. We have included these new data in the revised text.

3. In line 233, instead of mutants, please say "transposon insertion mutants" for clarity.

4. In line 266, the header should start at the subsequent paragraph.

5. In line 296, there is a typo: "Clostiridium" should be corrected to "Clostridium."

6. Figure 4e and Ext. data Figure 12C are too small; adjust for legibility.

7. In Ext. Data Figure 6, there is a typo: "suspention" should be corrected to "suspension."

Thank you for pointing out those errors. We have made the necessary amendments to the manuscript.

Reviewer Reports on the First Revision:

Referees' comments:

Referee #1 (Remarks to the Author):

Furuichi and colleagues have carefully and thoughtfully addressed the issues raised by reviewers. The authors have conducted additional experiments that, on the one hand, are clarifying while, on the other hand, reveal a new layer of complexity that is interesting and partially untangled. As noted previously, GntR deletion enhances gluconate driven Kp growth but results in enhanced clearance of Kp by the F18 consortium. By conducting additional experiments in response to the first review, the authors discovered that gntR also enhances glucosamine metabolism. This surprising and fascinating finding provides the authors with an alternative explanation for how gntR deficiency enhances the effectiveness of the F18 consortium: reduced ability of Kp to metabolize glucosamine. While the authors hypothesize that deletion of gntR may limit Kp growth in vivo by reducing its ability to metabolize glucosamine, they bury this fascinating idea in the supplementary discussion. It would be better to mention this idea in the main text of the manuscript. An additional layer of complexity was revealed by their analyses of Kp colonization and the presence of gluconate in the small intestine, leading them to point out that colonization resistance by carbohydrate depletion is compartmentalized.

Overall, the authors have provided a deeper, more comprehensive picture of commensal bacterial suppression of Kp persistence in the gut. In the process, their work has raised additional questions about the regulation of carbohydrate metabolism by Kp and the impact of commensal bacteria in different sections of the gut. Nevertheless, this study, by combining transcriptomics, transposon mutagenesis, metabolomics and carbohydrate quantitation, provides the most complete picture to date of an important mechanism of microbiota-mediated suppression of Klebsiella in the gut.

Referee #2 (Remarks to the Author):

The authors have addressed my prior concerns and have performed several additional experiments that further strengthen the manuscript's conclusions. I confirm my positive evaluation of this work.

Referee #3 (Remarks to the Author):

all concerns adequately addressed